# Non methane hydrocarbon (NMHC) fingerprints of major urban and agricultural emission sources for use in source apportionment studies

Ashish Kumar[1], Vinayak Sinha[1]*, Muhammed Shabin[1], Haseeb Hakkim[1], Bernard Bonsang[2] and Valerie Gros[2]

[1] Department of Earth and Environmental Sciences, Indian Institute of Science Education and Research Mohali, Sector 81, S.A.S Nagar, Manauli PO, Punjab, 140306, India.

[2] LSCE, Laboratoire des Sciences du Climat et de l'Environnement, CNRS-CEA-UVSQ, IPSL, Université Paris Saclay, Orme des Merisiers, F91191 Gif-sur-Yvette, France.

*Correspondence to*: Dr. Vinayak Sinha (vsinha@iisermohali.ac.in)

**Abstract**

In complex atmospheric emission environments such as urban agglomerates, multiple sources control the ambient chemical composition driving air quality and regional climate. In contrast to pristine sites, where reliance on single or few chemical tracers is often adequate to resolve pollution plumes and source influences, comprehensive chemical fingerprinting of sources using non-methane hydrocarbons and identification of suitable tracer molecules and emission ratios becomes necessary. Here, we characterize and present "chemical fingerprints" of some major urban and agricultural emission sources active in South Asia such as paddy stubble burning, garbage burning, idling vehicular exhaust and evaporative fuel emissions. A total of 121 whole air samples were collected actively from the different emission sources in passivated air sampling steel canisters and then analysed for 49 NMHCs (22 alkanes, 16 aromatics, 10 alkenes and 1 alkyne) using thermal desorption gas chromatography flame ionisation detection. Several new insights were obtained. Propane was found to be present in paddy stubble fire emissions (8%) and therefore for an environment impacted by crop residue fires, use of propane as a fugitive LPG emission tracer must be done with caution. Propene was found to be ~1.6 times greater (by weight) than ethene in smouldering paddy fires. Compositional differences were observed between evaporative emissions of domestic liquefied petroleum gas (LPG) and commercial LPG, which are used in South Asia. While the domestic LPG vapours had more propane ($40 \pm 6\%$) than $n$-butane ($19 \pm 2\%$), the converse were true for commercial LPG vapours ($7 \pm 6\%$ and $37 \pm 4\%$ respectively). Isoprene was identified as a new tracer for distinguishing paddy stubble and garbage burning in the absence of isoprene emissions at night from biogenic sources. Analyses of source specific inter-NMHC molar ratios revealed that toluene/benzene ratios can be used for distinguishing among paddy stubble fire emissions in flaming ($0.38 \pm 0.11$) and smouldering stages ($1.40 \pm 0.10$), garbage burning flaming ($0.26 \pm 0.07$) and smouldering emissions ($0.59 \pm 0.16$) and traffic emissions ($3.54 \pm 0.21$), whereas

*i*-pentane/*n*-pentane can be used distinguishing biomass burning emissions (0.06-1.46) from the petrol dominated traffic/fossil fuel emissions (2.83-4.13). *i*-butane/*n*-butane ratios were similar (0.20-0.30) for many sources and so could be used as a tracer for photochemical ageing. In agreement with previous studies, *i*-pentane, propane and acetylene were identified as suitable chemical tracers for petrol vehicular and evaporative emissions, LPG evaporative and vehicular emissions, and flaming stage biomass fires, respectively. The secondary pollutant formation potential and human health impact of the sources was also assessed in terms of their OH reactivity ($s^{-1}$), ozone formation potential (OFP, $gO_3/gNMHC$) and fractional BTEX content. Petrol vehicular emissions, paddy stubble fires and garbage fires were found to have higher pollution potential (at ≥95 % confidence interval), relative to the other sources studied in this work. Thus, many results of this study provide a new foundational framework for quantitative source apportionment studies in complex emission environments.

## 1 Introduction

Non-methane hydrocarbons (NMHCs) are an important class of volatile organic compounds (VOCs) that drive atmospheric chemistry and contribute towards formation of tropospheric ozone and secondary organic aerosols (SOA) (Poisson et al., 2000; Hallquist et al., 2009; Derwent et al., 2010; Ortega et al., 2016). Ground level ozone affects ambient air quality, human health and climate thus making it a primary target in air quality regulations (EPA, 1990). Furthermore, by reacting with hydroxyl radical (OH) they can also affect the oxidative capacity of the atmosphere (Atkinson, 2000). NMHCs have a wide variety of anthropogenic, pyrogenic and biogenic sources. In the urban areas, anthropogenic sources such as vehicular emissions, industries and fugitive solvent evaporation dominate the emissions (Barletta et al., 2005; Baker et al., 2008; Kansal, 2009; Jaimes-Palomera et al., 2016). However, in an agrarian and developing economy like India and

other parts of South Asia, other major anthropogenic activities like crop residue burning and garbage burning have emerged as poorly regulated emission sources. Every year the North West-Indo Gangetic Plain (NW-IGP) experiences episodes of large scale open burning of paddy stubble in the post-harvest months of October and November wherein > 12,685 $km^2$ of area of Punjab alone is estimated to be burnt in the open farm fields (Badarinath et al., 2006). This results in emission of a large number of gaseous and particulate pollutants into the air and causes severe deterioration in regional air quality (Sarkar et al., 2013; Chandra and Sinha, 2016; Kumar et al., 2016; Garg et al., 2016; Kumar et al., 2018; Sharma et al., 2019).

Previous studies have characterised the emissions of selected VOCs, greenhouse gases and primary air pollutants like benzenoids, carbon monoxide, nitrogen oxides and black carbon (Venkataraman et al., 2006; Sahai et al., 2007) from paddy stubble burning over NW-IGP. However, there is still a considerable deficit in knowledge concerning the speciated non-methane hydrocarbons, which are co-emitted in the smoke (Andreae, 2019; Sinha et al., 2019). The NMHC emissions from different sources when expressed as emission source profiles (Watson et al., 2001; Hong-li et al., 2017) provide detailed insights for quantitative source apportionment in source receptor models. Moreover, they are helpful for assessing human health risks due to exposure to toxic and hazardous compounds and secondary pollutant formation tendencies, and therefore assist in prioritization of pollution control strategies and policies.

The ratio of two NMHCs with different chemical lifetimes can also be used for constraining the photochemical age of air masses and atmospheric transport times (Parrish et al., 1992; McKeen and Liu, 1993). Several source profiles have been compiled for different emission sources in North America (Dallmann et al., 2012; Gentner et al., 2012), Europe (Passant, 2002; Niedojadlo et al., 2007), East Asia (Na et al., 2004; Liu et al., 2008; Zhang et al., 2013; Zheng et al., 2013; Mo et

al., 2016; Hong-li et al., 2017) and other areas (Doskey et al., 1999), however there is still considerable gap in data of speciated NMHCs from active emission sources in South Asia. Using the emission source profiles from different regions of the world for modelling and emission inventories can result in large uncertainties as the emissions can change from country to country depending upon the quality and composition of fuels, combustion practices and vehicular fleet. Therefore, it is essential to have a comprehensive database of regional and local source profiles which can be used to yield more accurate data for calculation of emissions and source apportionment tools such as positive matrix factorization.

In this study, we report the NMHC fingerprinting of paddy stubble burning emissions, garbage burning emissions, fuel evaporative emissions and idling exhaust emissions of vehicles powered by liquefied petroleum gas (LPG), compressed natural gas (CNG), diesel and petrol using 49 speciated NMHCs (22 alkanes, 16 aromatics, 10 alkenes and 1 alkyne). These compounds were measured using thermal desorption gas chromatography flame ionisation detection (TD-GC-FID). Based on the measured source profiles, chemical tracers were identified for distinguishing varied emission sources and also for use in PMF source apportionment models. Further, we assessed the secondary pollutant formation potential and health risks of the sources in terms of their OH reactivity ($s^{-1}$), ozone formation potential (OFP, $gO_3/gNMHC$) and fractional BTEX (sum of benzene, toluene, ethylbenzene and xylenes) content.

**2 Materials and Methods**

**2.1 Whole air sampling from specific sources in passivated steel canisters**

Table 1 summarizes the details of the whole air sample collection experiments for emissions from paddy stubble burning, garbage burning, busy traffic junctions, idling vehicular exhaust emissions

and fuel evaporation. The paddy stubble burning samples (three flaming and smouldering each) were collected at an agricultural field in Kurari, Mohali (30.605º N, 76.744º E) on 4 Nov 2017 between 16:30-18:30 LT. The garbage burning samples (five flaming and smouldering each) were collected at waste sorting and disposing stations in Mohali and surrounding villages (30.642-30.699ºN, 76.713-76.729ºE) between 7 - 17 February 2017. Figure S1 shows the flaming and smouldering fires, which were distinguished by visual inspection of presence of flame and white smoke as per previous studies (Chandra et al., 2017; Kumar et al., 2018). The fire in flaming stage showed clear flame with little smoke while in the smouldering stage there was white smoke and no flame. The traffic samples were collected from three busy traffic junctions in Chandigarh and Mohali (Sohana Gurudwara Chowk, 30.691º N, 76.698º E; Sector 79/80 Chowk, 30.678º N, 76.721º E; and Transport Chowk, 30.717º N, 76.812º E) from 3-15 March 2017. Although the vehicular emissions are known to be dependent upon several factors, their idling operation results in quite high emissions and fuel residues in the exhaust (Yamada et al., 2011; Shancita et al., 2014). This is because in idling operations the engine does not work at its peak operating temperature and efficiency (Brodrick et al., 2002) resulting in incomplete fuel combustion (Rahman et al., 2013). In this study, prior to vehicular exhaust sampling, the engine was left running for about 5 minutes until it warmed up to normal working temperature (70-90ºC) and then the air was sampled directly from the mouth of the tailpipe exhaust with the car in stationary position and engine running at idle speed. The idling vehicular exhaust samples were collected from 23 petrol vehicles (14 two wheelers and 9 light duty four wheelers), 33 diesel vehicles (6 three wheelers, 12 light duty four wheeler and 15 heavy duty wheelers), 9 LPG vehicles (three wheelers) and 7 CNG vehicles (6 three wheelers and 1 light duty four wheeler) from Mar 2017-Oct 2018 in Chandigarh and Mohali (30.660-30.750ºN, 76.700-76.840ºE). For a better representation, the most common vehicle

models on Indian roads were selected for this study based upon the personal field observations and motor vehicle data provided by Ministry of Road Transport and Highways (MoRTH 2017). The fuel evaporative emissions samples (ten each from the headspace of LPG, petrol and diesel) were collected in Mohali, Chandigarh and Panchkula on 13-14 Aug 2020. In India, the commonly used

LPG is of two types: domestic LPG for household cooking and commercial LPG for various commercial and industrial activities like hotels, restaurants, metallurgical applications, textiles, automotive etc. Out of the total 10 samples of LPG evaporative emissions, 5 samples each were of domestic and commercial LPG. For better representation, the samples of evaporative emissions were also collected from the most common brands of petrol, diesel and LPG fuels sold all over

India and in Nepal, Bangladesh and Sri Lanka (Indian Oil, Hindustan Petroleum, Bharat Petroleum, Bharatgas and Indane). In addition, prior to the lighting of fires or turning on the engine, ambient air samples were also collected from the aforementioned sites to correct for ambient background concentrations.

The whole air was actively sampled in commercially available 6L passivated SilcoCan air

sampling steel canisters (Restek, USA) and then analyzed using a thermal desorption gas chromatograph equipped with a flame ionisation detector (TD-GC-FID) within one day of sample collection as per collection procedure described in previous works (Chandra et al., 2017; Vettikkat et al., 2019). Stability tests of compounds in the canisters were also conducted which showed that all the measured compounds reported in this work including alkenes and alkyne remained stable

for upto 3 days. The air was sampled actively into the canisters using a Teflon VOC pump (Model − N86 KT.45.18; KNF Germany) operating at a flow rate of ~5500 ml/min and pressurized upto 30 psi. The steel canisters were protected from dust and air particles using a Teflon membrane filter (Pore size 0.45µm) in the sample inlet line. Prior to each sampling the canisters were cleaned

and preconditioned as per the EPA Method TO-15 using TO-Clean canister cleaner (Wasson ECE Instrumentation, Colorado, USA) and humidified nitrogen.

## 2.2 NMHC measurements by Thermal Desorption - Gas Chromatography-Flame Ionisation Detection (TD-GC-FID):

NMHCs in the sample air were measured using a gas chromatograph equipped with two flame ionisation detectors (GC-FID 7890B, Agilent Technologies, Santa Clara, United States). Sampling and pre-concentration was performed using a thermal desorption (CIA Advantage-HL and Unity 2, Markes International, UK) unit coupled to the GC-FID system. Helium (99.999% pure, Sigma gases, India) was used as the carrier gas. Hydrogen (99.9995%, Precision Hydrogen 100 - $H_2$

Generator, Peak Scientific, Scotland, UK), Synthetic air (99.999%, Sigma gases, India) and Nitrogen (99.9995%, Precision Nitrogen Trace 250 – $N_2$ Generator, Peak Scientific, Scotland, UK) were used as the FID gases (Table S1). Synthetic air (99.9995%, Precision Zero Air 1.5 - Gas Generator, Peak Scientific, Scotland, UK) was also used as the purge gas for the Markes Thermal desorption unit.

Figure S2 shows the schematic representation of the TD-GC-FID instrument during a typical sample injection and chromatographic run. In the first stage, sample air was passed through a Nafion dryer (integrated in CIA Advantage) to remove water (Badol et al., 2004; Gros et al., 2011). It was then preconcentrated at -30°C (maintained by a Peltier cooling system) at 20 ml/min, on an Ozone precursor trap (U-T17O3P-2S, Markes Internatioal, UK). The trap was a 2 mm-internal

diameter, 60 mm-long quartz tube containing Tenax TA, Carboxen 1003 and Carbosieve SIII as adsorbents. The preconcentrated trap was thermally desorbed by heating the trap rapidly to 325°C and held at this temperature for 20 min, so that all the preconcentrated NMHCs were thermally desorbed. Thermally desorbed NMHCs were then transferred via a heated inlet (130°C) line onto

the GC instrument consisting of two capillary columns (DB-1, dimethyl polysiloxane, 60 m x 0.25 mm, 1.00 µm film thickness; Alumina Plot, $Al_2O_3/Na_2SO_4$, 50 m x 0.32 mm, 8 µm film thickness, Agilent Technologies, Santa Clara, United States). Table S1 lists the settings at which the FIDs were operated and the oven temperature was ramped. Initially a temperature of 30$^o$C was maintained for 12 min and thereafter it was increased at two subsequent rates of 5$^o$C/min (upto 170$^o$C) and 15$^o$C/min (upto 200$^o$C). The two columns were connected via a Dean's switch which was turned on after 17 min of the chromatographic run. In these initial 17 min, the two columns were connected to each other in series and the eluents from the first column (DB-1) were directed onto the second column (Alumina-PLOT). After 17 min, the series connection between both the columns was broken by turning on the Dean's switch and the eluents from both columns were directed onto their respective FIDs. C6 and higher NMHCs were resolved on DB-1 column and detected on FID 1 while C2-C5 NMHCs were resolved on Alumina PLOT column and detected on FID 2. Thus, in a single run, C2-C10 compounds were measured simultaneously in two chromatograms.

Prior to the sampling, the instrument was calibrated by dynamic dilution with zero air at different mixing ratios (in the range of 2–200 ppb) using a standard gas calibration unit (GCU-s v2.1, Ionimed Analytik, Innsbruck, Austria). A NIST calibrated flow meter (BIOS Drycal definer 220) was used to measure the flows of both the standard gas and zero air mass flow controllers before and after the calibration experiments. Figure S3 shows the sensitivity and linearity of NMHCs obtained from the calibration experiments performed over a dynamic range of 2-200 ppb over two sets of calibrations: regular calibration of 2-20 ppb and a high mixing ratio calibration of 10-200 ppb. This covers a range of two orders of magnitude over which the instrument exhibited an excellent linearity ($r^2$>0.99) for all the 49 NMHCs (TD-GC-FID). Figure S4 shows a typical

chromatogram of the standard gas during the calibration experiment. Peak identification and quantification were performed using the PC software (Agilent OpenLAB CDS, Chemstation Edition, Rev. C.01.06 (61)). Further supplementary material 2 provides an example chromatogram for each source that was sampled. All the chromatograms were manually inspected to ensure correct peak identification, baseline determination and peak area calculation. The FID signal of a compound was recorded in form of current (picoampere, pA) by the instrument and area under the peak was calculated and expressed in units of pAs (picoampere seconds) and used to quantify the analyte. Individual peak areas (pAs) were converted to ppb using the sensitivity factors obtained from calibration experiments. For highly concentrated samples, appropriate dilution was performed prior to sample injection so that the measured concentrations were within the range of 5-30 ppb for most of the compounds. However, there were still a few compounds that were 50-200 ppb in some sources even after dilution (Table S2). The instrument linearity was therefore tested at high concentrations of upto 200 ppb and excellent linearity ($r^2 \geq 0.99$) was observed for all the compounds (Figure S3).

The supplementary material 3 (Supplement3 in excel file format) provides details of the measured mixing ratios for each individual sample measured by the TD-GC-FID system after dilution, mixing ratios of the compound in the actual sample after correcting for dilution alongwith uncertainty, as well as the values in the corresponding background samples. For the major compounds determining the normalised source profiles (presented and discussed in Section 3.1), the sample values were significantly higher than the background values (even by an order of magnitude or more for smoke and vehicular exhaust source categories). Therefore, while the background values were used to calculate excess concentrations, they hardly played any role in the determination of the emission profiles. The peaks in the chromatograms of the emission sources

were also well resolved and separated and were identified using the calibration gas standards. In case a shoulder peak was present then the parent peak was separately integrated, i.e., any interference from shoulder peak was subtracted from the parent signal. In the calibration gas standard some additional compounds were also present namely, 2,2,4-trimethylpentane, 2,3,4-trimethylpentane and methylcyclohexane, each of which had well resolved and separate peak during the calibration experiments. However, during the analysis of emission source samples, these compounds exhibited poor peak features like peak shape, several shoulder peaks, etc. Therefore, for remaining consistent across all samples, these compounds were excluded from analysis, and only those compounds were included that were well resolved. Table S3 lists the details of two VOC gas standards; 1) Gas standard (Chemtron Science Laboratories Pvt. Ltd., Navi Mumbai, India) containing VOCs at a mixing ratio of circa 1 ppmv (stated accuracy of ± 5%), 2) Gas standard (Apel-Riemer Environmental, Inc., Colorado, USA) containing VOCs at circa 500 ppb (stated accuracy better than 5%) using which the instrument was calibrated. Instrumental sensitivities can change during a long run deployment owing to change in settings and mechanical wear and tear and therefore regular calibrations are important to assess the instrumental stability. Table S4 shows the average sensitivity factors (pAs/ppb) and standard deviation derived from thirteen calibrations performed regularly between Dec 2016 and Oct 2018 with no major changes (8-12% for most of the measured compounds), observed in the instrumental sensitivities. A reasonable agreement (considering the maximum instrumental uncertainty error of <15%) was found for the average calibration factors between Dec 2016 - Oct 2018 derived from the two different gas standards for the common compounds such as isoprene ($53.2 \pm 4.9$ and $55.6 \pm 5.9$ pAs ppb$^{-1}$), benzene ($67.8 \pm 5.6$ and $69.2 \pm 5.5$ pAs ppb$^{-1}$) and toluene ($74.6 \pm 6.6$ and $81.3 \pm 7.7$ pAs ppb$^{-1}$). Table 2 lists the compound specific precision errors, limit of detection (LOD) and total

uncertainties. The precision of the instrument was evaluated under identical conditions using the

relative standard deviation of five individual measurements of 1 ppb and 5 ppb of standard gas

mixture and was in the range of 1-6% for 1ppb and 0.1-0.5% for 5ppb for the reported compounds.

The limit of detection of the instrument was evaluated according to Eq. (1) at 5% probability, using

the standard deviation of 8 zero/blank samples measurements under identical conditions (Penkett,

2007; ACTRiS 2014).

$$LOD = 2t\sigma \hspace{4cm} (1)$$

Here, $\sigma$ is the standard deviation of 8 blank measurements (manual integration of peaks in the

blank sample and if peaks were missing then integration of the baseline corresponding to the same

retention time and average peak width) and $t$ is the student's t-value for the 5% probability and 7

degrees of freedom. The instrumental LOD was in the range of 2-104 ppt. The total uncertainties

were calculated using the root mean square propagation of individual uncertainties like the 5%

accuracy error inherent in the VOC gas standard concentration, error in the linear fit of the

calibration curve, the error in the flow reproducibility of the two mass flow controllers, and the

precision error of the instrument. The overall uncertainties for all compounds were less than 15%.

**3 Results and Discussion**

**3.1 NMHC chemical fingerprinting of emission sources**

**3.1.1 Paddy stubble fires and garbage fires**

Figure 1a-d shows the normalised emission profiles of the whole air samples collected from paddy

stubble and garbage fires under flaming and smouldering conditions. The mixing ratios were

corrected for ambient background levels using samples collected just before the fires, normalised

to the NMHC with the maximum mass concentration in the respective source sample and averaged for the different fires.

The largest contributors to the ~~observed~~ mass concentrations in paddy fires under flaming conditions were ethene (16%), benzene (16%), propene (13%), acetylene (13%) and ethane (12%), while in smouldering conditions ethane (21%), isoprene (13%), propene (13%), propane (8%) and ethene (6%) were the highest ranked contributors. Acetylene was found to be negligible (<1%) in smouldering fires and therefore can be used as tracer for fires under flaming conditions. Amongst alkenes, the fraction of ethene and propene reduced in smouldering while that of isoprene increased by ~3 times relative to the flaming stage emissions. In the studies reported previously (Akagi et al., 2011; Andreae, 2019), ethene was reported to have higher emissions than propene from crop residue fires. Our study results reveal that ethene emissions were lower in the smouldering fires as compared to propene. While the previous studies have compiled results of mostly laboratory combustion of fuels in controlled environments and are more typical of flaming conditions, the smouldering stage of fire which are characterized by poor combustion efficiency and therefore different flame chemistry in the agricultural fields as encountered by us, may be a cause for this variance and emphasize why results from controlled burn experiments need to be complemented with field crop residue fire results.. In the garbage fire emissions, under both flaming and smouldering conditions, benzene (24 and 26% respectively), propene (15 and 11% respectively) and ethene (14% and 7% respectively) were the most dominant NMHCs. The differences in burning efficiency of the fires were highlighted again by the lower fraction of acetylene (~1%) in smouldering conditions, while in flaming conditions it was 11%. Ethane, propane and $n$-butane increased by ~2 times under smouldering conditions. The garbage burnt in this study mostly comprised of wet vegetable and food waste from households (Sharma et al., 2019) and therefore

had lower styrene (<1%), compared to garbage samples containing plastic and packaging material which can also be a source of styrene emissions due to the presence of plastic (polystyrene) waste (Lemieux et al., 2004; Tang et al., 2000). Isoprene was found to be very low (<1%) in the garbage fires as compared to the paddy stubble fires and therefore could be potentially employed for distinguishing paddy stubble and garbage burning activities in the absence of isoprene emissions at night from biogenic sources. Furthermore, propane has been widely used as an emission tracer for fugitive LPG emissions (Blake and Rowland, 1995; Barletta et al., 2002; Apel et al., 2010), but in a complex emission environment influenced by intensive paddy stubble fires, use of propane as a fugitive LPG emission tracer may not be ideal as it is one of the major species (8% of the total NMHC emissions) emitted from the paddy stubble burning.

### 3.1.2 Fuel evaporative emissions

Figure 1e-~~g~~h shows the normalised source profiles of the whole air samples collected from the headspace of liquefied petroleum gas (LPG), petrol and diesel.

Propane, *n*-butane, *i*-butane and butenes were the major constituents of LPG evaporative emissions. Remarkably, the composition was different in both the types of LPG evaporative emissions. The domestic LPG evaporative emissions were a mixture of propane and butanes, with propane (40%) by weight as the most dominant emission, followed by *n*-butane (19%) and *i*-butane (16%). However, the commercial LPG evaporative emissions were mostly "butane rich" with lower propane (7%) and higher butenes (31% in total from all isomers). *n*-butane (37%) and *i*-butane (18%) comprised of nearly half of the total evaporative emissions from commercial LPG cylinders. The most abundant species in petrol evaporative emissions were *i*-pentane (49%), *n*-pentane (12%), 2,2-dimethylbutane (6%), 2-methylpentane (5%), *n*-butane (5%) and toluene (2%). The total aromatic content in the petrol vapours was low (4%), which is consistent with previous

studies (Harley et al., 2000; Na et al., 2004). Diesel evaporative emissions were quite different from petrol and had high fraction of heavier C5-C8 alkanes (55%) and aromatics (36%) while unsaturated C2-C6 compounds comprised only about 4% of the total emissions. The alkane content in our diesel evaporative emissions was $60.6 \pm 1.8$ % and was comparable to the Guangzhou diesel ($53.8 \pm 10.0$ %), Zhuhai diesel ($57.4 \pm 5.3$ %) and Macau diesel ($64.3 \pm 1.6$ %) (Tsai et al., 2006) along with the characteristic of higher fraction of heavier alkanes and C8-C10 aromatics. *n*-octane (8%), *n*-heptane (7%), *o*-xylene (6%), 1,2,4-trimethylbenzene (5%), *i*-pentane (5%), methylcyclopentane (5%) and toluene (5%) were the major species identified in diesel vapours. C8 and C9 aromatic compounds were roughly 14% each of total emissions and constituted the major fraction of aromatic content in the diesel vapours.

**3.1.3 Vehicular exhaust and traffic emissions**

Figure 2 shows the normalised source profiles of the whole air samples collected from the tail pipe exhaust of idling vehicles with different fuel types and from busy traffic junctions. Among NMHCs, compressed natural gas (CNG) vehicular emissions (Figure 2a) had 70% ethane by mass concentration which is not surprising considering it is mostly composed of methane and ethane (Goyal and Sidhartha, 2003). The other major NMHC emissions from CNG exhaust were propane (11%) and ethene (10%). Overall alkanes (87%) and alkenes (12%) accounted for almost all NMHC emissions from the CNG vehicles.

Figure 2b shows that LPG vehicular emissions were mainly comprised of low molecular weight alkanes, i.e, C2-C4 NMHCs. *n*-butane (23%), *i*-butane (15%), propane (~~14~~ 13%), propene (12%), *trans*-2-butene (11%), 1-butene (8%) and *cis*-2-butene (6%) were the major components by mass concentration in these emissions. Alkanes accounted for 56% and alkenes 44% of the total emissions, whereas aromatics were negligible. LPG fuel is known to completely combust at higher

driving speeds and therefore the presence of propane and butanes in the exhaust was indicative of incomplete combustion at idling stage (Guo et al., 2011). The major compounds in LPG-fuelled vehicle emissions found in this study were similar to studies in Taiwan (Chang et al., 2001) and Guangzhou (Lai et al., 2009). One major difference was that *n*-butane was the most abundant

emission in the LPG vehicular exhaust in this study as compared to propane which is reported in aforementioned studies. In Hong Kong, the LPG fuel composition shows a relative ranking of *n*-butane> propane> *i*-butane (Tsai et al., 2006). However, in our evaporative emission samples we observed propane>*n*-butane>*i*-butane for domestic LPG cylinders and *n*-butane>*i*-butane>propane for commercial LPG cylinders. The differences in our observations from the studies from

Guangzhou (Lai et al., 2009), Taiwan (Chang et al., 2001) and Hong Kong (Guo et al., 2011), regarding the higher fraction of butanes as compared to propane in idling vehicular exhaust therefore could be because of different engine technology/efficiency and combustion conditions in addition to the fuel composition.

Figure 2c-d shows the averaged vehicular emissions for two wheelers and four wheelers fuelled

by petrol. Aromatics (44%) and alkanes (42%) were the major constituents of emissions from petrol vehicles with toluene (15%), *i*-pentane (11%), *m/p*-xylene (10%), benzene (4%), 2,2-dimethylbutane (4%) and acetylene (4%) as the most abundant NMHC species. These results are also similar to the studies conducted in Taiwan (Chang et al., 2001) and Pearl River Delta (Liu et al., 2008). The two wheeler motorbikes and scooters have different motor engines as compared to

the four wheeler vehicles and are known to combust the fuel inefficiently resulting in high VOC emissions (Costagliola et al., 2014; Dröge et al., 2011; Liu et al., 2008; Tsai et al., 2014). Furthermore, they comprise of nearly 73% of the registered Indian vehicular fleet (MoRTH, 2018) and dominate the emissions from the road transport sector. Therefore, we present the normalised

profiles of two wheelers and four wheelers separately to understand the emission profiles and assess their impact on regional air quality. The emissions from the tail pipe of two-wheelers majorly comprised of toluene (16%), *i*-pentane (11%), *m/p*-xylene (10%), acetylene (6%), ethylbenzene (5%), benzene (4%) and 2,2-dimethylbutane (4%). These NMHCs were also present in the emissions from four-wheeler vehicles which comprised of toluene (13%), *i*-pentane (10%), m/p-xylene (10%), benzene (7%), 2,2-dimethylbutane (5%) and ethane (5%). Higher fraction of C2-C4 alkanes were measured in four wheelers and was primarily dominated by ethane. High content of BTEX (~~35~~ 34%) in petrol exhaust emissions is also noteworthy considering their potential impact on air quality and human health.

Figure 2e-g shows the tailpipe emissions from light duty three wheelers, light duty four wheelers and heavy duty vehicles fuelled by diesel. The diesel exhaust emission profiles were much simpler than the petrol exhaust emissions. Alkenes and acetylene were the major constituents of the diesel vehicular exhaust contributing 58% to the total NMHC emissions. Furthermore, the BTEX (16%) and C6-C8 (8%) emissions were also lower than the petrol exhaust emissions. Diesel engines are known for their better combustion efficiency (Reiter and Kockelman, 2016) due to which most of the higher hydrocarbons get combusted and yield the characteristic source profile of diesel exhaust containing ethene (26%), propene (14%) and acetylene (11%) by weight percent (Liu et al., 2008; Schauer et al., 1999). There were no major differences in the profiles of different types of diesel vehicles and ethene, propene, acetylene, benzene, 1,2,3-trimethylbenzene and 1-butene were the most dominant NMHCs. However, the fraction of C9-C10 aromatics was higher in heavy duty vehicles (19%) and three wheelers (11%) as compared to four wheelers (6%). Since *i*-pentane was found to be negligible (~~0.2~~ <0.5%) in diesel exhaust, it was identified as an ideal tracer for petrol vehicular emissions as has also been reported previously (Tsai et al., 2006; Guo et al., 2011).

In comparison to petrol, the diesel exhaust had lower fraction of heavier C6-C8 alkanes (8%) which were likely combusted. Figure 2h shows the averaged source profile of whole air sample collected from three busy traffic junctions which therefore represent the ambient traffic emissions mixture. Although the Indian vehicular fleet comprises of vehicles running on petrol, diesel, LPG and CNG, more than 70% of on-road vehicles are petrol fuelled (Guttikunda and Mohan, 2014; Goel and Guttikunda, 2015; Prakash and Habib, 2018). Therefore, the petrol vehicular exhaust emissions were expected to dominate the ambient traffic mixing ratios. As the samples were collected during rush hour (afternoon and evening hours) within some of the busiest traffic thoroughfares in two cities (Chandigarh and Mohali) as mentioned in Table 1, the samples were influenced by a sufficiently diverse fleet mixture similar to most Indian cities. The sampling duration in each case was ~15min, therefore they are not biased by few individual vehicles and can be considered to be representative of the ambient city traffic emissions. These samples are not representative of highway emissions which on the other hand tend to be dominated by light duty diesel vehicles and heavy duty diesel vehicles. While more samples collected in other seasons in addition to spring, would have been better as combustion as environmental conditions can affect variability of emissions, changes in terms of the major compound mixture emitted are unlikely. In addition, since the traffic samples were collected from busy traffic junctions, these were more likely to be influenced by the emissions in the vehicular idling condition as discussed earlier. Alkanes (51%) and aromatics (34%) formed a major fraction of the traffic emissions. Major NMHC species measured from the traffic were $i$-pentane (15%), toluene (11%), $n$-pentane (5%), $m/p$-xylene (5%), 2,2-dimethylbutane (5%) and acetylene (4%).

Based on the emission characteristics discussed earlier for each fuel type, petrol vehicles and LPG vehicles were identified as the most likely sources of $i$-pentane and propane respectively in the

traffic plume. Even though the Indian vehicular fleet is dominated by petrol fuelled vehicles, the consumption of diesel in road transport sector is approximately twice as much as petrol (Sadavarte and Venkataraman, 2014; Prakash and Habib, 2018). This is because the maximum diesel consumption (40%) is by heavy duty vehicles (HDVs) which run over large distances across inter-city highways and have lower mileage than other vehicle classes.

In the past three decades, India has undergone rapid economic and industrial growth, which in turn has resulted in increased consumption of diesel to sustain increased freight transport across the country (Nielsen, 2013). As discussed previously, the diesel vehicular exhaust and evaporative emissions were dominated by heavier C6-C8 alkanes, alkenes and aromatics which are key precursors in OH reactivity and ozone formation. Furthermore, secondary organic aerosol (SOA) formed from the diesel vehicular exhaust are estimated to be 2-7 times more than petrol vehicular exhaust in urban areas where diesel generally accounts for 10-30% of total on-road fuel consumption (Gentner et al., 2012).

It was estimated that in 2009 the transport sector contributed 694 Gg of PM emissions in India, >70% of which came from vehicles fuelled by diesel (Sahu et al., 2014). Since LPG and CNG vehicular emissions comprise of mostly C2-C4 alkanes and alkenes, they have lower SOA formation potentials than petrol and diesel (Derwent et al., 2010), and therefore have emerged as cleaner fuel alternatives. However, the emission of large suite of reactive unsaturated NMHCs due to improper combustion of these fuels results in high OH reactivity and OFP which can severely impact local air chemistry and quality. Therefore, in order to mitigate the emissions the use of improved technologies (for better combustion and emission reduction like catalytic convertors), cleaner fuels (Bharat Stage V (BSV) and Bharat Stage VI (BSVI)) (GoI, 2016) and reduced idling times of the vehicles should be encouraged. Also, depending on which type of pollution is more

acute (PM or gaseous), promoting the appropriate less polluting fuel type for more usage could help reduce the overall ambient pollution.

## 3.2 Assessment of OH reactivity, ozone formation potential (OFP) and BTEX loading from different emission sources

Figure 3a-d show the comparison of percentage contribution of different chemical classes of NMHCs to the total mass concentrations, OH reactivity ($s^{-1}$), normalised reactivity ($gO_3/gNMHC$) and total BTEX loading (%) from various emission sources. The hydroxyl radical reactivity reflects the total pollutant loading of the air mass (Sinha et al., 2012) and was calculated using Eq. (2):

$$\text{Total NMHC OH reactivity} = \Sigma \ k_{OH+NMHCi} \ [\text{NMHC}_i] \qquad (2)$$

where, $k_{OH+NMHCi}$ is the first order rate coefficient for the reaction of NMHC$_i$ with OH radicals (Atkinson et al., 1982; Atkinson et al., 2006) and [NMHC$_i$] is the measured concentration of the NMHC.

The ozone formation potential (OFP) is used as a metric to measure the contribution of NMHCs

to the total O$_3$ formation potential in urban environments (Carter, 1994). Normalised reactivity $R$ (g O$_3$/g NMHCs emitted) is generally used to indicate OFP for NMHCs from emission sources using their source profiles and MIR values using Eq. (3) (Harley et al., 2000; Zhang et al., 2013)

$$R = \ \Sigma_i \ \omega_i \ \times (MIR_i) \qquad (3)$$

where, $\omega_i$ are the weight percentage of NMHC$_i$, present in the emission source and $MIR_i$ are the

maximum incremental reactivity coefficients (MIR) (Carter, 1994; Carter, 2009).

In order to ascertain any statistical difference between the average OFPs of the emission sources, we carried out Tukey's pairwise honestly significant difference test (which accounts for sample size) and the summary of the test results is provided in Table S5. Based on the statistical test, it could be concluded with more than 95% confidence that CNG vehicular emissions and the fuel evaporative emissions had different OFPs compared to other emission sources. The averaged OFP for the emission sources was: diesel vehicle exhaust ($6.5 \pm 0.6$ gO3/gNMHC), smouldering paddy stubble fire ($5.9 \pm 0.2$ gO3/gNMHC), LPG vehicle exhaust ($5.7 \pm 1.1$ gO3/gNMHC), flaming paddy stubble fire ($5.2 \pm 0.9$ gO3/gNMHC), flaming garbage fire ($4.9 \pm 1.1$ gO3/gNMHC), smouldering garbage fire ($4.4 \pm 1.3$ gO3/gNMHC), LPG evaporative emissions ($4.5 \pm 1.6$ gO3/gNMHC), petrol vehicle exhaust ($3.9 \pm 0.7$ gO3/gNMHC), diesel evaporative emissions ($3.6 \pm 0.9$ gO3/gNMHC), petrol evaporative emissions ($2.0 \pm 0.4$ gO3/gNMHC), CNG vehicle exhaust ($1.5 \pm 0.8$ gO3/gNMHC). Although, alkenes were not the largest emissions by mass, they were still the largest contributor to the OH reactivity (67-93%) and OFP (70-83%) in the fire and LPG evaporative emissions. In the paddy stubble and garbage fire emissions, alkenes and aromatics contributed the largest to the total OH reactivity (~90% and 6-9% respectively) and OFP (70-82% and 16-27% respectively). Alkanes have comparatively poor reactivity towards OH radical and therefore despite contributing 15-37% to the total NMHC mass concentration, their contribution to the OH reactivity was very low (<3%).

In paddy stubble fires under flaming conditions propene (33%), and under smouldering conditions isoprene (46%) were the largest contributors to the total OH reactivity (details in Figure S5 and S6). These two NMHCs were also the largest contributor (~40-50% in total) to the OFP from paddy stubble fires. In garbage fires under both flaming and smouldering conditions, propene was

the largest contributor to the OH reactivity (46% and 42% respectively) and OFP (37% and 30% respectively).

LPG evaporative and vehicular exhaust emissions comprised of 68-81% and 56% alkanes respectively, however, >90% of total OH reactivity was contributed by the alkenes. Butenes were the largest contributors to total OH reactivity from domestic LPG evaporative, commercial LPG evaporative and LPG vehicular exhaust emissions (90%, 79% and 72% respectively) and OFP (71%, 83% and 59% respectively). 81% NMHC emissions from CNG vehicular exhaust were C2-C3 alkanes, but the maximum contribution to the total OH reactivity and OFP was from ethene (47% and 62% respectively).

In diesel evaporative emissions there was approximately equal contribution to the total OH reactivity from alkanes (36%) and aromatics (44%). This is because of the presence of larger fraction of heavier C5-C8 branched alkanes which are generally more reactive towards OH radical as compared to the light C2-C4 alkanes. While the rate coefficient values of C2-C4 alkanes vary between (0.25-2.12) x $10^{-12}$ cm$^3$ molecule$^{-1}$ s$^{-1}$ at 298K, the rate coefficient values of C5-C8 alkanes are between (3.6-8.9) x $10^{-12}$ cm$^3$ molecule$^{-1}$ s$^{-1}$ at 298K. *trans*-2-butene (10%) and 1,2,4-trimethylbenzne (~~13~~ 9%) were the largest contributors to the total OH reactivity, ~~and~~ while 1,2,4-trimethylbenzene (13%) and *o*-xylene (12%) dominated the OFP from diesel evaporative emissions. OH reactivity from diesel vehicular exhaust emissions however were dominated by alkenes ( >75%), and propene ( 32-39%) and ethene (23-31%) were the largest contributors in all the diesel vehicle categories. Both of these NMHCs also contributed >50% to the total OFP calculated from diesel vehicular exhaust. In petrol evaporative emissions, largest contribution to the OH reactivity was *i*-pentane (~~17~~ ~30%), pentene (22%) and butene (20%) isomers. However, in petrol vehicular exhaust both aromatics and alkenes became the dominant contributors (~40-

50% each) to OH reactivity and the major contributors were propene (14-23%), m/p-xylene (11-13%), styrene (8-11%) ethene (6-13%) and toluene (7%). For OFP from petrol vehicular exhaust, the largest contributing NMHCs were: *m/p*-xylene (24-26%) > toluene (14-16%) > propene (5-9%) > ethene (6-9%).

The total OH reactivity from traffic emissions were dominated by alkenes (48%) and aromatics (35%). The NMHCs contributing the largest fractions to the total OH reactivity were styrene (9%) > trans-2-butene (9%) > isoprene (7%) > 1-hexene (6%) ~ m/p-xylene (6%) ~ propene (6%) and to OFP were *m/p*-xylene (14%) > toluene (12%) > 1,2,4-trimethylbenzene (7%) ~ ethene (7%) > *i*-pentane (6%). High contributions to OH reactivity from styrene, isoprene and 1-hexene are
noteworthy. Even though these compounds were not the most abundant in the traffic samples by mass concentration, however they are very reactive with hydroxyl radicals in ambient air (isoprene: $k_{OH}$ = 10.0 x $10^{-11}$ $cm^3$ $molecule^{-1}$ $s^{-1}$; styrene: $k_{OH}$ = 5.8 x $10^{-11}$ $cm^3$ $molecule^{-1}$ $s^{-1}$; 1-hexene: $k_{OH}$ = 3.7 x $10^{-11}$ $cm^3$ $molecule^{-1}$ $s^{-1}$ at 298K) (Atkinson, 1989; Atkinson et al., 1997). Isoprene, styrene and 1-hexene have been reported previously in various traffic and tunnel
experiments across the world (Mugica et al., 1998; Borbon et al., 2001; Barletta et al., 2002; Ho et al., 2009; Zhang et al., 2018). Our traffic samples have comparable mixing ratios of isoprene observed from roadside ambient air measurements in Karachi (1.2 ± 0.9 ppb) (Barletta et al., 2002), 43 Chinese cities (0.86 ± 0.83 ppb) (Barletta et al., 2005) and Longchuan tunnel, Hefei (0.47 ± 0.20 ppb) (Deng et al., 2018), but higher than Chapultepec Avenue tunnel, Mexico city (0.17 ±
0.02 ppb) (Mugica et al., 1998), and the Fu Gui Mountain Tunnel (0.14 ± 0.36 ppb) (Zhang et al., 2018). In the Hong Kong tunnel experiment (Ho et al., 2009) and Taipei tunnel experiment (Hwa et al., 2002), isoprene was however undetectable. This variability in isoprene emissions from

traffic/vehicular exhaust have been previously attributed to variable fuel types, vehicular engines and maintenance, driving patterns and sampling strategies.

The mixing ratios of styrene and 1-hexene measured in our traffic samples were higher than the Fu Gui Mountain Tunnel (styrene: $0.08 \pm 0.00$ ppb, 1-hexene: $0.07 \pm 0.00$ ppb), but comparable to 1-hexene reported from Taiwan tunnels (Cross-Harbor Tunnel ($0.99 \pm 0.20$ ppb), Chung-Bor Tunnel ($3.29 \pm 2.36$ ppb) and Chung-Cheng Tunnel ($2.49 \pm 1.27$ ppb) (Chen et al., 2003). Though high mixing ratios of styrene are remarkable, it has been previously reported that styrene is one of the major VOCs emitted from the diesel LDVs especially in cold transient mode (Tsai et al., 2012). Amongst our traffic samples, maximum mixing ratios of isoprene ($1.11 \pm 0.06$ ppb), styrene ($2.31 \pm 0.16$ ppb) and 1-hexene ($2.38 \pm 0.14$ ppb) were observed in Transport chowk ($30.717^{o}$N, $76.812^{o}$E) which is one of the busiest traffic junction in Chandigarh city during rush hours and witnesses a large vehicular fleet of diesel run commercial LDVs.

In order to assess the health risks associated with these sources, we compared the fraction of BTEX compounds in each of the emission sources. Benzene is classified as a human carcinogen (IARC, 2012) the potential health risk assessments of which have already been elucidated before, in NW-IGP during the periods influenced by intense paddy stubble fires (Chandra and Sinha, 2016). Other benzenoids like toluene and xylenes have also been associated with adverse effects on human health (ATSDR, 2000; ATSDR, 2007) and are classified as group "D" carcinogens by the US EPA. Using the BTEX fraction which is a well know metric (Słomińska et al. 2014), is useful for comparing the mass fractional BTEX content of the emission sources. The statistical differences in the average BTEX fraction between the different emission sources were ascertained by Tukey's pairwise honestly significant difference test and the summary for the same is provided in Table S6. Based on the statistical test, it could be concluded with more than 95% confidence that diesel

and petrol evaporative emissions, diesel vehicles and smouldering paddy fires had different average BTEX fraction as compared to other emission sources. Out of 28 possible pairwise comparisons, 14 pairs show statistically significant differences with ≥2 σ confidence, 3 are only significant at 1 σ level and the rest were not significant. The fraction of BTEX in the different

emission sources was: petrol vehicle exhaust (27 ± 5%), smouldering garbage fire (26 ± 1%), flaming garbage fire (24 ± 8%), flaming paddy stubble fire (22 ± 5%), diesel vehicle exhaust (19 ± 2%), diesel evaporative emissions (17 ± 2%), smouldering paddy stubble fire (13± 1%) and petrol evaporative emissions (3 ± 1%). LPG and CNG emission sources had <1% of benzene and therefore were identified as least harmful sources, while petrol vehicular exhaust, garbage fires

and paddy stubble fires were the most toxic emissions which could severely impact human health considering their BTEX emission potential.

**3.3 Molar emission ratios of NMHCs in different emission sources**

Inter NMHC molar ratios (ppb/ppb) are very useful tools that can not only be used to distinguish between different emission sources but also constrain the identity of the sources affecting ambient

mixing ratios in a complex environment (Barletta et al., 2005; Barletta et al., 2017). This is because, for the NMHC species with similar chemical lifetimes, the molar ratios remain preserved during chemical oxidation and ambient dilution (Parrish et al., 1998; Jobson et al., 1999). Further NMHC molar ratios that remain similar across sources can also be employed to assess the photochemical age of air masses.

Table 3 lists commonly used inter NMHC molar emission ratios for the emission sources studied in this work. Toluene/benzene (T/B) ratio is a widely used ratio in identifying vehicular emission sources (Barletta et al., 2002; Barletta et al., 2005). T/B measured for traffic in this study was 3.54 ± 0.21 which is comparable to previous studies from busy traffic junctions in Karachi (2.2 ± 2.9)

(Barletta et al., 2002), Hong Kong (3.0 ± 0.4) (Huang et al. 2015), Okhla, New Delhi (2.3 ± 1.7) (Hoque et al. 2008), Antwerp, Belgium (3.5 ± 0.2) (Buczynska et al. 2009) and Nanjing, China (2.6 ± 0.9) (Wang et al. 2008). For the idling vehicular exhausts of different fuel types, this ratio varied between 0.38-10.9 and was 3.68 ± 0.58 for petrol vehicles which is consistent with the previous works of Guo et al., 2011 (2.0-3.8). For the diesel vehicles, the T/B ratio in our study was 0.37 ± 0.20 which is similar to the average T/B ratio (0.37) from diesel vehicles in Australia (Anyon et al., 2003), Germany (0.56) (Siegl et al., 1999) and Tokyo (0.3) (Yamamoto et al., 2012). Furthermore, T/B ratios can also be useful in distinguishing the paddy stubble fire emissions in flaming (0.38 ± 0.11) and smouldering stages (1.40 ± 0.10).

$i$-butane/n-butane ratio (B/B) is another example of a widely used NMHC ratio to distinguish between different fossil fuel related emission sources. However, in our study we found that this ratio is not useful in a complex emission environment influenced by varied emission sources. This is because the ratio exhibits similar values (0.20-0.30) for paddy stubble fires, garbage fires, petrol evaporative, diesel evaporative and petrol vehicle exhaust and diesel vehicular exhaust emissions. Therefore, caution should be taken while using this ratio in complex emission environments where biomass burning, fossil fuel combustion and biogenic emission sources simultaneously occur in significant scale and strength to contribute to the chemical composition of ambient air. $i$-pentane/$n$-pentane can instead be used as a more reliable ratio for distinguishing biomass burning emissions (0.06-1.46) from the petrol dominated traffic/fossil fuel emissions (2.83-4.13).

**4 Conclusion**

Comprehensive chemical speciation source profiles of 49 NMHCs (22 alkanes, 16 aromatics, 10 alkenes and 1 alkyne) were obtained for several major emission sources, namely paddy stubble burning, garbage burning, idling vehicular exhaust and evaporative fuel emissions. Many of these

compounds like the higher C6-C8 alkanes, C9-C10 aromatics and alkenes have been quantified for the first time for these emission sources in the South Asian region, which is important to ascertain region to region variability of such common urban and agricultural emission sources. The work highlights the importance of identifying the local emission source profiles as some

NMHC emissions were found to be very different to the studies reported from North America, Europe and East Asia. Some of the major findings which provide new insights are:

i.      Propane was found to be one of the abundant NMHC compounds in paddy stubble fire emissions. This is in contrast to the existing literature which considers it as a tracer for fugitive LPG emissions. In a complex emission environment influenced by several sources like paddy fires,

the use of propane as an LPG tracer therefore calls for caution.

ii.      Propene emissions in smouldering fires were found to be more than ethene by ~1.6 times which is in contrast to the existing crop residue burning inventories which have ethene as the more abundant compound.

iii.      Isoprene was identified as a reliable tracer to distinguish between the paddy fires and

garbage fires at night.

iv.      It was also found that there were compositional differences in the evaporative emissions from the two types of LPG (commercial and domestic) used widely in South Asia. While, propane was the most dominant NMHC in the domestic LPG vapours, the commercial LPG vapours were dominated by butanes.

v.      Toluene/benzene ratios were identified as a good tracer to distinguish the paddy stubble fire emissions in flaming ($0.38 \pm 0.11$) and smouldering stages ($1.40 \pm 0.10$), garbage burning emissions (0.26-0.59) and traffic emissions ($3.54 \pm 0.21$).

vi.    *i*-butane/*n*-butane ratio was found to be similar (0.20-0.30) for many sources and therefore caution must be taken while using it in complex emission environments. *i*-pentane/*n*-pentane ratio instead turned out to be a better tracer for distinguishing biomass burning emissions (0.06-1.46) from the petrol dominated traffic/fossil fuel emissions (2.83-4.13).

These source profiles can be used for accurate and reliable emission calculations, source apportionment studies and assessing the choice of fuels from point of view of air quality impacts both as primary emission sources and also their potential to form secondary air pollutants like ozone and particulate matter. Ambient traffic emissions were found to be dominated by the petrol exhaust emissions due to the typically higher fraction of petrol fuelled vehicles among the on road

intra-city vehicular fleet in India. The potential toxicity and health impacts of the emissions sources were assessed by using the BTEX fraction as a metric, and petrol exhaust, paddy stubble fires and garbage fires were ranked higher in toxicity than other emissions based on this metric. Based on our limited measurements of ambient benzene in the traffic throughfares, the mass concentration was $6.1 \pm 1.3$ µg m$^{-3}$, which is higher than the 5 µgm$^{-3}$ annual exposure limit set in the National

Ambient Air Quality Standards (NAAQS) of India (NAAQS, 2009). Future studies should quantify annual ambient exposure of such toxic compounds from the sources which have high BTEX content for assessing compliance with the annual ambient air quality standards as has been done for paddy residue smoke previously (Chandra and Sinha, 2016). The diesel and petrol vehicular exhaust emissions, paddy stubble fire and garbage fire emissions were identified as the

most polluting emission sources in terms of OH reactivity and ozone formation potentials. Although LPG and CNG vehicular exhaust emissions were cleaner, however they comprised of large fraction of alkenes due to improper combustion of fuels. Thus they can impact local air

quality and atmospheric chemistry and therefore use of improved VOC scrubbing technologies, cleaner fuels and reduced idling times of the vehicles should be promoted.

The results and insights obtained from this study will aid identification of factor profiles in source apportionment models such as positive matrix factorization yielding more accurate quantitative data for mitigation of ambient air pollution.

**Data availability.** Data is available from the corresponding author upon request

**Author contributions.** V.S. and A.K. conceived and designed the study. A.K. carried out the sample collection, field work and performed TD-GC-FID measurements with help of M.S. and H.H and advices of B.B concerning the analytical system. A.K. carried out preliminary analysis and wrote the first draft. V.S. revised the paper and carried out advanced analyses and interpretation of the data and supervised all experimental aspects of the work. V.G. participated in the discussion of the analytical system and commented on the paper.

**Competing interests.** The authors have no competing interests to declare.

**Acknowledgements.** We acknowledge the IISER Mohali Atmospheric Chemistry facility for data and the Ministry of Human Resource Development (MHRD), India for funding the facility. A.K., H.H. and M.S. acknowledge MHRD and IISER Mohali for PhD (SRF) and PhD (JRF) fellowships. We acknowledge EGU for waiver of the APC through the EGU 2019 OSPP award to A.K. We also thank Dr. Baerbel Sinha (Department of Earth and Environmental Sciences, Indian Institute of Science Education and Research Mohali) and the two anonymous reviewers for their helpful suggestions and insightful comments which helped to improve the discussion paper. We also acknowledge the help and support of the members of IISER Mohali Atmospheric Chemistry facility: Harshita Pawar, Pallavi, Abhishek Mishra, Abhishek Verma, Bharti Sohpaul and Tess George for technical assistance during field sampling. This work was supported by funding received under the National Mission on Strategic knowledge for Climate Change (NMSKCC) MRDP Program of the Department of Science and Technology, India vide grant (SPLICE) DST/CCP/MRDP/100/2017(G).

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

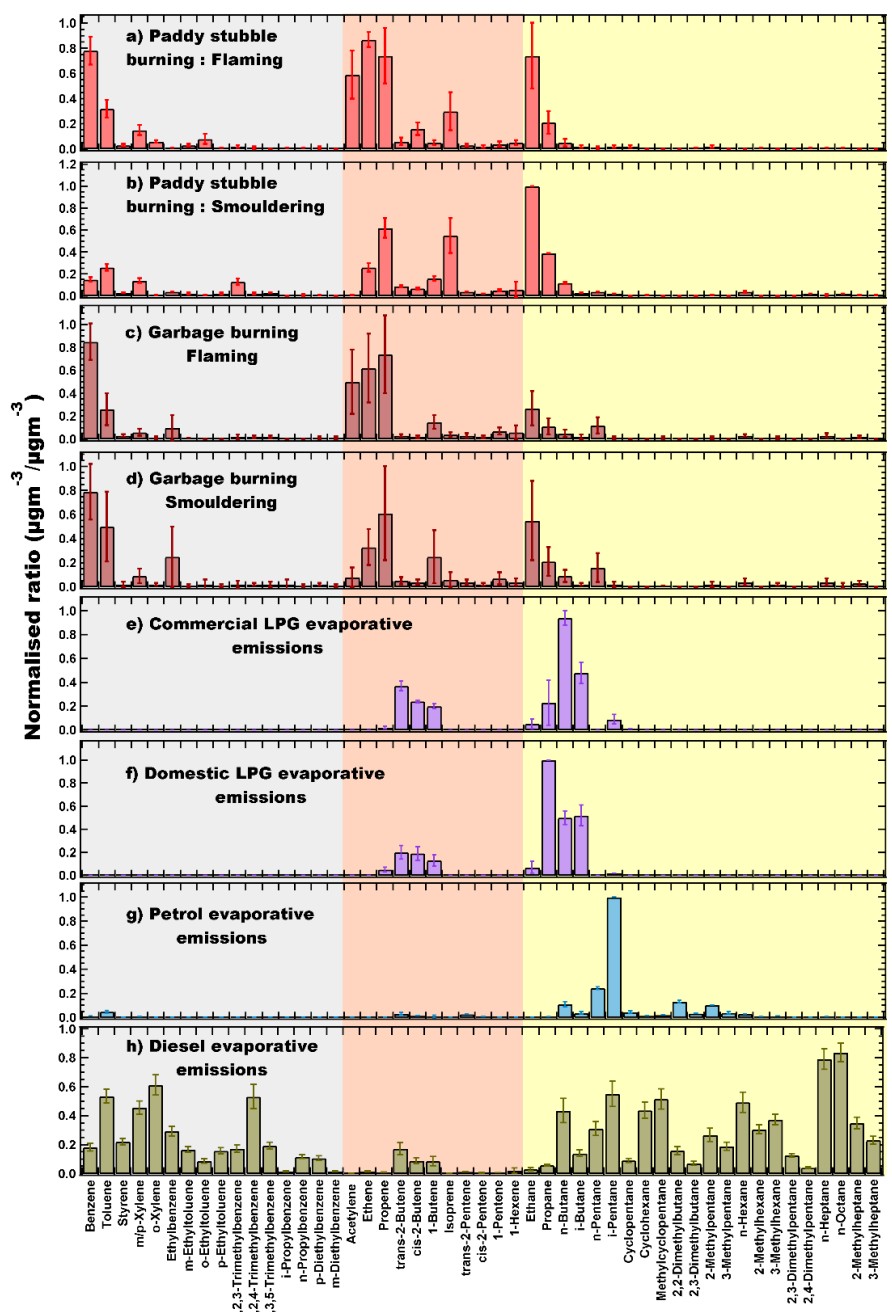

**Figure 1:** Normalised source profiles of **a)** Paddy stubble burning: Flaming; **b)** Paddy stubble burning: Smouldering; **c)** Garbage burning: Flaming; **d)** Garbage burning: Smouldering; **e)** Commercial LPG evaporative emissions; **f)** Domestic LPG evaporative emissions; **g)** Petrol evaporative emissions; **h)** Diesel evaporative emissions, derived from the TD-GC-FID measurements. Error bars represent the standard error of averaged normalised ratio. The grey colour highlights the aromatics, red colour highlights the alkenes and alkyne and the yellow colour highlights the alkanes.

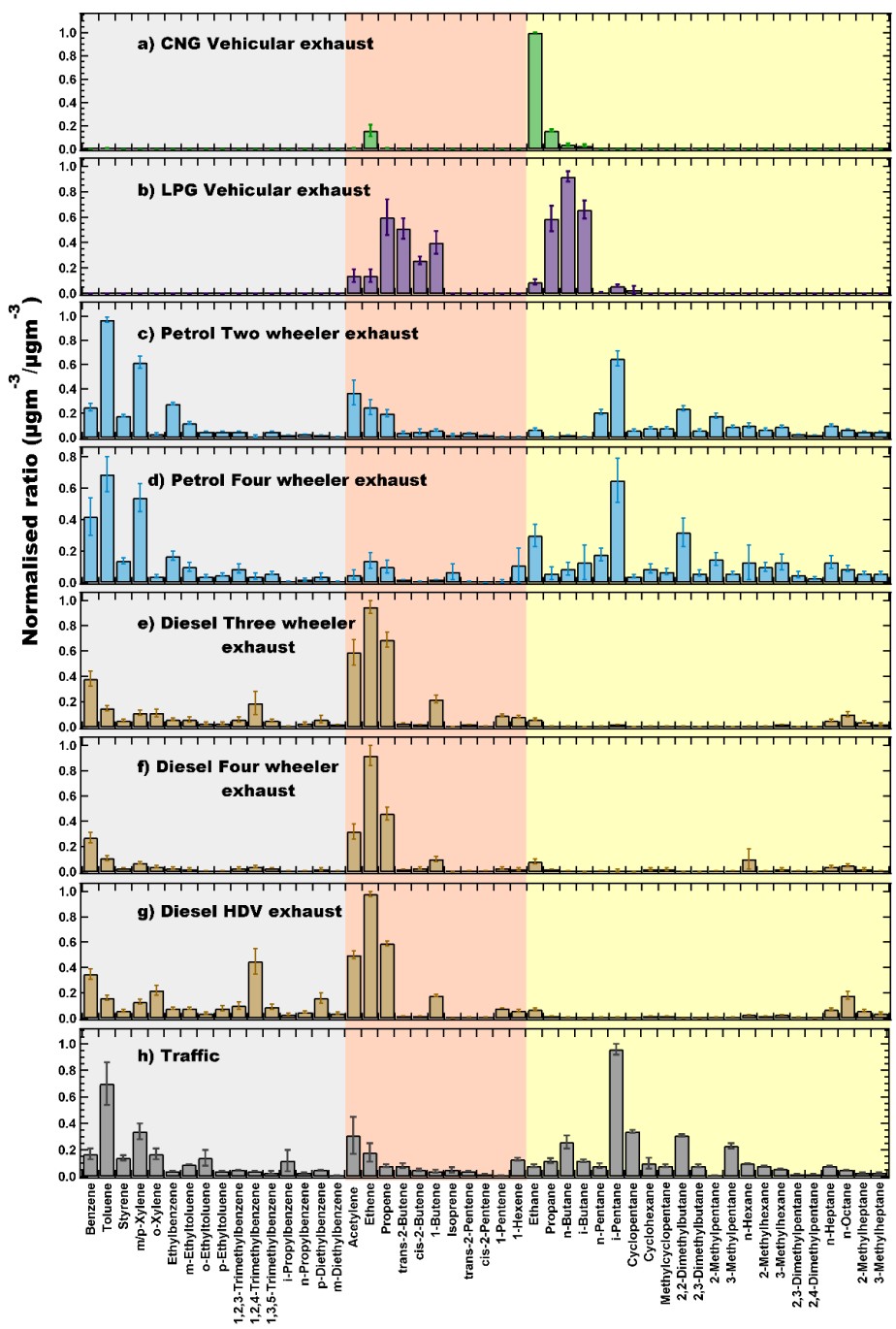

**Figure 2:** Normalised source profiles of **a)** CNG vehicular exhaust; **b)** LPG vehicular exhaust; **c)** Petrol two wheeler vehicular exhaust; **d)** Petrol four wheeler vehicular exhaust; **e)** Diesel three wheeler vehicular exhaust; **f)** Diesel four wheeler vehicular exhaust; **g)** Diesel heavy duty vehicle (HDV) exhaust; **h)** Traffic, derived from the TD-GC-FID measurements. Error bars represent the standard error of averaged normalised ratio. The grey colour highlights the aromatics, red colour highlights the alkenes and alkyne and the yellow colour highlights the alkanes.

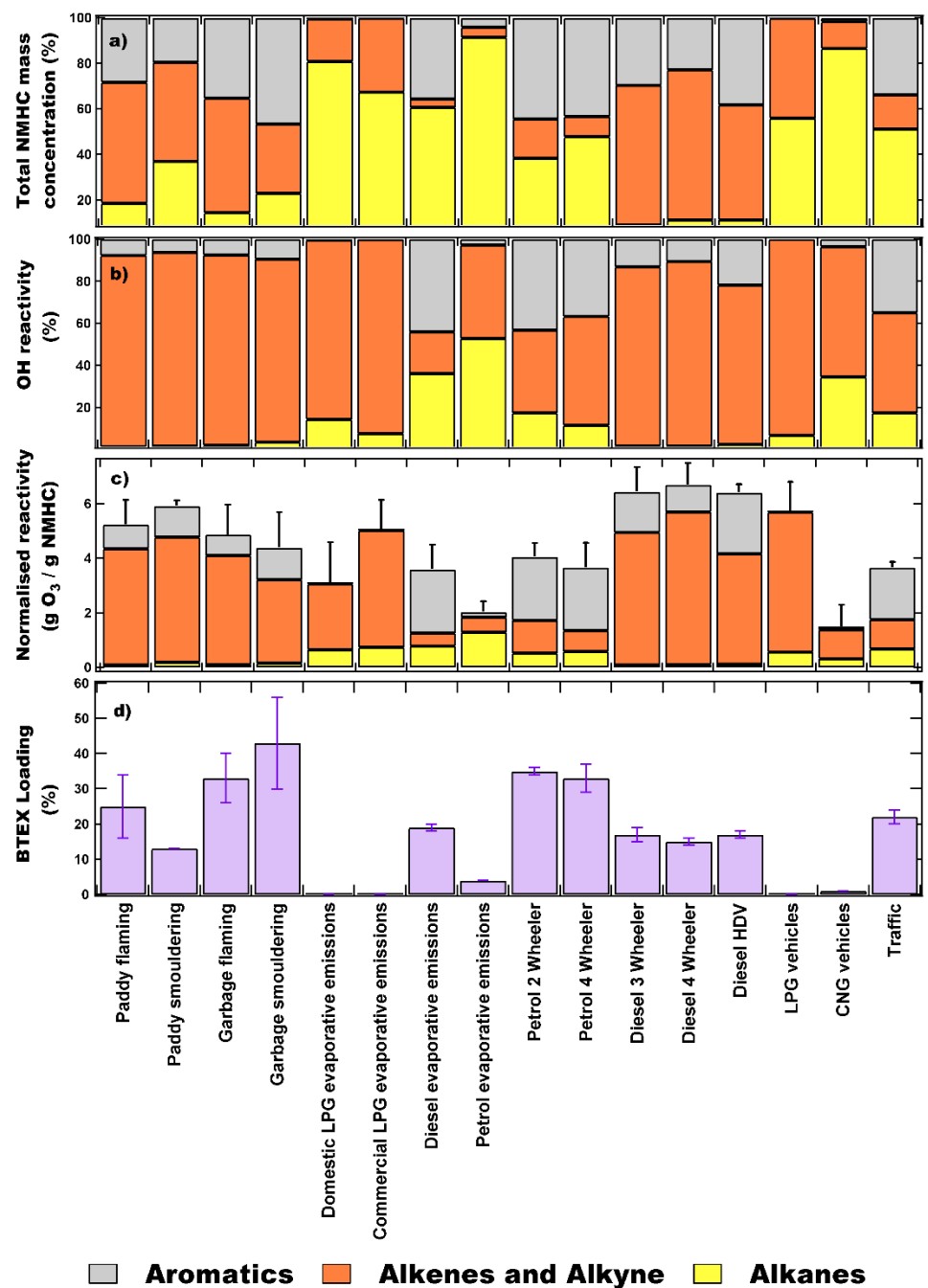

**Figure 3:** Comparison of contribution of chemical compositions in groups (aromatics, alkene and alkyne and alkanes) to **a)** NMHCs mass concentrations, **b)** OH reactivity (s$^{-1}$), and **c)** Normalised reactivity (gO$_3$/gNMHC) and **d)** BTEX loading (%) from various emission sources. Error bars represent the standard error of average OFP and BTEX fraction.

**Table 1:** Number of samples investigated per source for the measurements of NMHCs source profiles

| Sources | Description | # samples |
| --- | --- | --- |
| Paddy stubble burning (Flaming stage) | Agricultural field in Kurari, Mohali (30.605° N, 76.744° E) on 4 Nov 2017 | 3 |
| Paddy stubble burning (Smouldering stage) | Agricultural field in Kurari, Mohali (30.605° N, 76.744° E) on 4 Nov 2017 | 3 |
| Garbage burning (Flaming stage) | Waste sorting and disposing stations in Mohali and surrounding villages (30.642-30.699°N, 76.713-76.729°E) in February 2017 | 5 |
| Garbage burning (Smouldering stage) | Waste sorting and disposing stations in Mohali and surrounding villages (30.642-30.699°N, 76.713-76.729°E) in February 2017 | 5 |
| Traffic | Busy traffic junctions in Chandigarh and Mohali (30.691° N, 76.698° E; 30.678° N, 76.721° E and 30.717° N, 76.812° E) in on 3rd, 8th and 15th March 2017 b/w 11:00-17:00 local time | 3 |
| Petrol vehicular exhaust | Petrol LDV and 2 wheelers in idling stage in Chandigarh and Mohali (30.660-30.750°N,76.700-76.840°E) between Mar 2017-Oct 2018 | 23 |
| Diesel vehicular exhaust | Diesel LDV 4 wheelers and 3 wheelers and HDV in idling stage in Chandigarh and Mohali (30.660-30.750°N,76.700-76.840°E) between Mar 2017- Oct 2018 | 33 |
| LPG vehicular exhaust | LPG 3 wheelers in idling stage in Chandigarh and Mohali (30.660-30.750°N,76.700-76.840°E) between Mar 2017- Oct 2018 | 9 |
| CNG vehicular exhaust | CNG 3 wheelers and LDV 4 wheelers in idling stage in Chandigarh and Mohali (30.660-30.750°N,76.700-76.840°E) between Mar 2017- Oct 2018 | 7 |
| LPG evaporative emissions | LPG vapours collected directly from domestic (5) and commercial LPG cylinders (5) in Mohali, Chandigarh and Panchkula on 13-14 August 2020 | 10 |
| Petrol evaporative emissions | Petrol vapours collected directly from the headspace of fuel tank of the petrol vehicles between on 13-14 August 2020 in IISER Mohali campus (30.665° N, 76.730° E) | 10 |
| Diesel evaporative emissions | Diesel vapours collected directly from the headspace of fuel tank of the diesel vehicles between on 13-14 August 2020 in IISER Mohali campus (30.665° N, 76.730° E) | 10 |

**Table 2:** Compound specific Precision errors (%), Limit of Detection (LOD) (in ppt) and Total measurement uncertainties (%).

| Compounds | Precision at 1ppb (%) | Precision at 5ppb (%) | LOD (ppt) | Uncertainty (%) | Compounds | Precision at 1ppb (%) | Precision at 5ppb (%) | LOD (ppt) | Uncertainty (%) |
|---|---|---|---|---|---|---|---|---|---|
| Aromatics (n=16) | | | | | | | | | |
| Benzene | 1 | 0.2 | 21 | 5.9 | *p*-Ethyltoluene | 1 | 0.3 | 9 | 9.3 |
| Toluene | 2 | 0.3 | 87 | 6.2 | 1,2,3-Trimethylbenzene | 2 | 0.2 | 104 | 11.3 |
| Styrene | 2 | 0.4 | 19 | 7.0 | 1,2,4-Trimethylbenzene | 1 | 0.2 | 56 | 9.0 |
| *m/p*-Xylene | 1 | 0.1 | 45 | 7.1 | 1,3,5-Trimethylbenzene | 3 | 0.2 | 14 | 9.0 |
| *o*-Xylene | 2 | 0.2 | 24 | 5.8 | *i*-Propylbenzene | 2 | 0.3 | 7 | 6.6 |
| Ethylbenzene | 1 | 0.3 | 41 | 6.5 | *n*-Propylbenzene | 1 | 0.2 | 8 | 8.0 |
| *m*-Ethyltoluene | 1 | 0.4 | 9 | 8.8 | *m*-Diethylbenzene | 1 | 0.1 | 5 | 12.3 |
| *o*-Ethyltoluene | 2 | 0.3 | 9 | 8.9 | *p*-Diethylbenzene | 2 | 0.1 | 17 | 14.7 |
| Alkyne (n=1) | | | | | | | | | |
| Acetylene | 5 | 0.2 | 64 | 5.9 | | | | | |
| Alkenes (n=10) | | | | | | | | | |
| Ethene | 6 | 0.3 | 103 | 5.9 | Isoprene | 3 | 0.2 | 4 | 6.0 |
| Propene | 4 | 0.3 | 47 | 5.8 | 1-Pentene | 2 | 0.1 | 2 | 5.8 |
| 1-Butene | 3 | 0.2 | 3 | 5.8 | *trans*-2-Pentene | 1 | 0.2 | 4 | 5.8 |
| *trans*-2-Butene | 2 | 0.3 | 18 | 6.0 | *cis*-2-Pentene | 2 | 0.2 | 2 | 5.8 |
| *cis*-2-Butene | 1 | 0.2 | 8 | 5.8 | 1-Hexene | 2 | 0.3 | 4 | 5.8 |
| Alkanes (n=22) | | | | | | | | | |
| Ethane | 3 | 0.3 | 15 | 7.3 | *n*-Hexane | 2 | 0.5 | 3 | 5.8 |
| Propane | 5 | 0.2 | 20 | 5.8 | 2-Methylpentane | 1 | 0.2 | 2 | 5.8 |
| *n*-Butane | 2 | 0.1 | 3 | 5.8 | 3-Methylpentane | 1 | 0.2 | 3 | 5.8 |
| *i*-Butane | 4 | 0.2 | 6 | 5.8 | 2-Methylhexane | 2 | 0.2 | 15 | 5.8 |
| *i*-Pentane | 2 | 0.2 | 4 | 7.3 | 3-Methylhexane | 2 | 0.3 | 7 | 5.8 |
| *n*-Pentane | 1 | 0.1 | 4 | 5.8 | 2,3-Dimethylpentane | 1 | 0.1 | 1 | 5.8 |
| Cyclopentane | 1 | 0.2 | 3 | 6.3 | 2,4-Dimethylpentane | 2 | 0.2 | 11 | 5.8 |
| Cyclohexane | 1 | 0.2 | 2 | 5.8 | *n*-Heptane | 2 | 0.3 | 15 | 5.9 |
| Methylcyclopentane | 2 | 0.3 | 13 | 5.8 | *n*-Octane | 3 | 0.2 | 103 | 5.8 |
| 2,2-Dimethylbutane | 2 | 0.2 | 4 | 5.8 | 2-Methylheptane | 2 | 0.2 | 85 | 5.8 |
| 2,3-Dimethylbutane | 1 | 0.1 | 2 | 5.8 | 3-Methylheptane | 4 | 0.2 | 81 | 5.8 |

**Table 3:** Characteristic **i**nter-NMHC molar ratios (ppb/ppb) for the whole air samples collected from Paddy stubble fires, Garbage fires, evaporative fuel emissions (Petrol, Diesel and LPG), Traffic and vehicular exhaust from different fuel types (Petrol, Diesel, LPG and CNG).

| Emission ratio (ppb/ppb) | Paddy stubble burning (F) | Paddy stubble burning (S) | Garbage burning (F) | Garbage burning (S) | Evaporative emissions | | | Traffic | Vehicular exhaust emissions | | | |
|---|---|---|---|---|---|---|---|---|---|---|---|---|
| | | | | | Petrol | Diesel | LPG | | Petrol | Diesel | LPG | CNG |
| Toluene/Benzene | 0.38 (0.11) | 1.40 (0.10) | 0.26 (0.07) | 0.59 (0.16) | 3.13 (0.34) | 2.88 (0.38) | 3.41 (0.55) | 3.54 (0.21) | 3.68 (0.58) | 0.38 (0.02) | 0.59 (0.17) | 10.90 (2.98) |
| $i$-Butane/$n$-Butane | 0.41 (0.13) | 0.26 (0.00) | 0.24 (0.10) | 0.22 (0.04) | 0.34 (0.02) | 0.35 (0.02) | 0.79 (0.13) | 0.48 (0.03) | 0.50 (0.12) | 0.38 (0.02) | 0.73 (0.08) | 0.77 (0.11) |
| $i$-Pentane/$n$-Pentane | 1.46 (0.71) | 0.56 (0.02) | 0.06 (0.02) | 0.12 (0.04) | 4.13 (0.08) | 1.84 (0.13) | 12.13 (2.56) | 2.83 (0.17) | 3.27 (0.19) | 1.42 (0.10) | 14.99 (2.69) | 3.45 (0.32) |
| Propane/$n$-Butane | 8.05 (3.17) | 4.30 (0.04) | 2.81 (0.28) | 2.99 (0.28) | 0.04 (0.01) | 0.21 (0.02) | 1.61 (0.47) | 0.58 (0.05) | 0.64 (0.11) | 3.72 (0.35) | 0.89 (0.18) | 8.93 (3.01) |
| Propene/Ethene | 0.55 (0.14) | 1.52 (0.02) | 0.79 (0.15) | 1.06 (0.23) | 3.38 (3.38) | 0.27 (0.11) | 14.74 (11.78) | 0.38 (0.10) | 0.64 (0.06) | 0.40 (0.02) | 7.22 (3.64) | 0.06 (0.02) |
| *trans*-2-Butene/*cis*-2-Butene | 1.28 (0.03) | 1.33 (0.02) | 1.32 (0.03) | 1.42 (0.04) | 2.51 (0.39) | 1.89 (0.03) | 1.82 (0.09) | 1.80 (0.07) | 1.90 (0.60) | 1.35 (0.02) | 1.93 (0.17) | 1.71 (0.16) |
| *trans*-2-Pentene/*cis*-2-Pentene | 1.53 (0.05) | 1.83 (0.05) | 1.74 (0.02) | 1.54 (0.14) | 2.83 (0.28) | 2.91 (0.63) | 1.70 (0.43) | 2.04 (0.07) | 4.56 (2.54) | 1.65 (0.05) | 1.51 (0.08) | 0.99 (0.31) |
| Styrene/1,3,5-TMB | 1.77 (0.32) | 1.45 (0.04) | 3.29 (1.13) | 2.29 (1.10) | 7.42 (0.83) | 1.48 (0.19) | 1.68 (0.45) | 3.73 (0.76) | 4.19 (0.31) | 1.48 (0.27) | 2.67 (0.37) | 2.10 (0.18) |
| 1,2,3-TMB/1,2,4-TMB | 6.70 (3.35) | 8.36 (1.37) | 0.68 (0.35) | 1.39 (0.49) | 2.17 (1.46) | 0.33 (0.02) | 0.74 (0.53) | 1.78 (0.55) | 8.37 (0.89) | 1.21 (0.64) | 3.49 (1.62) | 3.29 (2.05) |

TMB: Trimethylbenzene, F: Flaming, S: Smouldering

