# Peer review of "Non methane hydrocarbon (NMHC) fingerprints of major urban and agricultural emission sources for use in source apportionment studies"

_Atmospheric Chemistry and Physics, 2019_

## Referee Comment (RC1) · Anonymous Referee #1 · 27 Feb 2020

This paper presents NMHC fingerprints of urban and agricultural emission sources in northern India based on whole air sampling. Similar to previous studies, i-pentane was found to be a tracer for petrol vehicular exhaust, propane for LPG emissions, acetylene for biomass burning, and alkenes for diesel exhaust. The authors report significant emissions of propane from paddy stubble fires, and suggest that isoprene can be used to distinguish paddy stubble burning from garbage burning.

This is an interesting research paper from an understudied part of the world with complex emission sources. However several issues need to be addressed before the paper

can be published, especially concerning uncertainty, data variability, and small sample size. The actual concentration data should be shown for both background and source samples, including average values with uncertainties. So far the data are shown as normalized source profiles, so the reader can't see the range of the data, how well the precision (which was determined for 5 ppb) applies to the measured concentrations, or how often the data were close to the detection limit. Many results in the paper need error bars. Another issue is the small sample size (as small as 1 sample for some sources) and how that limits the study and its application to other areas of India or South Asia. Also, since some of the source profile results are known from previous studies, the last sentence of the abstract perhaps overstates the novelty of this work. Additional comments are given below.

1. The title states "South Asia" but the paper is based on limited measurements near Mohali in northern India. What basis is there for extrapolating these results to all of South Asia? Emissions can vary from country to country (Page 5, Line 4) and even within countries. "Indo-Gangetic Plain" is probably a better choice for the title.

2. Page 5 Line 21: The sample size is quite small, with three flaming and three smoldering samples from paddy stubble fires, and five each from garbage burning. While any information is good from understudied sources and source regions, please discuss the limitations of the small sample size. Also show the reader the actual data (concentrations) by including a statistics table with uncertainties.

3. Page 7 Line 1: Does the text here mean that Panels e-g in Figure 1 and Figure S4 are based on just one sample each? This is a major source of uncertainty, since there is no way to quantify the variability in the evaporative source emissions or assign uncertainty. Please discuss this limitation.

4. Page 10 Line 11: Precision usually varies with concentration, but here the stated precisions are for 5 ppb. How does this change as the measurements go below 5 ppb or approach the detection limit? The precisions listed in Table 2 are very good (0.1-

0.4%), but for example 0.2% on an isoprene value of 200 ppt would be measuring to 0.4 ppt, which seems unrealistic. The overall uncertainty (Page 10, Line 23) will likewise be underestimated if the precision degrades at concentrations less than 5 ppb. Please show the reader the range of concentrations that was measured for each VOC, and discuss how well the precision applies across the range.

5. Page 10 Line 19: How often did the measurements go below detection? Please show a statistics table for the background measurements including their uncertainty, and also state the number of background samples that were collected for each source. On Page 11, Line 6 provide more detail about how the background data were used to correct the fire mixing ratios. How did uncertainty in the background measurements impact the normalized source profile concentrations?

6. Page 11 Line 5: From Table 1, benzene is the most abundant normalized compound in the garbage burning. What was its peak concentration? This is relevant for health considerations. Same question on Page 14 Line 5 for traffic.

7. Page 11 Line 9: The term "observed mass concentrations" is used here, but the text above describes an average excess concentration above background. Please clarify what is being discussed here and in the following text. By "acetylene was negligible in the smoldering fires", does this mean that the excess above background was <1%? What is the uncertainty in this result and how does it compare to previous work from smoldering fires?

8. Page 11 Line 11: How did the background isoprene concentration compare to the paddy fire concentration? Please show the data.

9. Page 13 Line 9: The evaporative LPG emissions were n-butane > propane > i-butane (Page 12 Line 10), but the LPG exhaust is n-butane > i-butane > propane. Why would the composition change from evaporative emissions to exhaust, to yield relatively less propane? How large are the error bars?

10. Page 14 Line 22: The three traffic samples were collected in March 2017 (Page 6 Line 9). This is very limited sampling at one time of year (and we don't know what time of day, or how long the sample duration was). While any information is good from undersampled regions, it seems a stretch to say that this represents the ambient traffic emissions mixture. Please discuss this limitation.

11. Page 15 Line 11: If diesel consumption is twice as much as petrol, please discuss in more detail why the traffic signal is dominated by the gasoline tracer i-pentane rather than the diesel tracer ethene. If diesel is more heavily used for freight transport across the country, how well can the three urban traffic samples represent other areas of the country, let alone South Asia? Please make sure to discuss the limitations of this study.

12. Page 15 Line 17: The normalized diesel exhaust graph (Figure 2d) shows very little C5-C8 alkane composition, with just some C6-C8 n-alkanes. Why is C5 included, especially if diesel emissions had <0.2% i-pentane? It would be better not to link the diesel exhaust and evaporative emissions together in this sentence, since their composition is quite different. In Figure 2, why is the diesel exhaust profile much simpler than the diesel evaporative signal? How does this compare to the literature, and what are the uncertainties?

13. Page 17 Line 6: Results are presented here to 2 significant figures without error bars. Please include uncertainty estimates and show the concentrations that were used to create the OFP results. Were average concentrations used? Discuss the limitations of sample variability and small sample size on the results.

14. Page 18: Similar comment as above. This paragraph has detailed OH reactivity and OFP results, without error bars, discussion of uncertainty, or comparison to the literature. For example, on Line 22 I was surprised to see styrene as the largest contributor to OH reactivity from traffic, and strong contributions from isoprene and 1-hexene. What were the measured styrene and isoprene concentrations in the three traffic samples? How do these results compare to the literature?

15. Page 19 Line 10: Same comment, all results need error bars. Here the first three results are probably not statistically different. In discussing BTEX, the fraction may be less important than the concentration of each VOC, especially benzene which is the most toxic. On Line 14, instead of "could severely impact", state how the concentrations compared to exposure limits. Same comment on P21 L22 of the conclusions.

16. Page 20 Line 4: Please use error bars and appropriate significant figures here and in Table 3. Show graphs of some ratios so we can see the variability, especially for small sample sizes. On Line 5, the T/B ratio of 3.41 is more like petrol evaporation rather than vehicular emissions, which has a lower ratio of about 2 in most of the studies cited on Line 6 (Barletta et al., 2005; Russo et al., 2010; Zhang et al. 2013).

17. Page 20 Line 17: Similar to earlier comments, please show the absolute amounts of i-pentane and n-pentane in the source samples, to see how large or small the concentrations were and how variable the ratios were. From Figure 2 there seems to be very little n-pentane in the LPG exhaust, which could lead to a high and uncertain pentane ratio. All ratios should have error bars.

18. Page 21 Line 3: Please clearly state which results were "very different" from the literature based on a statistical analysis. The uncertainties in your study are likely large but haven't been discussed or quantified. On Line 4 I disagree that these profiles can be used for accurate and reliable emissions estimates, since the sample size is so small. The study is a good beginning, but the results need realistic uncertainties.

19. Figure S4: Even though propene is more reactive than ethene, it's surprising to see so little ethene contribution to OH reactivity in smoldering paddy burning since its EF is typically higher than propene for crop residue fires. In Figure 1 I was surprised to see more propene than ethene in the normalized profiles for smoldering paddy burning, different from results for crop residue (rice straw) in Akagi et al. (ACP, 2011) and agricultural residue in Andreae (ACP, 2019). Please show the concentrations used in these calculations and expand the discussion.

20. Page 21 Lines 8-14: These sentences are very similar to the abstract. Page 22 Lines 9-12 and Line 13-15 also repeats from earlier text. The conclusions should provide fresh insights.

Please check the entire manuscript for grammar and typos. For example: Grammar (P4 L23, P6 L4-6, P7 L8-12, P8 L10-11, and so forth in the paper and supplement). Capitalization (P8, L1-8: synthetic, nitrogen, ozone; and so forth).

Page 5 Line 16: Define BTEX.

Page 5 Line 19: What was the sample duration for the whole air samples?

Page 6 Line 18: So 23 vehicles, with one sample per vehicle?

Page 9 Line 21: Please define the sensitivity factors and explain the results in Table S3. Avoid stating "with no drastic changes observed" and be specific about what the results mean.

Page 10 Line 1: What does respective refer to here?

Page 10 Line 7: pAs was first used on Page 9 Line 21; define there.

Page 11 Line 14: Reduced compared to what? Flaming?

Page 11 Line 20: Less styrene compared to what?

Page 12 Line 1: Why at night? I thought the daytime values subtracted off background?

Page 13 Line 18: Hong Kong also has an LPG fuel composition of n-butane > propane > i-butane, with about a 2:1 ratio of n-butane:i-butane, similar to your evaporative results (Tsai et al., ACP, 2006).

Page 14 Line 1: Guo et al. studied Hong Kong, not Taiwan.

Page 14 Line 19: i-Pentane is already known to be a gasoline tracer, but the wording here makes it seem like a novel result.

[Figure]

Page 15: This paragraph is more than 2 pages long.

Page 15 Line 9: Propane isn't listed on Line 6-7 as one of the major NMHC species. How much propane was measured from the traffic?

Page 16 Line 6: Define BSV and BSVI.

Page 17 Lines 6-12: These results are better presented as a Table. Similar comment on Page 18.

Figures 1, 2, S4, S5: State what the shading refers to (aromatic, alkene/alkyne, alkane). In Figure S4 the shading is shifted by one in the alkenes – please correct.

Figure S1: "Smouldering" here but "smoldering" in the main text.

Figure S2: The graphs are too small to clearly see.

Figure S3: There is no Figure S3, just S2 and S4 – please re-number.

Table 1: Please add the descriptions and sample sizes for the three evaporative fuel sources.

Table 2: Please put the compounds in a more logical order so they're easier to find.

Table 2: The abstract states that 49 NMHCs were measured, but Table 2 only lists 48. Was styrene double-counted as both aromatic and alkene? Same comment in the introduction and conclusions.

---

## Referee Comment (RC2) · Anonymous Referee #2 · 27 May 2020

**Referee Report:**
**Non methane hydrocarbon (NMHC) fingerprints of major urban and agricultural emission sources active in South Asia for use in source apportionment studies**

Anonymous Referee

**1 Overview**

Kumar et al. present measurement of 49 NMHCs using GC-FID from samples collected at difference sources (paddy stubble burning, garbage burning, idling vehicular exhaust and evaporative fuel emissions) in northern India. Normalized profiles were calculated based on the measured NMHCs for different sources. The authors identified *i*-pentane as a chemical tracer for petrol vehicular exhaust and evaporative emissions, propane as a chemical tracer for LPG evaporative and LPG vehicular exhaust emissions, and acetylene as a chemical tracer for the biomass fires in flaming conditions.

Instrument analysis is adequate. However the authors need to provide standard gas calibration data for compounds with higher concentrations (> 50 ppbv) to show that the instrument linearity is within tolerance at high mixing ratio level. The sample size for many sources are small (3 or 5 samples), which could introduce large variability and potentially undermine the data quality.

Overall, this study reports the source profiles of NMHCs over an understudied area of the world with complex emission sources. The data should be of interest to the atmospheric science community. This manuscript is within the scope of ACP. I recommend that the manuscript be published in ACP after minor revision.

**2 Minor comments**

(1) "South Asia" in the title covers a broad area. Please revise the title to reflect the specific sampling area (Mohali, India).

(2) Section 2.2: Please provide a schematic diagram of the instrument setup.

(3) Section 2.2: in peak identification and quantification section, there is no discussion on the peak separation. Are all the target compound peaks well separated? If not, how do you resolve the interference? Please provide a typical chromatogram showing all the target compounds taken during a standard gas calibration experiment. Please also include a typical chromatogram taken during the analysis of a sample collected from each source.

(4) Page 9, Line 9: in Figure S2, most compounds do show good linear association. However, certain compounds, such as *m*-Diethylbenzene, *p*-Diethylbenzene, exhibit larger uncertainties at about 20 ppbv mixing ratio level and larger deviations from the fitted line compared to the rest of the compounds. Please provide correlation coefficient values (with 4 significant figures) for all the target compounds in Figure S2.

(5) Page 9, Line 12: please list all compounds with concentrations > 50 ppbv after dilution.

(6) Page 9, Line 13–14: please provide data (similar to Figure S2) to show the standard gas calibration results for the target compounds with concentrations of up to 200 ppbv. Please also include correlation coefficient values (with 4 significant figures) for all the target compounds.

(7) Page 11, Line 5–7: please provide data to show a comparison of the target compound mixing ratios between before the fire and during the fire. Are the mixing ratios taken just before the fire (deemed as the ambient background level) significantly lower than during the fire? If not, does this bring large uncertainty to the interpretation of the calculated emission profiles?

(8) Page 17, Line 6–12: there is no need to list all the rankings here since the reader can get this information from Figure 3.

(9) Page 18, Line 10: it would be more informative to provide the rate coefficient value range for reactions between C2-C4, C5-C8 alkanes and OH here for comparison purpose instead of just saying "more reactive towards OH radical".

(10) Page 19, Line 3–15: using the fraction of BTEX to assess the health risks may not be the best way since most guidelines use concentration as benchmark. For example, smoldering paddy stubble fire (13%) > diesel evaporative emissions (11%) does not necessarily indicate that the BTEX concentration in diesel evaporative emissions is less than smoldering paddy stubble fire. Please provide the concentration (with uncertainty) here as well to assist the discussion.

(11) Page 33: the sample size for some NMHCs sources (e.g., paddy stubble burning, garbage burning, and traffic) are quite small (3 or 5). Please provide mixing ratios of the target compounds (together with uncertainties) in Figure 1 and Figure 2. If there are large uncertainties in the mixing ratios, please justify that such small sample size is representative of the sampling areas or even feasible to be extrapolated to represent South Asia.

**3  Technical corrections**

(1) Page 2, Line 13: "PMF": please give the full name of any acronym when it appears for the first time in the manuscript.

(2) Page 2, Line 15: "LPG": please give the full name here.

(3) Page 3, Line 1: "BTEX": please list all the compounds in BTEX.

(4) Page 3, Line 2: "most polluting": please provide data to support this conclusion.

(5) Page 4, Line 1–2: "North West-Indo Gangetic Plain" → "North West-Indo Gangetic Plain (NW-IGP)".

(6) Page 9, Line 21: please define "pAs" here.

(7) Page 16, Line 6: pleaes provide the full name for BSV and BSVI.

**References**

---

## Author Comment (AC1) · 3 Sep 2020

Please find attached the consolidated author response in the zipped file attached herewith

Please also note the supplement to this comment:
https://acp.copernicus.org/preprints/acp-2019-1172/acp-2019-1172-AC1-supplement.zip

---

## Author Response (AR1)

**Non methane hydrocarbon (NMHC) fingerprints of major urban and agricultural emission sources active in South Asia for use in source apportionment studies**

**Ashish Kumar et al. 2020**

**Authors' consolidated response to reviewers:**

At the outset we would like to thank the esteemed anonymous reviewers for their careful reading of the manuscripts, positive recommendations and helpful suggestions and constructive criticism to improve the manuscript. Their comments have helped us improve the manuscript significantly and we gratefully acknowledge the same. In the revised version we have undertaken every effort including additional replicates to heed the valid concerns of the reviewers. This and interruptions in our ability to carry out some additional experiments to address specific concerns raised by the reviewers due to COVID-19 shutdowns, was also the reason for the request to extend the deadline for submission of the revised MS, that was graciously granted by the Editor, Three main points were made by both reviewers which are summarized below:

1) It was recommended that uncertainties and variability of the individual compounds in the reported source profiles should be provided. We are pleased to include these in the revised version in all the revised Figures. To do this accurately, we used the normalised value of the NMHC in each sample to calculate the mean normalized ratio for a particular emission source. The standard error of mean for these values yields the overall uncertainty which also includes the sample to sample variability. As expected, this did not change the reported values in the previous version significantly and enables us to compute the uncertainty correctly. Earlier we had taken mean of the mixing ratios of the NMHC using all samples of the particular emission source and used these values to calculate the normalised ratios for the source, which led to loss of information and increased the uncertainty.

2) It was recommended that in addition to the normalised profiles, the absolute values measured in individual samples with the TD-GC-FID system be provided, including for the background/ control samples. Calibration data at higher mixing ratios of upto ~200 ppb and example chromatograms be provided in addition. All of this is now included in the revised manuscript in the supplements. Two new supplementary files Supplement 2 (pdf file which provides example chromatograms as desired by reviewer 2 ) supplement file 3 (excel sheet which provides the mixing ratios measured in individual samples with the TD-GC-

FID system after dilution, mixing ratios with uncertainty in sample after correcting for the dilution mixing ratios, and mixing ratios of the NMHCs in the background/ control samples).

3) It was pointed out quite rightly that only one sample each for the source profiles of evaporative fuel emissions poses a limitation on the confidence for the reported evaporative fuel NMHC fingerprints. To address this, we carried out additional sampling and analyses for these sources and now the evaporative profiles are derived from 10 samples each, instead of only 1. While the source fingerprint did not alter, the additional analyses helped us assess the variability within each source type and also obtain additional insights, which was impossible to do using just one sample. We are grateful to the reviewers for this improvement.

Below we list the point-wise replies to both the reviewer's specific comments. The reviewers' comments are shown in black while our replies are in blue. Changes made in response to the specific points are in red, to make it easy for the readers.

We believe that all the valid concerns have been addressed comprehensively in the revised submission with huge effort and hope that the same may now be found suitable for publication in ACP.

Thanks and regards,

Vinayak Sinha, On behalf of all co-authors

**Response to Anonymous referee #1**

Please find the point wise replies (**in blue**) to the referee's comments (**in black**) for easy perusal.

**General comments:**

This paper presents NMHC fingerprints of urban and agricultural emission sources in northern India based on whole air sampling. Similar to previous studies, i-pentane was found to be a tracer for petrol vehicular exhaust, propane for LPG emissions, acetylene for biomass burning, and alkenes for diesel exhaust. The authors report significant emissions of propane from paddy stubble fires, and suggest that isoprene can be used to distinguish paddy stubble burning from garbage burning.

This is an interesting research paper from an understudied part of world with complex emission sources. However several issues need to be addressed before the paper can be published, especially concerning uncertainty, data variability, and small sample size. The actual concentration data should be shown for both background and source samples, including average values with uncertainties. So far the data are shown as normalised source profiles, so the reader can't see the range of the data, how well the precision (which was determined

for 5 ppb) applies to the measured concentrations, or how often the data were close to the detection limit. Many results in the paper need error bars. Another issue is the small sample size (as small as 1 sample for some sources) and how that limits the study and its application to other areas of India or South Asia. Also, since some of the source profile results are known from previous studies, the last sentence of the abstract perhaps overstates the novelty of this work.

**Author response**: We sincerely thank the reviewer for her/his insightful review of the manuscript and general positive comments acknowledging the importance of this study for an understudied complex emission environment like India. The in-depth comments and suggestions raised by reviewer regarding the data uncertainty, variability and sample size were helpful in improving the overall clarity, presentation and discussion of the work. Detailed response to each comment and changes made in the manuscript are listed below.

**Specific comments:**

**1. The title states "South Asia" but the paper is based on limited measurements near Mohali in northern India. What basis is there for extrapolating these results to all of South Asia? Emissions can vary from country to country (Page 5, Line 4) and even within countries. "Indo-Gangetic Plain" is probably a better choice for the title.**

**Author response:** South Asia was included in the title "Non methane hydrocarbon (NMHC) fingerprints of major urban and agricultural emission sources active in South Asia for use in source apportionment studies" to highlight that paddy stubble burning, open waste burning, vehicular exhaust emissions and fossil fuel evaporative emissions are commonly occurring emission sources in the region of South Asia (Chandra and Sinha, 2016; Gadde et al., 2009; Liu et al., 2008; Mo et al., 2016; Sharma et al., 2019; Streets et al., 2003). Also, for better representation, the samples of evaporative emissions have now been collected from the most common brands of petrol, diesel and LPG fuels sold in India, Nepal, Bangladesh and Sri Lanka (Indian oil, Hindustan Petroleum, Bharat Petroleum, Bharatgas and Indane). However, upon reading both the reviewers' advice, we have removed "active in South Asia" from the title of the revised version. This way the NMHC fingerprints can be used for source apportionment studies wherever it is relevant.

The revised title now reads as follows:

"Non methane hydrocarbon (NMHC) fingerprints of major urban and agricultural emission sources for use in source apportionment studies"

**2. Page 5 Line 21: The sample size is quite small, with three flaming and three smoldering samples from paddy stubble fires, and five each from garbage burning. While any information is good from understudied sources and source regions, please discuss the limitations of the small sample size. Also show the reader the actual data (concentrations) by including a statistics table with uncertainties.**

**Author response:** We appreciate the reviewer's point about the small sample size and the uncertainties associated with it and have made revisions to the manuscript. For fuel evaporative emissions where earlier we had results from only one sample, we have now carried out analyses for ten samples for each of the evaporative fuels.

We also regret that we did not explain certain aspects pertaining to collection of the samples more clearly in the original version. As per law, open burning including for crop residue fires is prohibited in the region, and so the open burning activity occurs opportunistically and it is very difficult to convince actors to allow us to collect samples. Logistically therefore, it is a big challenge to obtain samples from such fires lit by the concerned parties.

Other approaches for determining emission factors from such burning activities either rely on controlled burns in a laboratory facility under conditions close to how they are burnt in the environment or employ mobile platforms to sample from an area over which burning is occurring or likely to occur at some distance from the emission source. Examples of such pioneering studies are available in Stockwell et al., 2014 who burnt a total of nine samples from South Asia (four from Taiwan, three from China and one from Malaysia) and Stockwell et al., 2016 who burnt one sample of paddy residue and six samples of garbage. Christian et al., 2003 earlier reported the emissions using three fires of paddy residue from Indonesia. From China, Zhang et al., 2008 reported emissions from four paddy straw fires while Mo et al., 2016 studied emissions from two fires. Zhang et al., 2013 measured emissions from twenty paddy straw fire samples, but just like other previous studies, it was carried out in a controlled laboratory environment simulating the on-field fires. These extant approaches have their trade-offs. While our method samples the fire smoke as it is released into the ambient air by the actors, the trade-off is that unlike laboratory burns one cannot reproduce as many burns as in controlled laboratory burns. While the latter are good as they capture the full burn, the former represent the actual burns without any possibility of deviation from burn conditions.

To summarize, in the revised version all source profiles were derived from atleast three or more samples.

We acknowledge the reviewer's point about variability by including the same in the revised version.

The excel file provided as part of the revised supplementary material provides for each sample and source, the mixing ratios (in ppb) all measured NMHCs after dilution with the TD-GC-FID system, the mixing ratios in the samples after accounting for dilution, as well as the actual mixing ratios for ambient traffic samples, background mixing ratios and the measurement uncertainties (in ppb).

Table 1 of the original version has also been updated with collection details for the evaporative fuel emission profiles.

**3. Page 7 Line 1: Does the text here mean that Panels e-g in Figure 1 and Figure S4 are based on just one sample each? This is a major source of uncertainty, since there is no way to quantify the variability in the evaporative source emissions or assign uncertainty. Please discuss this limitation.**

**Author response:** We thank the reviewer for this comment. In the original submission we indeed presented results derived from just one sample as we thought that these sources are industrially manufactured and therefore have well-regulated chemical composition.

However as pointed out by the esteemed reviewer, we agree that this posed a serious limitation to assess the variability of the evaporative source emissions. In order to address this very important and valid concern we have now carried out analyses using 10 samples for each fuel evaporative source with additional diversity in source samples (e.g. make and grade).

First we note that to address a subsequent suggestion of the reviewer to arrange the compounds on the x-axis of Figures 1 and 2 in a sequence that is easier to locate compounds, we now group the isomers together starting with aromatic compounds, followed by alkenes and alkanes in ascending number of carbon atoms within each functional group.

There are no unexpected or major changes for diesel evaporative emissions and petrol evaporative emissions due to additional samples relative to the emission profiles reported from the single sample in the original submission. However for LPG evaporative emissions while the top three compounds remained the same as reported from the single sample (which was a domestic usage LPG cylinder) reported in previous version namely propane, n-butane and i-butane, we found differences in the relative ranking between commercial grade and domestic grade LPG cylinders. In India, LPG is of two types: domestic LPG for household cooking and commercial LPG for various commercial and industrial applications like hotels, restaurants, metallurgical applications, textiles, automotive etc. The additional analyses revealed that the domestic LPG was a propane-butane mixture with more propane than butane. In contrast, the commercial LPG was found to have more butane with little or no propane and relatively higher butenes in its evaporative emissions. The results have now been incorporated into the revised manuscript and the LPG evaporative samples now have a source profile for domestic LPG and another one for commercial LPG. The confidence in the measured profiles and results is now definitely improved due to better statistics and we thank the reviewer for highlighting this limitation in previous version.

The text in the manuscript also has been modified as follows:

Page 7 Line 1:

*"The fuel evaporative emissions samples (ten each from the headspace of LPG, petrol and diesel) were collected in three Indian cities namely Mohali, Chandigarh and Panchkula on 13-14 Aug 2020. In India, the commonly used LPG is of two types: domestic LPG for household cooking and commercial LPG for various commercial and industrial applications like hotels, restaurants, metallurgical applications, textiles, automotive etc. Out of the total 10 samples of LPG evaporative emissions, 5 samples each were of domestic and commercial LPG types. For better representation, the samples of evaporative emissions were also collected from the most common brands of petrol, diesel and LPG fuels sold all over India and in Nepal, Bangladesh and Sri Lanka (Indian oil, Hindustan Petroleum, Bharat Petroleum, Bharatgas and Indane). The average ambient temperature and relative humidity during the sample collection were 30ºC and 75% respectively."*

**4. Page 10 Line 11: Precision usually varies with concentration, but here the stated precisions are for 5 ppb. How does this change as the measurements go below 5 ppb or approach the detection limit? The precisions listed in Table 2 are very good (0.1-0.4%), but for example 0.2% on an isoprene value of 200 ppt would be measuring to 0.4 ppt, which seems unrealistic. The overall**

**uncertainty (Page 10, Line 23) will likewise be underestimated if the precision degrades at concentrations less than 5 ppb. Please show the reader the range of concentrations that was measured for each VOC, and discuss how well the precision applies across the range.**

**Author response:** The new supplementary material excel file now provides the absolute concentrations measured in each case from the samples as well as background values. It can be seen for the major compounds determining the normalised source profiles, the sample values were significantly higher than the background values (typically by an order of magnitude or more for smoke and vehicular exhaust). Therefore while we did measure background values for completeness sake and to calculate excess concentrations, they have no significant implications for the reported profiles.

In Table R1 below provided for convenience, it can also be noted that major compounds in the normalised source profiles were always >5ppb when being measured with the TD-GC-FID after dilution and that compounds below 5 ppb had negligible contribution (normalised value < 0.1) to the source profiles reported in Figure 1 and Figure 2. This was the reason why we decided to report the precision at 5ppb. We apologize for not making this point clear in the original version.

We agree that precision error will be higher measurements below 5 ppb and so again for completeness sake in the revised version, the precision error of the instrument was also evaluated of ~1 ppb under using the relative standard deviation of five individual measurements concentration experiment and was found to be in the range of 1-6% for the reported compounds.

These points are now clarified in the revised version in Section 2.2 as follows:

"The supplementary material (excel file format) provides the details for each individual sample about the measured levels measured by the TD-GC-FID system after dilution, absolute concentrations of the compound in the actual sample after correcting for dilution alongwith uncertainty, as well as the values in the corresponding background samples. For the major compounds determining the normalised source profiles (presented and discussed in Section 3.1), the sample values were significantly higher than the background values (even by an order of magnitude or more for smoke and vehicular exhaust source categories). Therefore, while the background values were used to calculate excess concentrations, they hardly played any role in the determination of the emission profiles."

Also on Page 10 lines 10-13, we have modified the text to include information about the precision error at 1 ppb as follows:

"The precision of the instrument was evaluated under the identical conditions using the relative standard deviation of five individual measurements of 1 ppb and 5 ppb of standard gas mixture and was in the range of 1-6% for 1ppb and 0.1-0.5% for 5ppb for the reported compounds."

Table R1: Average concentrations (ppb) and standard deviation (in parentheses) of the most abundant compounds in the averaged normalised source profile of the different emission sources.

| Petrol 2 Wheelers (n = 14) | | | Petrol 4 Wheelers (n = 9) | | | Diesel HDVs (n = 15) | | |
|---|---|---|---|---|---|---|---|---|
| Compounds | Normalised ratio | diluted concentration | Compounds | Normalised ratio | diluted concentration | Compounds | Normalised ratio | diluted concentration |
| Toluene | 0.97 | 96.3 (45.7) | Toluene | 0.69 | 45.8 (38.9) | Ethene | 0.98 | 150.4 (35.2) |
| *i*-Pentane | 0.65 | 87.2 (56.3) | *i*-Pentane | 0.65 | 72.9 (71.6) | Propene | 0.59 | 59.5 (14.3) |
| *m/p*-Xylene | 0.62 | 55.1 (32.2) | *m/p*-Xylene | 0.54 | 36.7 (35.8) | Acetylene | 0.50 | 74.2 (35.0) |
| Acetylene | 0.37 | 95.6 (74.7) | Benzene | 0.42 | 48.8 (55.9) | 1,2,4-TMB | 0.45 | 13.9 (10.6) |
| Ethylbenzene | 0.28 | 24.7 (13.4) | 2,2-Dimethylbutane | 0.32 | 34.1 (39.7) | Benzene | 0.35 | 18.8 (9.4) |
| Benzene | 0.25 | 26.9 (13.6) | Ethane | 0.30 | 81.8 (62.3) | *o*-Xylene | 0.22 | 7.9 (4.7) |
| Ethene | 0.25 | 67.5 (48.9) | *n*-Pentane | 0.18 | 21.6 (23.5) | 1-Butene | 0.18 | 13.5 (3.8) |
| 2,2-Dimethylbutane | 0.24 | 26.9 (15.3) | Ethylbenzene | 0.17 | 8.6 (6.8) | *n*-Octane | 0.18 | 6.1 (3.3) |
| *n*-Pentane | 0.21 | 28.7 (18.8) | 2-Methylpentane | 0.15 | 15.5 (17.4) | Toluene | 0.16 | 7.3 (2.9) |
| Propene | 0.20 | 39.8 (22.8) | Ethene | 0.14 | 47.5 (64.4) | Ethane | 0.07 | 9.5 (3.9) |
| Diesel 3 Wheelers (n = 6) | | | Diesel 4 Wheelers (n = 12) | | | LPG Vehicles (n=9) | | |
| Ethene | 0.95 | 78.1 (18.3) | Ethene | 0.92 | 164.6 (27.2) | *n*-Butane | 0.92 | 132.8 (43.0) |
| Propene | 0.69 | 36.8 (8.9) | Propene | 0.46 | 53.2 (21.8) | *i*-Butane | 0.66 | 93.6 (37.7) |
| Acetylene | 0.59 | 51.7 (25.5) | Acetylene | 0.32 | 58.9 (37.89) | Propene | 0.60 | 107.5 (75.6) |
| Benzene | 0.38 | 10.9 (4.4) | Benzene | 0.27 | 16.5 (5.9) | Propane | 0.59 | 105.5 (57.6) |
| 1-Butene | 0.22 | 8.7 (2.7) | Toluene | 0.11 | 5.6 (2.0) | *trans*-2-Butene | 0.51 | 71.8 (35.2) |
| | | | 1-Butene | 0.10 | 9.9 (6.4) | 1-Butene | 0.40 | 55.2 (40.8) |
| | | | Ethane | 0.08 | 12.9 (8.3) | *cis*-2-Butene | 0.26 | 36.8 (13.4) |
| | | | | | | Ethene | 0.14 | 45.1 (48.5) |
| | | | | | | Acetylene | 0.14 | 44.7 (60.1) |
| CNG Vehicles (n=7) | | | Paddy fires Flaming (n=3) | | | Paddy fires Smouldering (n=3) | | |
| Ethane | 1.00 | 166.1 (17.8) | Ethene | 0.87 | 65.4 (23.5) | Ethane | 1.00 | 132.6 (97.7) |
| Ethene | 0.16 | 29.4 (27.9) | Benzene | 0.78 | 19.6 (2.5) | Propene | 0.63 | 75.9 (61.2) |
| Propane | 0.16 | 18.1 (5.3) | Propene | 0.74 | 38.5 (24.8) | Isoprene | 0.60 | 46.8 (37.9) |

| | | | | | | | | |
|---|---|---|---|---|---|---|---|---|
| Propene | 0.01 | 6.5(12.41) | Ethane | 0.74 | 55.8 (40.6) | Propane | 0.39 | 43.3 (35.3) |
| Acetylene | 0.01 | 14.1 (30.3) | Acetylene | 0.59 | 42.4 (11.3) | Ethene | 0.27 | 49.5 (38.8) |
| Toluene | 0.01 | 14.7 (10.8) | Toluene | 0.32 | 8.3 (4.0) | Toluene | 0.26 | 14.4 (11.5) |
| | | | Isoprene | 0.30 | 9.3 (7.4) | cis-2-Butene | 0.16 | 5.9 (4.8) |
| | | | Propane | 0.21 | 11.2 (8.9) | Benzene | 0.16 | 10.3 (8.2) |
| | | | 1-Butene | 0.16 | 6.3 (4.4) | m/p-Xylene | 0.14 | 6.5 (5.0) |
| | | | | | | 1-Hexene | 0.13 | 10.1 (9.1) |

| Garbage fires Flaming (n=5) | | | Garbage fires Smouldering (n=5) | | | Traffic (n=3) | | |
|---|---|---|---|---|---|---|---|---|
| Benzene | 0.85 | 41.9 (18.0) | Benzene | 0.79 | 48.9 (61.8) | Toluene | 0.97 | 6.8 (1.3) |
| Propene | 0.74 | 61.2 (35.3) | Propene | 0.61 | 44.9 (46.4) | i-Pentane | 0.65 | 12.9 (5.1) |
| Ethene | 0.62 | 83.8 (54.7) | Ethane | 0.55 | 59.2 (65.8) | Acetylene | 0.37 | 10.2 (5.7) |
| Acetylene | 0.50 | 76.9 (56.6) | Toluene | 0.50 | 14.5 (5.1) | | | |
| Ethane | 0.27 | 30.89 (16.7) | Ethene | 0.33 | 39.4 (32.2) | | | |
| Toluene | 0.26 | 9.2 (2.3) | 1-Butene | 0.25 | 11.4 (9.1) | | | |
| 1-Butene | 0.15 | 10.1 (5.7) | Propane | 0.21 | 15.2 (16.6) | | | |
| n-Pentane | 0.12 | 5.3 (2.8) | n-Pentane | 0.16 | 6.2 (6.2) | | | |
| Propane | 0.11 | 8.5 (4.4) | Acetylene | 0.08 | 8.0 (7.3) | | | |

| LPG evaporative emissions (n=10) | | | | | |
|---|---|---|---|---|---|
| Domestic LPG evaporative (n=5) | | | Commercial LPG evaporative (n=5) | | |
| Propane | 1.00 | 147.4 (48.7) | n-Butane | 0.94 | 127.9 (38.9) |
| i-Butane | 0.52 | 55.1 (22.5) | i-Butane | 0.48 | 62.6 (22.4) |
| n-Butane | 0.50 | 53.6 (21.2) | trans-2-Butene | 0.37 | 52.5 (20.6) |
| trans-2-Butene | 0.20 | 20.9 (12.5) | 1-Butene | 0.24 | 27.8 (9.8) |
| 1-Butene | 0.19 | 19.8 (13.2) | Propane | 0.23 | 30.2 (53.0) |
| cis-2-Butene | 0.13 | 12.5 (7.8) | cis-2-Butene | 0.20 | 33.4 (9.4) |
| Ethane | 0.07 | 19.9 (28.2) | i-Pentane | 0.09 | 9.9 (8.5) |
| Propene | 0.05 | 8.1 (5.1) | Ethane | 0.05 | 9.8 (15.9) |

| Petrol evaporative (n=10) | | | Diesel evaporative (n=10) | | |
|---|---|---|---|---|---|
| i-Pentane | 1.00 | 29.4 (32.5) | n-Octane | 0.84 | 24.0 (14.2) |
| n-Pentane | 0.24 | 17.0 (18.5) | n-Heptane | 0.79 | 29.2 (24.5) |
| 2,2-Dimethylbutane | 0.13 | 6.7 (7.7) | o-Xylene | 0.61 | 16.6 (5.6) |
| n-Butane | 0.11 | 25.4 (21.0) | i-Pentane | 0.55 | 29.4 (32.5) |

| | | | | | |
|---|---|---|---|---|---|
| 2-Methylpentane | 0.11 | 11.9 (12.7) | Toluene | 0.54 | 20.9 (16.3) |
| Toluene | 0.05 | 20.9 (16.3) | 1,2,4-Trimethylbenzene | 0.53 | 11.7 (3.9) |
| | | | Methylcyclopentane | 0.52 | 22.6 (21.9) |
| | | | *n*-Hexane | 0.49 | 22.0 (21.3) |
| | | | *m/p*-Xylene | 0.46 | 13.8 (7.2) |
| | | | Cyclohexane | 0.44 | 18.9 (17.8) |

**5. Page 10 Line 19: How often did the measurements go below detection? Please show a statistics table for the background measurements including their uncertainty, and also state the number of background samples that were collected for each source. On Page 11, Line 6 provide more detail about how the background data were used to correct the fire mixing ratios. How did uncertainty in the background measurements impact the normalised source profile concentrations?**

**Author response:** The reply to point 4 above answers the first part of point 5 already. To add to it, the compounds that were found to be lower than the background were considered as non-emitted, i.e, 0 ppb and such compounds had a negligible contribution (normalised ratio < 0.05) to the normalised source profile, so anyway were of no consequence for the results and conclusions. For example, in the paddy flaming fire sample 1, *m/p*-xylene, *o*-ethyltoluene, *i*-propylbenzene, 1,2,4-trimethylbenzene, *i*-butane, *i*-pentane, cyclohexane, 2,2,-dimethylbutane, 2,3-dimethylbutane and 2,3-dimethylpentane were lower than the ambient background concentrations and therefore were considered non-emitted (0ppb). In every emission source sample, the number of such compounds were usually less than 10. Hence the background concentrations and their correction have negligible role in determining the source NMHC fingerprints. This can easily be seen in the new supplementary material (excel sheet) provided with the response that shows the background and actual sample concentrations for each emission source sample as desired by the reviewer.

**6. Page 11 Line 5: From Table 1, benzene is the most abundant normalised compound in the garbage burning. What was its peak concentration? This is relevant for health considerations. Same question on Page 14 Line 5 for traffic.**

**Author response:** Indeed high benzene in normalised profiles is of concern as it shows that benzene is a major emission from that source. However, we would like to caution that the absolute values we measured for paddy fire smoke or garbage fire smoke or fuel evaporative emissions at the source cannot be used for assessing ambient exposure, except if one were to inhale the vapours or smoke directly. The reason is because our measured absolute values for a given sample were collected through direct sampling of the vapours /smoke, without any/negligible ambient dilution, which is a process that occurs to the emitted smoke/vapours. Thus, we limited ourselves to commenting on the fraction of such compounds in the sources for relative comparison.

For the ambient traffic samples, measured in at traffic light, we agree with the reviewer that this could be useful for comparison as it represents concentrations for the traffic exhaust emissions after ambient dilution, and which people at street level would be inhaling. The average concentration of benzene in ambient traffic samples was $6.1 \pm 1.3 \ \mu gm^{-3}$ which was higher than the 5 $\mu gm^{-3}$ annual exposure limit set by The National Air Quality Standards (NAAQS) of India (NAAQS, 2009). However again since the data in the manuscript cannot be used for assessing annual average at these traffic thoroughfares, we would like to refrain from further interpretation.

To clarify this point in the revised manuscript we have added the following new points to the conclusion as follows:

"*Based on our limited measurements of ambient benzene in the traffic throughfares, the mass concentration was $6.1 \pm 1.3 \ \mu g \ m^{-3}$ which is higher than the 5 $\mu gm^{-3}$ annual exposure limit set in the National Air Quality Standards (NAAQS) of India (NAAQS, 2009). Future studies should quantify annual ambient exposure of such toxic compounds from the sources which have high BTEX content for assessing compliance with the annual ambient air quality standards as has been done for paddy residue smoke previously (Chandra and Sinha, 2016).*"

**7. Page 11 Line 9: The term "observed mass concentrations" is used here, but the text above describes an average excess concentration above background. Please clarify what is being discussed here and in the following text. By "acetylene was negligible in the smoldering fires", does this mean that the excess above background was <1%? What is the uncertainty in this result and how does it compare to previous work from smoldering fires?**

**Author response:** We apologize for the confusion due to usage of the word "observed" before mass concentrations. By "observed" we implied "measured" and indeed these represent measured mass concentrations after correcting for the background values. As discussed previously, all the mixing ratios derived from paddy straw burning and garbage burning were corrected for the background, i.e, the background concentrations were subtracted from the source mixing ratio values. These mixing ratios were then converted to mass concentrations ($\mu gm^{-3}$) assuming normal temperature and pressure conditions. In other words, all the discussed concentrations are excess of the background.

To make this clear we have revised the text as follows:

Page 11 Line 9:

"*The largest contributors to the mass concentrations in paddy fires under flaming conditions were....*"

Yes, the concentration of acetylene in smouldering fires was <1% ($\pm 0.04\%$) of the total mass fraction of the smouldering emissions. The results reported in Table 36.2 in the pioneering study of Lobert, 1991 from variety of biomass fuels (but not paddy crop residue) do report that acetylene is primarily emitted during the flaming stage. Recent studies of crop residue burning emissions including from our own group (Kumar et al., 2018) have employed primarily proton transfer reaction mass spectrometers (see for e.g. Inomata et al. 2015)

and/or studied wheat crop residue fires and we could not find information concerning acetylene from paddy crop residue fires in smouldering stage.

**8. Page 11 Line 11: How did the background isoprene concentration compare to the paddy fire concentration? Please show the data.**

**Author response:** The data requested is provided in the new supplementary file (excel sheet; please also see reply to point 4). The isoprene mixings ratio in the background sample was 0.56 ppb while the mixing ratios were greater than 8 ppb for flaming stage and greater than 500 ppb for smouldering stage in all three samples.

**9. Page 13 Line 9: The evaporative LPG emissions were n-butane > propane > i-butane (Page 12 Line 10), but the LPG exhaust is n-butane > i-butane > propane. Why would the composition change from evaporative emissions to exhaust, to yield relatively less propane? How large are the error bars?**

**Author response:** We thank the reviewer for the comment. These results have changed in light of additional samples for LPG evaporative emissions from both domestic and commercial/industrial LPG cylinders. Now both LPG exhaust LPG commercial emissions have $n$-butane > $i$-butane > propane. As discussed previously in response to point 3 of the reviewer, the commercial LPG is used for various commercial and industrial applications including road transport and evidentially we see a similar composition trend in commercial LPG and LPG vehicular exhaust. Error bars are now provided in Figures 1 and 2.

This new discussion is also added to the revised manuscript at Page 12 as follows:

Page 12 Line 10:

*"Propane, n-butane, i-butane and butenes were the major constituents in the LPG evaporative emissions. However, interestingly the composition was different in both the types of LPG evaporative emissions. The domestic LPG evaporative emissions were a mixture of propane and butanes, with propane (40%) by weight as the most dominant emission, followed by n-butane (19%) and i-butane (16%). However, the commercial LPG evaporative emissions were mostly "butane rich" with lower propane (7%) and higher butenes (31% in total from all isomers). n-butane (37%) and i-butane (18%) comprised of nearly half of the total evaporative emissions from commercial LPG cylinders."*

**10. Page 14 Line 22: The three traffic samples were collected in March 2017 (Page 6 Line 9). This is very limited sampling at one time of year (and we don't know what time of day, or how long the sample duration was). While any information is good from undersampled regions, it seems a stretch to say that this represents the ambient traffic emissions mixture. Please discuss this limitation.**

**Author response:** We regret that we did not provide all the details regarding the collection of ambient traffic samples. The samples were collected from three busy traffic junctions in Chandigarh-Mohali area from 3-15 March 2017 during rush hours. The details are as follows:

1) Sample 1:

Sampling site: Sohana Gurudwara Chowk (30.691°N, 76.698°E), ~2m above ground level.

Sampling date and time: 03/03/20017 11:00-11:15

2) Sample 2:

Sampling site: Sector 79/80 Chowk (30.678°N, 76.721°E), ~2m above ground level.

Sampling date and time: 08/03/2017 14:25-14:40

3) Sample 3:

Sampling site: Transport Chowk (30.717°N, 76.812°E), ~2m above ground level.

Sampling time: 15/03/2017 16:50-17:05

Table 1 has now been updated to include the above information.

As the samples were collected during rush hour (afternoon and evening hours) within some of the busiest traffic thoroughfares in two cities (Chandigarh and Mohali), the samples were influenced by a sufficiently diverse fleet mixture similar to most Indian cities. The sampling duration in each case was ~15min, therefore they are not biased by few individual vehicles and can be considered to be representative of the ambient city traffic emissions. These samples are not representative of highway emissions which on the other hand tend to be dominated by Light Duty Vehicles (LDV) and Heavy Duty diesel vehicles. In general diesel LDV are preferably owned by people who regularly drive large distances between cities and diesel HDV's transport ~60% of India's freight share. The latter ply mostly on major highways known as the golden quadrilateral, between the major ports and the urban centres of the Indo-Gangetic plain. There are restrictions within India on the plying of heavy duty diesel trucks within the city.

We do acknowledge that samples from other seasons would have been better as combustion as environmental conditions can increase variability of emissions, but in terms of the major compound mixture emitted significant changes are unlikely.

We clarify these aspects and acknowledge the limitation in the revised version at Page 14 of ACPD version by adding the following new text:

"As the samples were collected during rush hour (afternoon and evening hours) within some of the busiest traffic thoroughfares in two cities (Chandigarh and Mohali) as mentioned in Table 1, the samples were influenced by a sufficiently diverse fleet mixture similar to most Indian cities. The sampling duration in each case was ~15min, therefore they are not biased by few individual vehicles and can be considered to be representative of the ambient city traffic emissions. These samples are not representative of highway emissions which on the other hand tend to be dominated by light duty diesel vehicles and heavy duty diesel vehicles. While more samples collected in other seasons in addition to spring, would have been better as combustion as environmental conditions can affect variability of emissions, changes in terms of the major compound mixture emitted are unlikely."

**11. Page 15 Line 11: If diesel consumption is twice as much as petrol, please discuss in more detail why the traffic signal is dominated by the gasoline tracer i-pentane rather than the diesel tracer ethene. If diesel is more heavily used for freight transport across the country, how well can the three urban traffic samples represent other areas of the country, let alone South Asia? Please make sure to discuss the limitations of this study.**

**Author response:** This point has been partially explained already in reply to point 10. According to the data of officially registered vehicles in India (MoRTH, 2018) the Indian vehicular fleet on an average is comprised of 73% of two wheelers which run on petrol/gasoline and consume ~61% of total gasoline used in transport sector (Nielsen, 2013). Even within the different states of India (except a few states in North east which have <1% of total registered vehicles in India), the two wheelers comprise of >50% (upto 80% in some states) of vehicular fleet. Furthermore, vehicles were intentionally chosen for sampling based upon their sales, engine types, manufacturer, etc to represent the Indian vehicular fleet. Therefore, once can have confidence that the urban (city) traffic emissions across India are dominated by the petrol/gasoline emissions and hence our samples can be considered representative of traffic emissions within the city.

The consumption of diesel is more than that of petrol because ~40% of the total diesel consumed by transport sector in India is consumed by heavy duty vehicles like buses and trucks that run over large distances across inter-city highways (on lower mileage).

We have revised the relevant text in the revised version to make this clearer:

"This is because the maximum diesel consumption (40%) is by heavy duty vehicles (HDVs) which run over large distances *across inter-city highways* and have lower mileage than other vehicle classes."

**12. Page 15 Line 17: The normalised diesel exhaust graph (Figure 2d) shows very little C5-C8 alkane composition, with just some C6-C8 n-alkanes. Why is C5 included, especially if diesel emissions had <0.2% i-pentane? It would be better not to link the diesel exhaust and evaporative emissions together in this sentence, since their composition is quite different. In Figure 2, why**

**is the diesel exhaust profile much simpler than the diesel evaporative signal? How does this compare to the literature, and what are the uncertainties?**

**Author response:** We agree with the reviewer's suggestions thank him/her for the comment. In the revised version we have now excluded the C5 compounds while discussing the diesel exhaust results and modified the manuscript as follows:

Page 14 line 11 was modified as follows:

"Furthermore, the BTEX (16%) and C6-C8 (9%) emissions were also lower than the petrol exhaust emissions."

Page 14 line 19: removed

Page 15 line 17 was modified as follows:

*"As discussed previously, the diesel vehicular exhaust were dominated by heavier C6-C8 alkanes, alkenes and aromatics which are key precursors in OH reactivity and ozone formation."*

In Figure 2, the diesel exhaust profile was much simpler than the diesel evaporative signal. Diesel engines have better combustion efficiency (Reiter and Kockelman, 2016) for several hydrocarbons which get combusted and yield the characteristic source profile of diesel exhaust containing ethene, propene and acetylene (Liu et al., 2008; Schauer et al., 1999). The alkane content in our diesel evaporative emissions was (54.6 ± 7.4 %) and was comparable to the Guangzhou diesel (53.8 ± 10.0 %), Zhuhai diesel (57.4 ± 5.3 %) and Macau diesel (64.3 ± 1.6 %) (Tsai et al., 2006) along with a characteristic trend of higher fraction of heavier alkanes and C8-C10 aromatics. As already mentioned, the overall uncertainty containing also the sample to sample variability are also shown in all the figures now.

In the revised version we also found it useful to resolve the diesel and petrol exhaust emissions by vehicle types (e.g. 2 wheeler, 4 wheeler, heavy duty and light duty) as that can have an influence on the combustion efficiency due to the technology the vehicle is equipped with for combustion.

We have added these points in the revised manuscript and modified the text as follows:

Page 14 Line 7-14:

*"Figure 2d shows the tailpipe emissions from light duty three vehicles, light duty four wheelers and heavy duty vehicles fuelled by diesel. The diesel exhaust emission profiles were much simpler than the petrol exhaust emissions. Alkenes and alkyne were the major constituents of the diesel vehicular exhaust contributing 58% to the total NMHC emissions. Furthermore, the BTEX (16%) and C6-C8 (8%) emissions were also lower than the petrol exhaust emissions. Diesel engines are known for their better combustion efficiency (Reiter and Kockelman, 2016) due to which most of the higher hydrocarbons get combusted and yield the characteristic source profile*

*of diesel exhaust containing ethene (26%), propene (14%) and acetylene (11%)by weight percent (Liu et al., 2008; Schauer et al., 1999). There were no major differences in the profiles of different types of diesel vehicles and ethene, propene, acetylene, benzene, 1,23-trimethylbenzene and 1-butene were the most dominant NMHCs. However, the fraction of C9-C10 aromatics was higher in heavy duty vehicles (19%) and three wheelers (11%) as compared to four wheelers (6%)."*

Page 14 Line 15: removed

**13. Page 17 Line 6: Results are presented here to 2 significant figures without error bars. Please include uncertainty estimates and show the concentrations that were used to create the OFP results. Were average concentrations used? Discuss the limitations of sample variability and small sample size on the results.**

**Author response:** The OFP was calculated separately for each sample within an emission source using its respective NMHC mass fractions (wt/wt %) and MIR rates. This was then averaged and reported with the standard error of mean. The concentrations used to derive mass fractions are presented in the excel sheet. We agree with the reviewer that the small sample size of some sources can be a limitation and therefore now in order to ascertain any differences between the average OFP of different sources, we applied the Tukey's pairwise HSD (honest significant for difference of mean) statistical test, which account for sample size. The summary of the test results is provided in the new Table S5 provided in the supplement (also provided below for ease). Out of 55 possible pairwise comparisons 28 pairs show statistically significant differences with $\geq 2\ \sigma$ confidence, 6 are only significant at 1 $\sigma$ level and 21 pairs are not significant. The highest number of non-significant differences are observed for flaming (5) and smouldering (4) paddy residue due to the small number of paddy residue burning samples, and for garbage burning samples (6 for flaming and 5 for smouldering) as garbage is not a standardized material and consequently there is a large fire to fire variability. Relatively high statistical significance levels were observed for pairs involving combustion engine exhaust e.g. Diesel vehicles and CNG vehicles.

The results from the statistical test has been added to the supplement as Table S5 and the relevant text in the manuscript has been replaced by the following new text:

Page 17 Line 6-12:

*"In order to ascertain any statistical difference between the average OFPs of the emission sources, we carried out Tukey's pairwise honestly significant difference test (which accounts for sample size) and the summary of the test results is provided in Table S5. Based on the statistical test, it could be concluded with more than 95% confidence that CNG vehicular emissions and the fuel evaporative emissions had different OFPs compared to other emission sources. The averaged OFP for the emission sources was: diesel vehicle exhaust (6.5 ± 0.6 gO3/gNMHC) , smoldering paddy stubble fire (5.9 ± 0.2 gO3/gNMHC) , LPG vehicle exhaust (5.7 ± 1.1 gO3/gNMHC) , flaming paddy stubble fire (5.2 ± 0.9 gO3/gNMHC) , flaming garbage fire (4.9 ± 1.1 gO3/gNMHC) , smoldering garbage fire (4.4 ±*

[Figure]

*1.3 gO3/gNMHC) , LPG evaporative emissions (4.5 ± 1.6 gO3/gNMHC) , petrol vehicle exhaust (3.9 ± 0.7 gO3/gNMHC) , diesel evaporative emissions (3.6 ± 0.9 gO3/gNMHC) , petrol evaporative emissions (2.0 ± 0.4 gO3/gNMHC) , CNG vehicle exhaust (1.5 ± 0.8 gO3/gNMHC)."*

**Table S5:** Summary of Tukey pairwise HSD (honestly significant difference) test results performed for the averaged OFP values from the different emission sources. The significant differences in the mean values at confidence interval > 95% are ascertained by p (same mean) < 0.05 and are highlighted in bold.

| | Diesel vehicles | Paddy smoldering | Paddy flaming | LPG vehicles | Garbage flaming | Garbage smoldering | Petrol vehicles | LPG evaporative | Diesel evaporative | Petrol evaporative | CNG vehicles |
|---|---|---|---|---|---|---|---|---|---|---|---|
| Diesel vehicles | - | - | σ | - | 2σ | 3σ | 4σ | 4σ | 4σ | 4σ | 4σ |
| Paddy smoldering | | - | - | - | - | σ | 2σ | 2σ | 4σ | 4σ | 4σ |
| Paddy flaming | | | - | - | - | - | σ | - | 2σ | 4σ | 4σ |
| LPG vehicles | | | | - | - | σ | 2σ | 2σ | 3σ | 4σ | 4σ |
| Garbage flaming | | | | | - | - | - | - | σ | 4σ | 4σ |
| Garbage smouldering | | | | | | - | - | - | - | 4σ | 4σ |
| Petrol vehicles | | | | | | | - | - | - | 2σ | 4σ |
| LPG evaporative | | | | | | | | - | - | - | 4σ |
| Diesel evaporative | | | | | | | | | - | σ | 3σ |
| Petrol evaporative | | | | | | | | | | - | - |
| CNG vehicles | | | | | | | | | | | - |

**14. Page 18: Similar comment as above. This paragraph has detailed OH reactivity and OFP results, without error bars, discussion of uncertainty, or comparison to the literature. For example, on Line 22 I was surprised to see styrene as the largest contributor to OH reactivity from traffic, and strong contributions from isoprene and 1-hexene. What were the measured styrene and isoprene concentrations in the three traffic samples? How do these results compare to the literature?**

**Author response:** The average concentrations of styrene, isoprene and 1-hexene measured from the traffic samples were $1.39 \pm 0.18$ ppb, $0.69 \pm 0.08$ ppb and $1.57 \pm 0.17$ ppb respectively. Amongst all the 49 NMHCS from traffic samples, these compounds rank 13th, 15th and 30th in terms of mass concentrations, however their contribution to the OH reactivity is much higher because of their high rate constant when reacting with hydroxyl radicals ($k_{OH}$ values from (Atkinson, 1997; Atkinson et al., 1989)):

isoprene: $k_{OH} = 10.0$ x $10^{-11}$ cm$^3$ molecule$^{-1}$ s$^{-1}$ at 298K

styrene: $k_{OH} = 5.8$ x $10^{-11}$ cm$^3$ molecule$^{-1}$ s$^{-1}$ at 298K

1-hexene: $k_{OH} = 3.7$ x $10^{-11}$ cm$^3$ molecule$^{-1}$ s$^{-1}$ at 298K

Isoprene, styrene and 1-hexene have been reported previously in various traffic and tunnel experiments across the world (Barletta et al., 2002; Borbon et al., 2001; Ho et al., 2009; Mugica et al., 1998; Zhang et al., 2018). Our traffic samples have comparable mixing ratios of isoprene observed from average roadside ambient air measurements in Karachi ($1.2 \pm 0.9$ ppb) (Barletta et al., 2002), 43 Chinese cities ($0.86 \pm 0.83$ ppb) (Barletta et al., 2005) and Longchuan tunnel, Hefei ($0.47 \pm 0.20$ ppb) (Deng et al., 2018), but higher than Chapultepec Avenue tunnel, Mexico city ($0.17 \pm 0.02$ ppb) (Mugica et al., 1998), Fu Gui Mountain Tunnel ($0.14 \pm 0.36$ ppb) (Zhang et al., 2018). In the Hong Kong tunnel experiment (Ho et al., 2009) and Taipei tunnel experiment (Hwa et al., 2002) isoprene was however undetectable. This variability in isoprene emissions from traffic/vehicular exhaust have been previously attributed to variable fuel types, vehicular engines and maintenance, driving patterns and sampling strategies. The mixing ratios of styrene and 1-hexene measured in our traffic samples were higher than the Fu Gui Mountain Tunnel (styrene: $0.08 \pm 0.00$ ppb, 1-hexene: $0.07 \pm 0.00$ ppb), but comparable to 1-hexene reported from Taiwan tunnels (Cross-Harbor Tunnel ($0.99 \pm 0.20$ ppb), Chung-Bor Tunnel ($3.29 \pm 2.36$ ppb) and Chung-Cheng Tunnel ($2.49 \pm 1.27$ ppb) (Chen et al., 2003). Though high mixing ratios of styrene in our samples are remarkable, it has been previously reported that styrene is one of the major VOCs emitted from the diesel LDVs especially in cold transient mode (Tsai et al., 2012). Amongst our three traffic samples, maximum mixing ratios of isoprene ($1.11 \pm 0.06$ ppb), styrene ($2.31 \pm 0.16$ ppb) and 1-hexene ($2.38 \pm 0.14$ ppb) were observed in Transport chowk ($30.717^oN$, $76.812^oE$) which is one of the busiest traffic junction in Chandigarh city during rush hours and witnesses a large vehicular fleet of diesel run commercial LDVs.

We now clarify these points in the revised manuscript by adding above discussion as follows

Page 19 Line 2:

*"High contributions to OH reactivity from styrene, isoprene and 1-hexene are noteworthy. Even though these compounds were not the most abundant in the traffic samples by mass, however they are very reactive with hydroxyl radicals in ambient air.(isoprene: $k_{OH} = 10.0$ x $10^{-11}$ cm$^3$ molecule$^{-1}$ s$^{-1}$; styrene: $k_{OH} = 5.8$ x $10^{-11}$ cm$^3$ molecule$^{-1}$ s$^{-1}$; 1-hexene: $k_{OH} = 3.7$ x $10^{-11}$ cm$^3$*

*molecule$^{-1}$ s$^{-1}$ at 298K) (Atkinson, 1997; Atkinson et al., 1989). Isoprene, styrene and 1-hexene have been reported previously in various traffic and tunnel experiments across the world (Barletta et al., 2002; Borbon et al., 2001; Ho et al., 2009; Mugica et al., 1998; Zhang et al., 2018). Our traffic samples have comparable mixing ratios of isoprene observed from average roadside ambient air measurements in Karachi (1.2 ± 0.9 ppb) (Barletta et al., 2002), 43 Chinese cities (0.86 ± 0.83 ppb) (Barletta et al., 2005) and Longchuan tunnel, Hefei (0.47 ± 0.20 ppb) (Deng et al., 2018), but higher than Chapultepec Avenue tunnel, Mexico city (0.17 ± 0.02 ppb) (Mugica et al., 1998), Fu Gui Mountain Tunnel (0.14 ± 0.36 ppb) (Zhang et al., 2018). In the Hong Kong tunnel experiment (Ho et al., 2009) and Taipei tunnel experiment (Hwa et al., 2002) isoprene was however undetectable. This variability in isoprene emissions from traffic/vehicular exhaust have been previously attributed to variable fuel types, vehicular engines and maintenance, driving patterns and sampling strategies. The mixing ratios of styrene and 1-hexene measured in our traffic samples were higher than the Fu Gui Mountain Tunnel (styrene: 0.08 ± 0.00 ppb, 1-hexene: 0.07 ± 0.00 ppb), but comparable to 1-hexene reported from Taiwan tunnels (Cross-Harbor Tunnel (0.99 ± 0.20 ppb), Chung-Bor Tunnel (3.29 ± 2.36 ppb) and Chung-Cheng Tunnel (2.49 ± 1.27 ppb) (Chen et al., 2003). Though high mixing ratios of styrene in our samples are remarkable, it has been previously reported that styrene is one of the major VOCs emitted from the diesel LDVs especially in cold transient mode (Tsai et al., 2012). Amongst our three traffic samples, maximum mixing ratios of isoprene (1.11 ± 0.06 ppb), styrene (2.31 ± 0.16 ppb) and 1-hexene (2.38 ± 0.14 ppb) were observed in Transport chowk (30.717°N, 76.812°E) which is one of the busiest traffic junction in Chandigarh city during rush hours and witnesses a large vehicular fleet of diesel run commercial LDVs."*

**15. Page 19 Line 10: Same comment, all results need error bars. Here the first three results are probably not statistically different. In discussing BTEX, the fraction may be less important than the concentration of each VOC, especially benzene which is the most toxic. On Line 14, instead of "could severely impact", state how the concentrations compared to exposure limits. Same comment on P21 L22 of the conclusions.**

**Author response:** As mentioned already in reply to point 13, we respectfully disagree about using the concentration exposure as these are not ambient concentration exposures because they ignore atmospheric mixing and dilution. The purpose of using BTEX fraction which is a well know metric (Słomińska et al. 2014; Truc et al 2007) is simply to compare the mass fractional content between the sources' emissions. The statistical differences in the average BTEX fraction between the different emission sources were ascertained by Tukey's pairwise honestly significant difference test and the summary for the same is provided in new Table S6 (also reproduced below for ease). Based on the statistical test, it could be concluded with more than 95% confidence that diesel and petrol evaporative emissions, diesel vehicles and smouldering paddy fires had different average BTEX fraction as compared to other emission sources. Out of 28 possible pairwise comparisons 14 pairs show statistically significant differences with ≥2 σ confidence, 3 are only significant at 1 σ level and the rest were not significant.

We have now added the results from the statistical test to the supplement as Table S6 and replaced the main text in the manuscript has been modified as follows:

Page 19 Line 10:

*"Using the BTEX fraction which is a well know metric (Słomińska et al. 2014) is useful for comparing the mass fractional BTEX content in emissions from the sources. The statistical*

*differences in the average BTEX fraction between the different emission sources were ascertained by Tukey's pairwise honestly significant difference test and the summary for the same is provided in Table S6 (also reproduced below for ease). Based on the statistical test, it could be concluded with more than 95% confidence that diesel and petrol evaporative emissions, diesel vehicles and smouldering paddy fires had different average BTEX fraction as compared to other emission sources. Out of 28 possible pairwise comparisons, 14 pairs show statistically significant differences with ≥2 σ confidence, 3 are only significant at 1 σ level and the rest were not significant. The fraction of BTEX in the different emission sources was: petrol vehicle exhaust (27 ± 5%), smouldering garbage fire (26 ± 1%), flaming garbage fire (24 ± 8%), flaming paddy stubble fire (22 ± 5%), diesel vehicle exhaust (19 ± 2%), diesel evaporative emissions (17 ± 2%), smouldering paddy stubble fire (13 ± 1%) and petrol evaporative emissions (3 ± 1%)."*

**Table S6:** Summary of Tukey pairwise HSD (honestly significant difference) test results performed for the averaged BTEX% from the different emission sources. The significant differences in the mean values at confidence interval > 95% are ascertained by p (same mean) < 0.05 and are highlighted in bold.

| | Petrol vehicles | Garbage smoldering | Garbage flaming | Paddy flaming | Diesel vehicles | Paddy smouldering | Diesel evaporative | Petrol evaporative |
|---|---|---|---|---|---|---|---|---|
| Petrol vehicles | - | - | - | - | **2σ** | **4σ** | **2σ** | **4σ** |
| Garbage smouldering | | - | - | **2σ** | **4σ** | **4σ** | **4σ** | **4σ** |
| Garbage flaming | | | - | - | **2σ** | **3σ** | σ | **4σ** |
| Paddy flaming | | | | - | - | σ | - | **4σ** |
| Diesel vehicles | | | | | - | - | - | σ |
| Paddy smouldering | | | | | | - | - | - |
| Diesel evaporative | | | | | | | - | **2σ** |
| Petrol evaporative | | | | | | | | - |

**16. Page 20 Line 4: Please use error bars and appropriate significant figures here and in Table 3. Show graphs of some ratios so we can see the variability, especially for small sample sizes. On Line 5, the T/B ratio of 3.41 is more like petrol evaporation rather than vehicular emissions, which has a lower ratio of about 2 in most of the studies cited on Line 6 (Barletta et al., 2005; Russo et al., 2010; Zhang et al. 2013).**

**Author response:** We thank the reviewer for this important comment which has made us revisit our approach for determining the molar ratios to include uncertainties for the molar ratios. All molar ratios have now been calculated to propagate the standard error of the mean accounting for the number of samples. Thus, we calculate the VOC1/VOC2 molar ratio for each sample of a particular source (e.g. paddy fire flaming) and then take the average of these individual ratios and use the standard deviation and number of samples for calculating the overall uncertainty of the ratio. Earlier we were taking the average of VOC1 in all samples and the average of VOC2 in all samples and then reporting the values as the ratios of these two average values without accounting for sample size and uncertainties. While none of the molar ratios values change by much and none of the conclusions change from the original submission, we are now able to account for the uncertainty and sample size.

In the revised version the average molar ratios are presented with standard error of mean after including the results of the additional samples from evaporative emissions (please see also reply to point 3). The uncertainties are now included in Table 3 too and the text. Given the detection limit for these compounds significant figures were appropriate.

To clarify the same and include the uncertainties as suggested by the reviewer, the revised text in manuscript at the relevant lines now reads as follows:

Page 20 line 4 of original submission:

*"Toluene/benzene (T/B) ratio is a widely used molar ratio in identifying the vehicular emission sources (Barletta et al., 2002; Barletta et al., 2005). T/B measured for traffic in this study was 3.54 ± 0.21 which is comparable to previous studies from busy traffic junctions and tunnels in Karachi, Pakistan (2.2 ± 2.9) (Barletta et al., 2002), Hong Kong (3.0 ± 0.4) (Huang et al. 2015), Okhla, New Delhi (2.3 ± 1.7) (Hoque et al. 2008), Antwerp, Belgium (3.5 ± 0.2) (Buczynska et al. 2009) and Nanjing, China (2.6 ± 0.9) (Wang et al. 2008). For the idling vehicular exhausts of different fuel types, this ratio varied between 0.38-10.90 and was 3.68 ± 0.58 for petrol vehicles which falls within the range reported in the previous work of Guo et al., 2011 (2.0-3.8). For the diesel vehicles, the T/B ratio in our study was 0.37 ± 0.20 which is similar to the average T/B ratio (0.37) from diesel vehicles in Australia (Anyon et al., 2003), Germany (0.56) (Siegl et al., 1999) and Tokyo (0.3) (Yamamoto et al., 2012)"*

**17. Page 20 Line 17: Similar to earlier comments, please show the absolute amounts of i-pentane and n-pentane in the source samples, to see how large or small the concentrations were and how variable the ratios were. From Figure 2 there seems to be very little n-pentane in the LPG exhaust, which could lead to a high and uncertain pentane ratio. All ratios should have error bars.**

**Author response:** The first part of the comment has already been addressed through the new excel file supplement as mentioned in previous replies. Reviewer is correct in surmising that about low n-pentane in the LPG vehicular exhaust. Additionally, as per reviewer's suggestion, the averaged molar ratios in Table 3 are now presented with standard errors of mean to represent the uncertainty associated with the ratios. For LPG exhaust the *i*-pentane/*n*-pentane molar ratio is 14.99±2.69.

The relevant text is modified as follows:

"Therefore, caution should be applied while using this ratio in complex emission environments where biomass burning, fossil fuel combustion and biogenic emission sources simultaneously occur in significant scale and strength to contribute to the chemical composition of ambient air. $i$-pentane/$n$-pentane can instead be used as a more reliable ratio for distinguishing biomass burning emissions (0.06-1.46) from the petrol dominated traffic/fossil fuel emissions (2.83-4.13)."

**18. Page 21 Line 3: Please clearly state which results were "very different" from the literature based on a statistical analysis. The uncertainties in your study are likely large but haven't been discussed or quantified. On Line 4 I disagree that these profiles can be used for accurate and reliable emissions estimates, since the sample size is so small. The study is a good beginning, but the results need realistic uncertainties.**

**Author response:** We regret that the results that were different from the literature were not clarified and we now present the results unique to this study.

These are now included in the Conclusion and replace the redundant text pointed out by the esteemed reviewer in point 20 below.

Some of the major findings which provide new insights are:

i.       Propane was found to be one of the abundant NMHC compounds in the paddy stubble fire emissions. This is in contrast to the existing literature which considers it as a tracer for fugitive LPG emissions. In a complex emission environment influenced by several sources like paddy fires, the use of propane as an LPG tracer only requires caution.

ii.      Propene emissions in smouldering fires were found to be more than ethene by ~1.6 times which is in contrast to the existing crop residue burning inventories which have ethene as the abundant specie.

iii.     Isoprene was identified as a reliable tracer to differentiate between the paddy fires and garbage fires at night.

iv.      It was also found that there were compositional differences in the evaporative emissions from the two types of LPG (commercial and domestic) used widely in South Asia. While, propane was the most dominant NMHC in the domestic LPG vapours, the commercial LPG vapours were dominated by butanes.

v.       Toluene/benzene ratios were found to be a good tracer for distinguishing the paddy stubble fire emissions in flaming (0.38 ± 0.11) and smouldering stages (1.40 ± 0.10), garbage burning emissions (0.26-0.59) and traffic emissions (3.54 ± 0.21).

vi.      $i$-butane/$n$-butane ratio was found to be similar (0.20-0.30) for many sources and therefore caution must be taken while using it in complex emission environments. $i$-pentane/$n$-pentane ratio instead turned out to be a better tracer for distinguishing biomass burning emissions (0.06-1.46) from the petrol dominated traffic/fossil fuel emissions (2.83-4.13).

Abstract of revised version has also been modified to include some of these points.

The uncertainties have now been incorporated in the revised version as per reviewers' suggestions. The issue of inadequate samples for evaporative emissions has also been addressed by increasing the numbers of samples by factor of 10.

Considering these changes, we hope now the statement is justified especially as now we also improve the discussion by listing limitations more clearly.

**19. Figure S4: Even though propene is more reactive than ethene, it's surprising to see so little ethene contribution to OH reactivity in smoldering paddy burning since its EF is typically higher than propene for crop residue fires. In Figure 1 I was surprised to see more propene than ethene in the normalised profiles for smoldering paddy burning, different from results for crop residue (rice straw) in Akagi et al. (ACP, 2011) and agricultural residue in Andreae (ACP, 2019). Please show the concentrations used in these calculations and expand the discussion.**

**Author response:** The reviewer is right that based upon the existing knowledge of paddy stubble burning, ethene has higher EF than propene as reported in (Akagi et al., 2011;Andreae, 2019). However, it should be noted that these EFs were mainly based on laboratory experiments and none of them measured in-situ within an agricultural field in India, where local conditions (e.g. wetness of ground) and biomass variability and environmental conditions can be significantly different. Our data is derived from the open burning of paddy stubble in the agricultural fields. We have found out that these previously reported EFs are more representative of the flaming conditions of crop residue fires. The propene emissions in smoldering fires (which have lower combustion efficiency) were more than ethene (~1.6 times) and when coupled to its higher reactivity than ethene (~4times), we saw higher contribution of propene than ethene in OH reactivity. We thank the reviewer for drawing attention to this point. The following text has now been added to expand the discussion:

Page 11 Line 19:

*"In the studies reported previously (Akagi et al., 2011; Andreae, 2019), ethene was reported to have higher emissions than propene from crop residue fires. Our study results reveal that ethene emissions were lower in the smouldering fires as compared to propene. While the previous studies have compiled results of mostly laboratory combustion of fuels in controlled environments and are more typical of flaming conditions, the smouldering stage of fire which are characterized by poor combustion efficiency and therefore different flame chemistry in the agricultural fields as encountered by us, may be a cause for this variance and emphasize why results from controlled burn experiments need to be complemented with field crop residue fire results."*

**20. Page 21 Lines 8-14: These sentences are very similar to the abstract. Page 22 Lines 9-12 and Line 13-15 also repeats from earlier text. The conclusions should provide fresh insights.**

**Author response:** The Conclusion section has been rewritten in the revised version and modified as also specified in reply to point 18.

**Please check the entire manuscript for grammar and typos. For example: Grammar (P4 L23, P6 L4-6, P7 L8-12, P8 L10-11, and so forth in the paper and supplement). Capitalization (P8, L1-8: synthetic, nitrogen, ozone; and so forth).**

**Author response:** We thank the reviewer to point out the grammar and typos in the text, all of which have been addressed in the manuscript now.

**Page 5 Line 16: Define BTEX.**

**Author response:** BTEX refers to sum of benzene, toluene, ethylbenzene and xylenes. The text in the manuscript has been modified accordingly.

Page 5 Line 14-16:

*"Further, we assessed the secondary pollutant formation potential and health risks of the sources in terms of their OH reactivity ($s^{-1}$), ozone formation potential (OFP, $gO_3/gNMHC$) and fractional BTEX (sum of benzene, toluene, ethylbenzene and xylenes) content."*

**Page 5 Line 19: What was the sample duration for the whole air samples?**

**Author response:** The sampling duration for each sample was roughly 15 min during which the whole air was actively collected in a 6L passivated SilcoCan air sampling steel canisters (Restek, USA) using Teflon VOC pump (Model − N86 KT.45.18; KNF Germany) operating at a flow rate of ~5500 ml/min. The canisters were protected from dust and air particles using a Teflon membrane filter (Pore size 0.45µm) in the sample inlet line. The details pertaining to the sampling methodology have been described in detail in previous works from our lab and cited in the manuscript already (Chandra et al., 2017;Vettikkat et al., 2019).

**Page 6 Line 18: So 23 vehicles, with one sample per vehicle?**

**Author response:** Yes, 23 idling vehicular exhaust samples were collected from petrol fuelled vehicles (14 two wheelers and 9 light duty four wheelers) with each vehicle sampled once.

**Page 9 Line 21: Please define the sensitivity factors and explain the results in Table S3. Avoid stating "with no drastic changes observed" and be specific about what the results mean.**

**Author response:** The FID signal of a compound is recorded in form of current (picoampere, pA) by the instrument. While generating the chromatogram the y-axis corresponds to the measured signal (peak height) in pA and x-axis corresponds to the retention time (in minutes). The area under the peak is calculated and expressed in units of pAs (picoampere seconds) and used to quantify the analyte. The instrument's response factor or sensitivity factor for a particular NMHC is expressed in picoamperes (pAs) signal of FID / per ppb of NMHC and is referred to as the sensitivity factor. This is determined from the calibration experiments (Figure S3) which yield the slope and also show that instrument response is linear so as to be able to apply the sensitivity factor over the relevant linear dynamic range of interest for the target compound. Between Dec 2016 and Oct 2018, thirteen calibrations were performed and the sensitivity factors derived from these calibrations were used to quantify the concentrations in the samples discussed in this work. In Table S3, we list the average sensitivity factors (pAs/ppb) and standard deviation derived from these calibration experiments and show that the changes in instrumental sensitivities for most of the reported compounds were approximately 8-12% (except acetylene (18%) and diethylbenzenes and trimethylbenzenes (14-16%)) and therefore the instrumental response was fairly stable.

We agree with the reviewer and have removed the phrase "with no drastic changes", and instead state based on the data presented in Table S3 of original submission and Table S4 of new submission that the sensitivity factors showed no major changes in sensitivity from Dec 2016 till Oct 2018. The manuscript has been modified as follows:

Page 9 Line 21

*"Table S4 shows the average sensitivity factors (pAs/ppb) and standard deviation derived from thirteen calibrations performed regularly between Dec 2016 and Oct 2018 with no major*

*changes (8-12% for most of the measured compounds) observed in the instrumental sensitivities."*

**Page 10 Line 1: What does respective refer to here?**

**Author response:** We are sorry for the confusion. Here we were comparing the results obtained from the calibration experiments for benzene, toluene and isoprene which are the common compounds in both the calibration gas standards used in this work. Considering the overall instrument uncertainty was <15%, there was good agreement between the two calibration standards for the common compounds.

The text has therefore been modified in the manuscript as follows:

Page 9 line 23 to Page 10 Line 4:

*"A reasonable agreement (considering the maximum instrumental uncertainty error of 15%) was found for the average calibration factors between Dec 2016 - Oct 2018 derived from the two different gas standards for the common compounds such as isoprene (53.2 ± 4.9 and 55.6 ± 5.9 pAs ppb$^{-1}$), benzene (67.8 ± 5.6 and 69.2 ± 5.5 pAs ppb$^{-1}$) and toluene (74.6 ± 6.6 and 81.3 ± 7.7 pAs ppb$^{-1}$)."*

**Page 10 Line 7: pAs was first used on Page 9 Line 21; define there.**

**Author response:** The following text has been modified according to suggestions of both the reviewers:

Page 9 line 8-23:

[revised manuscript text omitted]

Page 10 line 4-8: Removed

**Page 11 Line 14: Reduced compared to what? Flaming?**

**Author response:** Yes. The fraction of ethene and propene reduced in smouldering fires as compared to flaming while that of isoprene increased by ~3 times.

The same has now been clarified by revising the relevant text in the revised version.

**Page 11 Line 20: Less styrene compared to what?**

**Author response:** Compared to garbage samples containing plastic and packaging material as garbage burning can also be a source for styrene emissions due to the presence of plastic (polystyrene) waste (Lemieux et al., 2004;Tang et al., 2000). However, in our study we noticed less fraction of styrene (<1%), which was because of presence of wet vegetable and food waste from households in our garbage samples.

The same has now been clarified by revising the relevant text in the revised version.

**Page 12 Line 1: Why at night? I thought the daytime values subtracted off background?**

**Author response:** We are confused by this comment. We believe that the reviewer is under the impression that the biomass burning samples were collected as ambient plumes. It has already been discussed in detail and shown in Figure S1 that the whole air samples were collected in passivated steel canisters directly from the source, i.e., the burning paddy straw fields and garbage. To subtract the background ambient air, one sample each was also collected of the ambient air prior to lighting the fires. Even after background correction, high emissions of isoprene were observed in the paddy fires (5-13% w/w) as compared to garbage fires (1% w/w), which shows that paddy stubble burning is a stronger source of isoprene than garbage

burning. It is already known that isoprene is emitted in daytime from a vegetation, but during the night, no such biogenic sources are known to emit isoprene. Therefore, at night time, in the absence of biogenic emissions, high isoprene concentrations can be helpful in distinguishing the active sources especially between paddy fires, garbage fires and traffic plumes.

**Page 13 Line 18: Hong Kong also has an LPG fuel composition of n-butane > propane > i-butane, with about a 2:1 ratio of n-butane:i-butane, similar to your evaporative results (Tsai et al., ACP, 2006).**

**Author response:** We are thankful to reviewer for pointing this out. However in view of the new results, we find that the LPG fuel composition in Hong Kong may be different to that in India. Here, we see that in evaporative emissions propane>*n*-butane>*i*-butane for domestic LPG cylinders and *n*-butane>*i*-butane>propane for commercial LPG cylinders. The differences in our observations from the studies in Guangzhou (Lai et al., 2009), Taiwan (Chang et al., 2001) and Hong Kong (Guo et al., 2011), regarding the higher fraction of butanes as compared to propane in idling vehicular exhaust therefore could be because of different engine technology/efficiency and combustion conditions in addition to the fuel composition.

The same has now been clarified by revising the relevant text in the final version.

**Page 14 Line 1: Guo et al. studied Hong Kong, not Taiwan.**

**Author response:** We are extremely sorry for the inadvertent error and thank the reviewer for bringing it our attention. The text has been modified accordingly with the correct reference.

Page 14 Line 1:

*".....similar to the studies conducted in Taiwan (Chang et al., 2001) and Pearl river delta (Liu et al., 2008)."*

**Page 14 Line 19: i-Pentane is already known to be a gasoline tracer, but the wording here makes it seem like a novel result.**

**Author response:** We agree with the reviewer that i-pentane has been reported as a tracer of gasoline in previous studies too. We now refrain from reporting it as a novel result in the revised version.

New text reads as:

"Since *i*-pentane was found to be negligible (<0.5%) in diesel exhaust, it was identified as an ideal tracer for petrol vehicular emissions as has also been reported previously."

**Page 15: This paragraph is more than 2 pages long.**

**Author response:** We thank the reviewer for pointing out the long paragraph, which has now been split into two different paragraphs.

**Page 15 Line 9: Propane isn't listed on Line 6-7 as one of the major NMHC species. How much propane was measured from the traffic?**

**Author response:** Propane was the 17[th] most abundant compound with an overall mass fraction of ~2% in the traffic samples and hence not listed.

**Page 16 Line 6: Define BSV and BSVI.**

**Author response:** BSV and BSVI stand for Bharat Stage V and Bharat Stage VI. In order to tackle the problem of increasing vehicular pollution and bring the Indian motor vehicle

regulations in alignment with European Union regulations, the Indian government decided to apply BSVI standards w.e.f. April 2020. These are the new emission norms set up by the Government of India (GoI, 2016) and the fuels used therein. BS VI proposal clearly specifies the emission standards and commercial fuel specifications for each vehicle category and sub-classes (ARAI, 2018).

Full forms of BS have been now been added.

**Page 17 Lines 6-12: These results are better presented as a Table. Similar comment on Page 18**

**Author response:** We have provided a new Table S5 listing the results of the statistically significant differences based on Tukey's test for OFP. Figure 3 now has error bars for uncertainty and the text has also been modified in response to previous comments.

**Figures 1, 2, S4, S5: State what the shading refers to (aromatic, alkene/alkyne, alkane). In Figure S4 the shading is shifted by one in the alkenes – please correct.**

**Author response:** The compounds are shaded as follows:

Grey: Aromatics

Red: Alkenes + Alkyne

Yellow: Alkanes

The information regarding this shading is now added to the Figure description of Figure 1, 2 and S4,S5. As follows:

*"Figure 1: Normalised source profiles of a) Paddy stubble burning: Flaming; b) Paddy stubble burning: Smouldering; c) Garbage burning: Flaming; d) Garbage burning: Smouldering; e) LPG evaporative emissions; f) Petrol evaporative emissions; g) Diesel evaporative emissions, derived from the TD-GC-FID measurements. The grey colour highlights the aromatics, red colour highlights the alkenes and alkyne and the yellow colour highlights the alkanes."*

*"Figure 2: Normalised source profiles of a) CNG vehicular exhaust; b) LPG vehicular exhaust; c) Petrol vehicular exhaust; d) Diesel vehicular exhaust; e) Traffic, derived from the TD-GC-FID measurements. The grey colour highlights the aromatics, red colour highlights the alkenes and alkyne and the yellow colour highlights the alkanes."*

*"Figure S3: Normalised profiles of calculated OH reactivity (s-1) in a) Paddy stubble burning: Flaming; b) Paddy stubble burning: Smouldering; c) Garbage burning: Flaming; d) Garbage burning: Smouldering; e) Commercial LPG evaporative emissions; f) Domestic LPG evaporative emissions; g) Petrol evaporative emissions; h) Diesel evaporative emissions. The grey colour highlights the aromatics, red colour highlights the alkenes and alkyne and the yellow colour highlights the alkanes. The OH reactivities of the NMHCs are normalised to the NMHC with the maximum OH reactivity in the respective source sample as:*

$$f = [Yi]/[Ymax]$$

*Where, [Yi] is the NMHC OH reactivity and [Ymax] is the NMHC with the maximum OH reactivity in the respective source sample.*

*"Figure S4: Normalised profiles of calculated OH reactivity (s-1) in a) CNG vehicular exhaust; b) LPG vehicular exhaust; c) Petrol two wheeler exhaust; d) Petrol four wheeler exhaust; e) Diesel three wheeler vehicular exhaust; f) Diesel four wheeler vehicular exhaust*

*g) Diesel heavy duty vehicle (HDV) exhaust h) Traffic, derived from the TD-GC-FID measurements. The grey colour highlights the aromatics, red colour highlights the alkenes and alkyne and the yellow colour highlights the alkanes."*

**Figure S1: "Smouldering" here but "smoldering" in the main text.**

**Author response:** We thank the reviewer for pointing out the mistake in the inconsistent spelling of smouldering and the Figure S1 caption is hereby corrected as follows:

*"Figure S1: Whole air sample collection from a) paddy stubble fire: flaming; b) paddy stubble fire: smouldering; c) garbage fire: flaming; and d) garbage fire: smouldering. The flaming and smouldering fires were distinguished based upon the presence of flame and white smoke as per past experiences (Chandra et al 2017, Kumar et al., 2018)."*

*We use smouldering consistently now throughout in the revised MS.*

**Figure S2: The graphs are too small to clearly see.**

**Author response:** Based upon the suggestions of both the reviewers, we have modified the previous Figure S2 (see below for perusal). Which now shows that the calibration of the instrument was performed over a dynamic range of 2-200ppb over two sets of calibrations: regular calibration of 2-20 ppb and a high mixing ratio calibration of 10-200 pbb. This covers a range of two orders of magnitude over which the instrument exhibited an excellent linearity ($r^2$>0.99) for all the 49 NMHCs. In order to make it more visible, out of all the 49 NMHCs, only 24 major NMHCs determining the normalised profiles of the emission sources studied in this work were used to reconstruct the figure.

[Figure]

**"Figure S2:** *Sensitivity and linearity of NMHCs obtained from the calibration experiments performed over a dynamic range of 2-200ppb over two sets of calibrations: regular calibration of 2-20 ppb and a high mixing ratio calibration of 10-200 ppb. This covers a range of two orders of magnitude over which the instrument exhibited an excellent linearity ($r^2 > 0.99$) for all the 49 NMHCs(TD-GC-FID). The calibrations were performed via dynamic dilution with zero air using a standard gas calibration unit (GCU-s v2.1, Ionimed Analytik, Innsbruck, Austria). The horizontal error bars represent the root mean square propagation of errors due 5 % uncertainty in the VOC standard and 2% error for each of the two mass flow controllers used for calibration. The vertical error bars represent the uncertainty in instrumental measurements while sampling the standard gas at each dilution mixing ratio."*

**Figure S3: There is no Figure S3, just S2 and S4 – please re-number.**

**Author response:** We are sorry for the mistake and thank the reviewer for bringing it our attention. The figure numbers are re-numbered correctly in the supplement now. In the main manuscript too, the text has been modified accordingly:

Page 17 Line 19-20:

*"In paddy stubble fires under flaming conditions propene (33%), and under smoldering conditions isoprene (46%) were the largest contributors to the total OH reactivity (details in Figure S5 and S6).*

**Table 1: Please add the descriptions and sample sizes for the three evaporative fuel sources.**

**Author response:** The details pertaining to the fuel evaporative sampling for the additional samples as well have now been added to Table 1.

**Table 2: Please put the compounds in a more logical order so they're easier to find.**

**Author response:** Heeding the reviewer's suggestion the compounds are now arranged according to functional groups (aromatics, alkyne, alkenes and alkanes) and increasing molecular weights within each class, in the tables and revised figures.

**Table 2: The abstract states that 49 NMHCs were measured, but Table 2 only lists 48. Was styrene double-counted as both aromatic and alkene? Same comment in the introduction and conclusions.**

**Author response:** We are sorry for the mistake and thank the reviewer for bringing it to our attention. 1-pentene was the compound that was missing from Table 2 and has now been added. Styrene was considered as a single compound in aromatics category. The total number of quantified NMHCs were 49 as mentioned in abstract and elsewhere in the main text.

**Response to Anonymous referee #2**

Please find the point wise replies (**in blue**) to the referee's comments (**in black**) for easy perusal.

**1 General comments:**

Kumar et al. present measurement of 49 NMHCs using GC-FID from samples collected at difference sources (paddy stubble burning, garbage burning, idling vehicular exhaust and evaporative fuel emissions) in northern India. Normalised profiles were calculated based on the measured NMHCs for different sources. The authors identified i-pentane as a chemical tracer for petrol vehicular exhaust and evaporative emissions, propane as a chemical tracer for LPG evaporative and LPG vehicular exhaust emissions, and acetylene as a chemical tracer for the biomass fires in flaming conditions.

Instrument analysis is adequate. However the authors need to provide standard gas calibration data for compounds with higher concentrations (> 50 ppbv) to show that the instrument linearity is within tolerance at high mixing ratio level. The sample size for many sources are small (3 or 5 samples), which could introduce large variability and potentially undermine the data quality.

Overall, this study reports the source profiles of NMHCs over an understudied area of the world with complex emission sources. The data should be of interest to the atmospheric science community. This manuscript is within the scope of ACP. I recommend that the manuscript be published in ACP after minor revision.

**Author response**: We sincerely thank the reviewer for his/her insightful assessment of our manuscript and deeming it appropriate for ACP following the suggested minor revisions. Detailed response to each comment and changes made in the manuscript are listed below.

**2 Minor comments:**

**1. "South Asia" in the title covers a broad area. Please revise the title to reflect the specific sampling area (Mohali, India)**

**Author response:** South Asia was included in the title "Non methane hydrocarbon (NMHC) fingerprints of major urban and agricultural emission sources active in South Asia for use in source apportionment studies" to highlight that paddy stubble burning, open waste burning, vehicular exhaust emissions and fossil fuel evaporative emissions are commonly occurring emission sources in the region of South Asia (Chandra and Sinha, 2016; Gadde et al., 2009; Liu et al., 2008; Mo et al., 2016; Sharma et al., 2019; Streets et al., 2003). Also, for better representation, the samples of evaporative emissions have now been collected from the most common brands of petrol, diesel and LPG fuels sold in India, Nepal, Bangladesh and Sri Lanka (Indian oil, Hindustan Petroleum, Bharat Petroleum, Bharatgas and Indane). However, upon reading both the reviewers' advice, we have removed "active in South Asia" from the title of the revised version. This way the NMHC fingerprints can be used for source apportionment studies wherever it is relevant.

The revised title now reads as follows:

"Non methane hydrocarbon (NMHC) fingerprints of major urban and agricultural emission sources for use in source apportionment studies"

**2. Section 2.2: Please provide a schematic diagram of the instrument setup.**

**Author response:** We appreciate reviewer's query about the schematic setup of the instrument. We have now added a schematic to the supplement as Figure S2 (also shown below for ease).

[Figure]

**New Figure S2:** Schematic representation of the TD-GC-FID instrument during a typical sample injection and chromatographic run.

The following revision is added to the main manuscript:

Page 8 Line 6 of original submission:

*"Figure S2 shows the schematic representation of the TD-GC-FID instrument during a typical sample injection and chromatographic run…."*

**(3) Section 2.2: in peak identification and quantification section, there is no discussion on the peak separation. Are all the target compound peaks well separated? If not, how do you resolve the interference? Please provide a typical chromatogram showing all the target compounds taken during a standard gas calibration experiment. Please also include a typical chromatogram taken during the analysis of a sample collected from each source.**

**Author response:** We regret that the details regarding the peak separation were not mentioned in our initial submission. In response to reviewer's query about the chromatographic separation, a typical chromatogram of standard gas calibration is presented as Figure S3, and the representative chromatograms of the samples analysed are presented in additional supplementary pdf file. The peaks were well resolved and separated and were identified using compounds in the calibration gas standards. In case a shoulder peak was present then the parent peak was separately integrated, i.e., any interference from shoulder peak was subtracted from the parent signal. In the calibration gas standard some additional compounds were also present namely, 2,2,4-trimethylpentane, 2,3,4-trimethylpentane and methylcyclohexane, each of which had a well resolved and separate peak. However, during the analysis of some point source samples, these compounds exhibited poor peak features like peak shape, shoulder peaks, etc. Therefore, for remaining consistent across all samples, these compounds were excluded from analysis and only those compounds were included which were well resolved and characterised.

This has now been clarified in the revised text of the manuscript as following:

Page 9 line 8-23 of original submission:

[revised manuscript text omitted]

Page 10 line 4-8: Removed

**4. Page 9, Line 9: in Figure S2, most compounds do show good linear association. However, certain compounds, such as m-Diethylbenzene, p-Diethylbenzene, exhibit larger uncertainties at about 20 ppbv mixing ratio level and larger deviations from the fitted line compared to the rest of the compounds. Please provide correlation coefficient values (with 4 significant figures) for all the target compounds in Figure S2.**

**Author response:** Table R1 below lists the correlation coefficient values (with 4 digits) for all the 49 target compounds derived from the calibration experiments performed between Dec 2016 and Oct 2018. The reviewer is correct to identify that certain compounds like m-Diethylbenzene ($r^2 = 0.9769$) and p-Diethylbenzene ($r^2 = 0.9651$) exhibit larger deviation from the linear fitted calibration line. However, we note that none of these compounds is major contributor to the observed NMHC fingerprints for the sources or ozone formation or reactivity. We note that for the findings of this study their linearity being little less than others has no major consequence on the results and conclusions reported in this work. During seven of the thirteen calibration experiments, both these compounds did exhibit excellent linearity ($r^2 = 0.9920 \pm 0.002$ and $0.9888 \pm 0.004$ respectively). The variability in the linear response for higher compounds like m-Diethylbenzene and p-Diethylbenzene therefore could be related to mixing and introduction in the calibration standard during those 5 experiments rather than instrument performance. In any case above 0.9 is sufficiently good linearity for quantification in any instrument in our opinion.

**Table R1:** Correlation coefficient values (with 4 significant figures) for 49 NMHCs derived from the calibration experiments

| Compounds | 19-Dec-16 | 06-Mar-17 | 15-Jun-17 | 02-Sep-17 | 25-Sep-17 | 30-Oct-17 | 18-Jan-18 | 26-Feb-18 | 26-Mar-18 | 30-Apr-18 | 21-May-18 | 09-Jul-18 | 27-Oct-18 |
|---|---|---|---|---|---|---|---|---|---|---|---|---|---|
| | | | | | **Aromatics** | | | | | | | | |
| Benzene | 0.9998 | 0.9997 | 0.9999 | 0.9993 | 0.9953 | 0.9993 | 0.9962 | 0.9998 | 0.9993 | 0.9976 | 0.9997 | 0.9998 | 0.9995 |
| Toluene | 0.9992 | 1.0000 | 1.0000 | 0.9978 | 0.9967 | 0.9987 | 0.9956 | 0.9999 | 0.9989 | 0.9934 | 0.9989 | 0.9997 | 0.9923 |
| Styrene | 0.9971 | 0.9995 | 0.9994 | 0.9913 | 0.9988 | 0.9984 | 0.9953 | 0.9995 | 0.9970 | 0.9869 | 0.9974 | 0.9983 | 0.9947 |
| *m/p*-Xylene | 0.9967 | 0.9994 | 0.9994 | 0.9929 | 0.9989 | 0.9979 | 0.9958 | 0.9993 | 0.9969 | 0.9876 | 0.9973 | 0.9987 | 0.9941 |
| *o*-Xylene | 1.0000 | 0.9995 | 0.9994 | 0.9917 | 0.9993 | 0.9979 | 0.9949 | 0.9995 | 0.9980 | 0.9852 | 0.9976 | 0.9983 | 0.9943 |
| Ethylbenzene | 0.9982 | 0.9996 | 0.9986 | 0.9944 | 0.9989 | 0.9988 | 0.9961 | 0.9998 | 0.9981 | 0.9893 | 0.9979 | 0.9990 | 0.9945 |
| *m*-Ethyltoluene | 0.9919 | 0.9972 | 0.9982 | 0.9851 | 0.9941 | 0.9967 | 0.9951 | 0.9977 | 0.9916 | 0.9788 | 0.9950 | 0.9945 | 0.9940 |
| *o*-Ethyltoluene | 0.9913 | 0.9967 | 0.9980 | 0.9831 | 0.9931 | 0.9962 | 0.9950 | 0.9975 | 0.9919 | 0.9805 | 0.9958 | 0.9947 | 0.9882 |
| *p*-Ethyltoluene | 0.9897 | 0.9954 | 0.9973 | 0.9833 | 0.9935 | 0.9948 | 0.9947 | 0.9963 | 0.9903 | 0.9771 | 0.9944 | 0.9936 | 0.9955 |
| 1,2,3-Trimethylbenzene | 0.9908 | 0.9958 | 0.9976 | 0.9818 | 0.9930 | 0.9957 | 0.9947 | 0.9969 | 0.9906 | 0.9779 | 0.9948 | 0.9937 | 0.9953 |
| 1,2,4-Trimethylbenzene | 1.0000 | 0.9999 | 1.0000 | 0.9997 | 0.9995 | 0.9997 | 0.9975 | 0.9999 | 0.9997 | 0.9991 | 0.9998 | 0.9999 | 0.9985 |
| 1,3,5-Trimethylbenzene | 1.0000 | 0.9999 | 1.0000 | 0.9994 | 0.9989 | 0.9996 | 0.9967 | 0.9998 | 0.9996 | 0.9976 | 0.9997 | 0.9999 | 0.9975 |
| *i*-Propylbenzene | 0.9999 | 0.9998 | 0.9998 | 0.9997 | 0.9977 | 0.9995 | 0.9962 | 0.9998 | 0.9996 | 0.9983 | 0.9997 | 1.0000 | 0.9984 |
| *n*-Propylbenzene | 0.9942 | 0.9981 | 0.9986 | 0.9870 | 0.9957 | 0.9974 | 0.9952 | 0.9982 | 0.9936 | 0.9812 | 0.9959 | 0.9960 | 0.9945 |
| *m*-Diethylbenzene | 0.9769 | 0.9834 | 0.9925 | 0.9707 | 0.9886 | 0.9927 | 0.9933 | 0.9915 | 0.9808 | 0.9707 | 0.9908 | 0.9847 | 0.9949 |
| *p*-Diethylbenzene | 0.9651 | 0.9734 | 0.9865 | 0.9643 | 0.9863 | 0.9865 | 0.9916 | 0.9870 | 0.9740 | 0.9644 | 0.9871 | 0.9793 | 0.9969 |
| | | | | | **Alkyne** | | | | | | | | |
| Acetylene | 0.9998 | 0.9934 | 0.9992 | 0.9983 | 0.9924 | 0.9933 | 0.9987 | 0.9999 | 0.9998 | 0.9996 | 0.9995 | 0.9998 | 0.9999 |
| | | | | | **Alkenes** | | | | | | | | |
| Ethene | 1.0000 | 0.9783 | 0.9999 | 0.9994 | 0.9953 | 0.9820 | 1.000 | 1.0000 | 0.9997 | 0.9998 | 0.9994 | 0.9999 | 0.9994 |
| Propene | 1.0000 | 1.0000 | 1.0000 | 1.0000 | 0.9999 | 0.9999 | 0.9976 | 0.9999 | 0.9996 | 0.9990 | 0.9998 | 1.0000 | 0.9999 |
| 1-Butene | 0.9999 | 1.0000 | 1.0000 | 0.9998 | 1.0000 | 0.9994 | 0.9972 | 0.9998 | 0.9996 | 0.9987 | 0.9999 | 1.0000 | 1.0000 |
| *trans*-2-Butene | 0.9995 | 0.9998 | 0.9999 | 0.9998 | 0.9998 | 0.9994 | 0.9970 | 0.9997 | 0.9996 | 0.9988 | 0.9999 | 1.0000 | 1.0000 |
| *cis*-2-Butene | 1.0000 | 0.9999 | 0.9996 | 0.9997 | 1.0000 | 0.9996 | 0.9971 | 0.9998 | 0.9995 | 0.9987 | 0.9999 | 1.0000 | 1.0000 |

| | | | | | | | | | | | | | |
|---|---|---|---|---|---|---|---|---|---|---|---|---|---|
| Isoprene | 0.9994 | 0.9992 | 0.9998 | 0.9980 | 1.0000 | 0.9995 | 0.9970 | 0.9998 | 0.9996 | 0.9995 | 0.9998 | 1.0000 | 0.9998 |
| 1-Pentene | 1.0000 | 1.0000 | 0.9999 | 0.9997 | 1.0000 | 0.9999 | 0.9973 | 0.9999 | 0.9996 | 0.9991 | 0.9998 | 0.9998 | 1.0000 |
| *trans*-2-Pentene | 1.0000 | 0.9999 | 1.0000 | 0.9999 | 1.0000 | 0.9999 | 0.9971 | 0.9999 | 0.9996 | 0.9992 | 0.9997 | 1.0000 | 1.0000 |
| *cis*-2-Pentene | 1.0000 | 0.9999 | 1.0000 | 0.9998 | 1.0000 | 0.9999 | 0.9973 | 0.9999 | 0.9996 | 0.9992 | 0.9997 | 0.9998 | 1.0000 |
| 1-Hexene | 1.0000 | 0.9999 | 1.0000 | 0.9997 | 0.9999 | 0.9999 | 0.9972 | 0.9999 | 0.9994 | 0.9986 | 0.9997 | 1.0000 | 0.9998 |
| **Alkanes** | | | | | | | | | | | | | |
| Ethane | 0.9964 | 0.9967 | 0.9948 | 0.9990 | 0.9991 | 0.9999 | 0.9998 | 0.9999 | 0.9993 | 0.9998 | 0.9996 | 0.9999 | 0.9996 |
| Propane | 0.9999 | 0.9999 | 1.0000 | 1.0000 | 0.9999 | 1.0000 | 0.9968 | 0.9999 | 0.9995 | 0.9992 | 0.9998 | 0.9999 | 0.9999 |
| *n*-Butane | 1.0000 | 1.0000 | 0.9999 | 0.9999 | 0.9999 | 0.9998 | 0.9968 | 0.9998 | 0.9995 | 0.9991 | 0.9998 | 0.9999 | 0.9999 |
| *i*-Butane | 1.0000 | 0.9999 | 0.9989 | 0.9999 | 0.9999 | 0.9998 | 0.9970 | 0.9999 | 0.9997 | 0.9992 | 0.9998 | 1.0000 | 0.9995 |
| *i*-Pentane | 0.9960 | 0.9929 | 0.9998 | 0.9994 | 1.0000 | 0.9999 | 0.9971 | 0.9998 | 0.9997 | 0.9993 | 0.9998 | 1.0000 | 1.0000 |
| *n*-Pentane | 1.0000 | 0.9999 | 1.0000 | 0.9999 | 1.0000 | 0.9999 | 0.9973 | 0.9999 | 0.9995 | 0.9990 | 0.9997 | 1.0000 | 0.9999 |
| Cyclopentane | 0.9989 | 0.9987 | 0.9942 | 0.9993 | 0.9978 | 0.9997 | 0.9961 | 0.9994 | 0.9997 | 0.9997 | 0.9999 | 0.9996 | 0.9993 |
| Cyclohexane | 0.9999 | 1.0000 | 0.9997 | 0.9997 | 0.9985 | 0.9997 | 0.9965 | 0.9998 | 0.9991 | 0.9985 | 0.9997 | 1.0000 | 0.9996 |
| Methylcyclopentane | 1.0000 | 0.9999 | 0.9999 | 0.9998 | 0.9993 | 0.9998 | 0.9969 | 0.9999 | 0.9996 | 0.9988 | 0.9999 | 1.0000 | 0.9993 |
| 2,2-Dimethylbutane | 0.9999 | 0.9999 | 1.0000 | 0.9999 | 1.0000 | 0.9999 | 0.9973 | 0.9999 | 0.9996 | 0.9991 | 0.9998 | 1.0000 | 0.9999 |
| 2,3-Dimethylbutane | 1.0000 | 0.9999 | 1.0000 | 0.9999 | 0.9999 | 0.9999 | 0.9973 | 0.9999 | 0.9995 | 0.9990 | 0.9997 | 1.0000 | 1.0000 |
| *n*-Hexane | 1.0000 | 1.0000 | 1.0000 | 0.9998 | 0.9998 | 0.9998 | 0.9988 | 0.9993 | 0.9994 | 0.9982 | 0.9993 | 0.9998 | 0.9991 |
| 2-Methylpentane | 1.0000 | 0.9998 | 1.0000 | 0.9998 | 0.9998 | 0.9999 | 0.9980 | 0.9996 | 0.9992 | 0.9981 | 0.9994 | 1.0000 | 0.9999 |
| 3-Methylpentane | 1.0000 | 0.9999 | 0.9998 | 0.9997 | 0.9998 | 0.9998 | 0.9963 | 0.9999 | 0.9997 | 0.9993 | 0.9998 | 0.9998 | 1.0000 |
| 2-Methylhexane | 1.0000 | 1.0000 | 0.9999 | 0.9995 | 0.9996 | 0.9996 | 0.9969 | 0.9997 | 0.9994 | 0.9980 | 0.9996 | 0.9999 | 0.9981 |
| 3-Methylhexane | 1.0000 | 0.9999 | 1.0000 | 0.9996 | 0.9995 | 0.9997 | 0.9976 | 0.9999 | 0.9995 | 0.9982 | 0.9995 | 0.9998 | 0.9985 |
| 2,3-Dimethylpentane | 1.0000 | 0.9999 | 0.9999 | 0.9995 | 0.9994 | 0.9997 | 0.9967 | 0.9977 | 0.9995 | 0.9986 | 0.9997 | 1.0000 | 0.9986 |
| 2,4-Dimethylpentane | 1.0000 | 0.9998 | 0.9999 | 0.9997 | 0.9998 | 0.9999 | 0.9969 | 0.9998 | 0.9996 | 0.9988 | 0.9998 | 1.0000 | 0.9967 |
| *n*-Heptane | 0.9981 | 0.9999 | 0.9993 | 0.9925 | 0.9989 | 0.9986 | 0.9955 | 0.9996 | 0.9977 | 0.9881 | 0.9976 | 0.9987 | 0.9970 |
| *n*-Octane | 1.0000 | 0.9996 | 0.9999 | 0.9975 | 0.9989 | 0.9991 | 0.9958 | 0.9998 | 0.9989 | 0.9925 | 0.9993 | 0.9997 | 0.9956 |
| 2-Methylheptane | 1.0000 | 0.9998 | 0.9999 | 0.9987 | 0.9993 | 0.9994 | 0.9963 | 0.9999 | 0.9995 | 0.9961 | 0.9996 | 0.9999 | 0.9968 |
| 3-Methylheptane | 0.9999 | 0.9993 | 0.9998 | 0.9985 | 0.9992 | 0.9994 | 0.9965 | 0.9999 | 0.9995 | 0.9962 | 0.9995 | 0.9995 | 0.9924 |

**5. Page 9, Line 12: please list all compounds with concentrations > 50 ppbv after dilution.**

**Author response:** As per both the esteemed reviewers' suggestions, an excel sheet has been included as additional supplementary material that provides the mixing ratios of the NMHCs after dilution for each source.

In addition Table S2 of the revised main supplement lists the compounds with mixing ratios > 50ppbv in different emission sources after the dilution during sample analysis in the TD-GC-FID.

Table S2: Average concentrations (ppb) and standard deviation (in parentheses) of the compounds >50ppb after dilution

| Petrol 2 Wheelers (n = 14) | | Petrol 4 Wheelers (n = 9) | | Diesel HDVs (n = 15) | |
|---|---|---|---|---|---|
| Compounds | diluted concentration | Compounds | diluted concentration | Compounds | diluted concentration |
| Toluene | 96.3 (45.7) | Ethane | 81.8 (62.28) | Ethene | 150.4 (35.16) |
| *m/p*-Xylene | 55.1 (32.2) | *i*-Pentane | 72.9 (71.55) | Propene | 59.5 (14.26) |
| Acetylene | 95.6 (74.7) | | | | |
| Ethene | 67.5 (48.9) | | | | |
| *i*-Pentane | 87.2 (56.3) | | | | |
| Toluene | 96.3 (45.8) | | | | |
| *m/p*-Xylene | 55.1 (32.2) | | | | |
| **Diesel 4 Wheelers (n = 12)** | | **Diesel 3 Wheelers (n = 7)** | | **LPG Vehicles (n = 9)** | |
| Acetylene | 58.9 (37.9) | Acetylene | 51.7 (25.5) | Propene | 107.5 (75.6) |
| Ethene | 164.6 (27.2) | Ethene | 78.1 (18.3) | *trans*-2-Butene | 71.8 (35.2) |
| Propene | 53.2 (21.8) | | | 1-Butene | 55.2 (40.8) |
| | | | | Propane | 105.5 (57.6) |
| | | | | *n*-Butane | 132.8 (43.0) |
| | | | | *i*-Butane | 93.6 (37.7) |
| | | | | Propene | 107.5 (75.6) |
| **CNG Vehicles (n=7)** | | **Paddy fires Flaming (n=3)** | | **Paddy fires Smoldering (n=3)** | |
| Ethane | 166.1 (17.8) | Ethene | 65.4 (23.5) | Ethane | 132.6 (97.67) |
| **Garbage fires Flaming (n=5)** | | **Garbage fires Smouldering (n=5)** | | **Traffic (n=3)** | |
| Acetylene | 76.9 (56.6) | Ethane | 59.2 (65.8) | None | |
| Ethene | 83.8 (54.7) | | | | |
| Propene | 61.2 (35.3) | | | | |
| **Domestic LPG evaporative (n=5)** | | **Commercial LPG evaporative (n=5)** | | **Petrol evaporative (n=10)** | **Diesel evaporative (n=10)** |
| Propane | 147.4 (48.7) | *trans*-2-Butene | 52.5 (20.6) | None | None |
| *n*-Butane | 55.1 (22.5) | *n*-Butane | 127.9 (38.9) | | |
| *i*-Butane | 53.6 (21.2) | *i*-Butane | 62.6 (22.4) | | |

**6. Page 9, Line 13–14: please provide data (similar to Figure S2) to show the standard gas calibration results for the target compounds with concentrations of up to 200 ppbv. Please also include correlation coefficient values (with 4 significant figures) for all the target compounds.**

**Author response:** Based upon the suggestions of both the reviewers, we have modified the FigureS2 which now shows that the calibration of the instrument was performed over a dynamic range of 2-200ppb over two sets of calibrations: regular calibration of 2-20 ppb and a high mixing ratio calibration of 10-200 pbb. This covers a range of two orders of magnitude over which the instrument exhibited an excellent linearity ($r^2$>0.99) for all the 49 NMHCs. In order to make it more visible, out of all the 49 NMHCs, only 24 major NMHCs determining the normalised profiles of the emission sources studied in this work were used to reconstruct the figure. It can be clearly seen that excellent linearity ($r^2 > 0.99$) was observed for all the targeted compounds during the high concentration calibration experiment. Table R3 lists the correlation values (with 4 significant figures) for all the targeted compounds during the high range calibration.

**Table R3:** Correlation coefficient values (with 4 significant figures) for 49 NMHCs derived from calibration experiment of high concentrations (10-200 ppb)

| Compounds | correlation coefficient values ($r^2$) | Compounds | correlation coefficient values ($r^2$) |
|---|---|---|---|
| Benzene | 0.9998 | *p*-Ethyltoluene | 0.9994 |
| Toluene | 0.9996 | 1,2,3-Trimethylbenzene | 0.9985 |
| Styrene | 1.0000 | 1,2,4-Trimethylbenzene | 0.9995 |
| *m/p*-Xylene | 1.0000 | 1,3,5-Trimethylbenzene | 0.9993 |
| *o*-Xylene | 0.9997 | *i*-Propylbenzene | 1.0000 |
| Ethylbenzene | 1.0000 | *n*-Propylbenzene | 0.9996 |
| *m*-Ethyltoluene | 0.9984 | *m*-Diethylbenzene | 0.9982 |
| *o*-Ethyltoluene | 0.9995 | *p*-Diethylbenzene | 0.9967 |
| Acetylene | 0.9996 | | |
| Ethene | 0.9997 | Isoprene | 0.9919 |
| Propene | 0.9997 | 1-Pentene | 0.9971 |
| 1-Butene | 0.9997 | *trans*-2-Pentene | 0.9945 |
| *trans*-2-Butene | 0.9993 | *cis*-2-Pentene | 0.9949 |
| *cis*-2-Butene | 0.9994 | 1-Hexene | 0.9998 |
| Ethane | 0.9996 | *n*-Hexane | 0.9996 |
| Propane | 0.9995 | 2-Methylpentane | 0.9976 |
| *n*-Butane | 0.9963 | 3-Methylpentane | 0.9978 |
| *i*-Butane | 0.9977 | 2-Methylhexane | 1.0000 |
| *i*-Pentane | 0.9976 | 3-Methylhexane | 0.9998 |
| *n*-Pentane | 0.9980 | 2,3-Dimethylpentane | 0.9981 |
| Cyclopentane | 0.9923 | 2,4-Dimethylpentane | 0.9997 |
| Cyclohexane | 0.9975 | *n*-Heptane | 0.9998 |
| Methylcyclopentane | 0.9988 | *n*-Octane | 1.0000 |
| 2,2-Dimethylbutane | 0.9998 | 2-Methylheptane | 0.9999 |
| 2,3-Dimethylbutane | 0.9979 | 3-Methylheptane | 0.9999 |

[Figure]

**"Figure S2:** *Sensitivity and linearity of NMHCs obtained from the calibration experiments performed over a dynamic range of 2-200ppb over two sets of calibrations: regular calibration of 2-20 ppb and a high mixing ratio calibration of 10-200 ppb. This covers a range of two orders of magnitude over which the instrument exhibited an excellent linearity ($r^2 > 0.99$) for all the 49 NMHCs(TD-GC-FID). The calibrations were performed via dynamic dilution with zero air using a standard gas calibration unit (GCU-s v2.1, Ionimed Analytik, Innsbruck, Austria). The horizontal error bars represent the root mean square propagation of errors due 5 % uncertainty in the VOC standard and 2% error for each of the two mass flow controllers used for calibration. The vertical error bars represent the uncertainty in instrumental measurements while sampling the standard gas at each dilution mixing ratio."*

**7. Page 11, Line 5–7: please provide data to show a comparison of the target compound mixing ratios between before the fire and during the fire. Are the mixing ratios taken just before the fire (deemed as the ambient background level) significantly lower than during the fire? If not, does this bring large uncertainty to the interpretation of the calculated emission profiles?**

**Author response:** This has already been addressed in reply to similar point raised by reviewer 1. It is to be noted that the measured concentrations for NMHCs that determine the source profile (normalised ratio > 0.2) were higher than the ambient background by an order of two or more. The compounds that were found to be lower than the background were considered as non-emitted, i.e, 0 ppb and such compounds had a negligible contribution (normalised ratio < 0.05) to the normalised source profile. For example, in the paddy flaming fire sample 1, *m/p*-xylene, *o*-ethyltoluene, *i*-propylbenzene, 1,2,4-trimethylbenzene, *i*-butane, *i*-pentane, cyclohexane, 2,2,-dimethylbutane, 2,3-dimethylbutane and 2,3-dimethylpentane were lower than the ambient background concentrations and therefore were considered non-emitted (0ppb). In every emission source sample, the number of such compounds were usually less than 10. Hence the background concentrations and their correction have negligible role in determining the source NMHC fingerprints. This can easily be seen in the excel sheet provided with the response that shows the background and actual sample concentrations for each emission source sample as desired by the reviewer.

**8. Page 17, Line 6–12: there is no need to list all the rankings here since the reader can get this information from Figure 3.**

**Author response:** We thank the reviewer for their suggestion. Since similar points were also raised by the other reviewer, we therefore performed the statistical tests to see if the OFP from the sources are significantly different or not and changes made have been listed in reply to Reviwer1's similar comment.

**9. Page 18, Line 10: it would be more informative to provide the rate coefficient value range for reactions between C2-C4, C5-C8 alkanes and OH here for comparison purpose instead of just saying "more reactive towards OH radical".**

**Author response:** We appreciate the reviewer's suggestion to incorporate the rate coefficient values of the reactions of C2-C4 and C5-C8 alkanes with OH radical while comparing them. The rate coefficient values of C2-C4 alkanes varies between (0.25-2.12) x $10^{-12}$ $cm^3$ molecule$^{-1}$ s$^{-1}$ at 298K, while of C5-C8 alkanes between (3.6-8.9) x $10^{-12}$ $cm^3$ molecule$^{-1}$ s$^{-1}$ at 298K. (Atkinson, 1997; Atkinson et al., 1989; Atkinson et al., 2006). In general, the branched C5-C8 alkanes are roughly 2-6 times more reactive than the C2-C4 alkanes and therefore resulted in more contribution of alkanes tot the total OH reactivity in diesel evaporative emissions.

The text has been modified accordingly:

Page 18 line 9-10:

*"This is because of the presence of larger fraction of heavier C5-C8 branched alkanes which are roughly 2-6 times more reactive towards OH radical as compared to the light C2-C4 alkanes. While the rate coefficient values of C2-C4 alkanes vary between (0.25-2.12) x $10^{-12}$ $cm^3$ molecule$^{-1}$*

*s⁻¹ at 298K, the rate coefficient values of C5-C8 alkanes are between (3.6-8.9) x 10⁻¹² cm³*
*molecule⁻¹ s⁻¹ at 298K."*

**10. Page 19, Line 3–15: using the fraction of BTEX to assess the health risks may not be the best way since most guidelines use concentration as benchmark. For example, smoldering paddy stubble fire (13%) > diesel evaporative emissions (11%) does not necessarily indicate that the BTEX concentration in diesel evaporative emissions is less than smoldering paddy stubble fire. Please provide the concentration (with uncertainty) here as well to assist the discussion.**

**Author response:** We agree. As also mentioned in reply to reviewer 1's comment, we would like to caution that the absolute values we measured cannot be used for assessing ambient exposure except if one were to inhale the vapours or smoke directly. The reason is because our measured absolute values for a given sample (listed in the excel sheet attached with response) were collected through direct sampling of the vapours /smoke, without any/negligible ambient dilution. For the ambient traffic samples, measured at a traffic thoroughfare, we agree with the reviewer that this is helpful for assessing exposure as it represent concentrations for the traffic exhaust emissions after ambient dilution, and which people at street level would be inhaling. We include this information in the revised manuscript in the conclusion section as explained in reply to point 6 of reviewer 1.

**11. Page 33: the sample size for some NMHCs sources (e.g., paddy stubble burning, garbage burning, and traffic) are quite small (3 or 5). Please provide mixing ratios of the target compounds (together with uncertainties) in Figure 1 and Figure 2. If there are large uncertainties in the mixing ratios, please justify that such small sample size is representative of the sampling areas or even feasible to be extrapolated to represent South Asia.**

**Author response:** This point has been addressed already in detail to reviewer 1's points so we refrain from repetition.

**3 Technical corrections**

**1. Page 2, Line 13: "PMF": please give the full name of any acronym when it appears for the first time in the manuscript.**

**Author response:** We regret for not providing the relevant full name of the acronym appearing for the first time in the manuscript. PMF refers to the EPA's Positive Matrix Factorization model which is a mathematical receptor model used for determining the contribution of different emission sources to the ambient air plumes. The text has been modified accordingly:

Page 2 line 13:

*"Based on the measured source profiles, chemical tracers were identified for distinguishing varied emission sources and also for use in PMF (Positive Matrix Factorization) source apportionment models."*

**2. Page 2, Line 15: "LPG": please give the full name here.**

**Author response:** We regret for not providing the relevant full name of the acronym appearing for the first time in the manuscript. LPG refers to Liquefied Petroleum gas. The text has been modified accordingly:

Page 2 line 15:

*"Thus, we were able to identify chemical tracers such as i-pentane for petrol vehicular exhaust and evaporative emissions, propane for LPG (Liquefied Petroleum gas) evaporative and LPG vehicular exhaust emissions, and acetylene for the biomass fires during the flaming stage."*

**14. Page 3, Line 1: "BTEX": please list all the compounds in BTEX.**

BTEX refers to sum of benzene, toluene, ethylbenzene and xylenes. The text in the manuscript has been modified accordingly.

Page 5 Line 14-16:

*"Further, we assessed the secondary pollutant formation potential and health risks of the sources in terms of their OH reactivity (s-1), ozone formation potential (OFP, gO3/gNMHC) and fractional BTEX (sum of benzene, toluene, ethylbenzene and xylenes) content."*

**15. Page 3, Line 2: "most polluting": please provide data to support this conclusion.**

**Author response:** The impact of sources over the regional atmospheric chemistry and human health were assessed in form of ozone formation potential (OFP) and fraction of BTEX in the emissions. As mentioned in reply to point 13 and 15 of reviewer 1, the statistical differences in the average OFP and BTEX fraction between the different emission sources were ascertained by Tukey's pairwise honestly significant difference test and the summary for the same is provided in new Table S5 and S6. Statistically significant differences between the sources with ≥2 σ confidence were found between the sources and are also highlighted in the table S5 and S6. Based upon these results the term "most polluting" was used.

**16. Page 4, Line 1–2: "North West-Indo Gangetic Plain" ! "North West-Indo Gangetic Plain (NW-IGP)".**

**Author response:** The text has been modified accordingly:

Page 4 line 1-2:

*"Every year the North West-Indo Gangetic Plain (NW-IGP) experiences episodes of large scale open burning of paddy stubble…"*

**17. Page 9, Line 21: please define "pAs" here.**

**Author response:** The FID signal of a compound is recorded in form of current (picoampere, pA) by the instrument. While generating the chromatogram the y-axis corresponds to the measured signal (peak height) in pA and x-axis corresponds to the retention time (in minutes). The area under the peak is calculated and expressed in units of pAs (picoampere seconds) and used to quantify the analyte. The text has been modified accordingly as replied to point 3.

**18. Page 16, Line 6: please provide the full name for BSV and BSVI.**

**Author response:** BSV and BSVI stand for Bharat Stage V and Bharat Stage VI. In order to tackle the problem of increasing vehicular pollution and bring the Indian motor vehicle regulations in alignment with European Union regulations, the Indian government decided to apply BSVI standards w.e.f. April 2020. These are the new emission norms set up by the Government of India (GoI, 2016) and to be met by all on road motor vehicles (including the three wheelers) and the

fuels used therein. BS VI proposal clearly specifies the mass emission standards and commercial fuel specifications for each vehicle category and sub-classes.(ARAI, 2018). The text has been modified accordingly:

Page 16 line 5-6:

*"Therefore, in order to mitigate the emissions the use of improved technologies (for better combustion and emission reduction like catalytic convertors), cleaner fuels (Bharat Stage V (BS V) and Bharat Stage VI (BS VI) and reduced idling times of the vehicles should be encouraged."*

Additional remarks by authors with compiled list of changes to Abstract, Figures, Tables and supplementary material :

Reviewers were very kind to provide suggestions for improved presentation of the results of this work. Following their advice we have made the following minor changes to figures and tables in the main manuscript and supplement.

Abstract has been re-written to highlight the new results better.

Figure 1: New figure was constructed using the averaged normalised values of each sample in the emission sources and standard error of mean was used to put the error bars to show the overall uncertainty and sample variability.

Figure 2: New figure was constructed using the averaged normalised values of each sample in the emission sources and standard error of mean was used to put the error bars to show the overall uncertainty and sample variability.

Figure 3: New figure was constructed with changed colours for better visualisation and overall uncertainty in OFP and BTEX fraction was denoted with the error bars.

Table 1: Sample details of the fuel evaporative emissions were added.

Table 2: The compounds were arranged now in class wise order for more logical representation. Results of the precision experiment conducted at 1ppb were also added.

Table 3: In view of the new results, the average molar emission ratios were updated with the standard error of mean to represent the overall uncertainty.

Figure S2: A new figure S2 was added to show the instrument schematic as per second reviewer's request.

Figure S3: The old Figure S2 was modified for better visualisation and representation of the overall dynamic range of the instrument spanning over two order of magnitude of measurement (2-200 ppb).

Figure S4: A typical chromatogram showing all the target compounds measured during a standard gas calibration experiment using VOC gas standard 1 (Chemtron Science Laboratories Pvt. Ltd., Navi Mumbai, India).

Figure S5: Originally Figure S3 was modified with the incorporation of the results of new samples. The average normalised OH reactivity of the emission sources and overall uncertainty were expressed in form of standard error of mean as errors bars.

Figure S6: Originally Figure S4 was modified by providing the average normalised OH reactivity of the emission sources and overall uncertainty was expressed in form of standard error of mean as errors bars.

Table S2: A new Table S2 was added to show the compounds with mixing ratios > 50ppbv in different emission sources after the dilution during sample analysis in the TD-GC-FID. The compounds were arranged now in class wise order for more logical representation.

Table S3: Originally Table S2 was modified so as to arrange the compounds in class wise order for more logical representation.

Table S4: Originally Table S3 was modified so as to arrange the compounds in class wise order for more logical representation.

Tables S5: A new Table S5 was added to show the summary of the results of Tukey's pairwise HSD (honest significant for difference of mean) statistical test to ascertain any differences between the average OFP of different sources.

Table S6: A new Table S6 was added to show the summary of the results of Tukey's pairwise HSD (honest significant for difference of mean) statistical test to ascertain any differences between the average BTEX% of different sources.

Additionally, following electronic supplements are provided:

1. A pdf file containing the typical chromatograms of calibration and emission source samples measured with the instrument. Supplementary material 2
2. An excel file containing the actual concentration of the NMHCs in the emission sources, diluted concentration of NMHCs in each sample at the time of measurement and the concentration of NMHCs in the ambient backgrounds at the time of sample collection. Supplementary material 3

[revised manuscript text omitted]
, 30.691$^o$ N, 76.698$^o$ E; Sector 79/80 Chowk, 30.678$^o$ N, 76.721$^o$ E; and Transport Chowk, 30.717$^o$ N, 76.812$^o$ E) from 3-15 March 2017. Although the vehicular emissions are known to be dependent upon several factors, their idling operation results in particularly higher emissions and fuel residues in the exhaust (Yamada et al., 2011; Shancita et

al., 2014). This is because in idling operations the engine does not work at its peak operating temperature and efficiency (Brodrick et al., 2002) resulting in incomplete fuel combustion (Rahman et al., 2013). In this study, prior to vehicular exhaust sampling, the engine was left running for about 5 minutes until it warmed up to normal working temperature (70-90$^o$C) and then the air was sampled directly from the mouth of the tailpipe exhaust with the car being in stationary position and engine left running at idle speed. The idling vehicular exhaust samples were collected from 23 Petrol vehicles (14 two wheelers and 9 light duty four wheelers), 33 diesel vehicles (6 three wheelers, 12 light duty four wheeler and 15 heavy duty wheelers), 9 LPG vehicles (three wheelers) and 7 CNG vehicles (6 three wheelers and 1 light duty four wheeler) from Mar 2017-Oct 2018 in Chandigarh and Mohali (30.660-30.750$^o$N, 76.700-76.840$^o$E). For a better representation, the most common vehicle models on Indian roads were selected for this study based upon the personal field observations and motor vehicle data provided by Ministry of Road Transport and Highways (MoRTH 2017). The fuel evaporative emissions samples (one ten each from the headspace of LPG, petrol and diesel) were collected in the IISER Mohali campus on 5 Jul 2017Mohali, Chandigarh and Panchkula on 13-14Aug 2020. In India, the commonly used LPG is of two type: domestic LPG for household cooking and commercial LPG for various commercial and industrial applications like hotels, restaurants, metallurgical applications, textiles, automotive etc. Out of the total 10 samples of LPG evaporative emissions, 5 samples each were of domestic and commercial LPG types. For better representation, the samples of evaporative emissions were also collected from the most common brands of petrol, diesel and LPG fuels sold all over India and in Nepal, Bangladesh and Sri Lanka (Indian oil, Hindustan Petroleum, Bharat Petroleum, Bharatgas and Indane). The average ambient temperature and relative humidity during the sample collection were 30$^o$C and 75% respectively. 
[revised manuscript text omitted]

5      1. Methylcyclopentane, 2. 2,4-Dimethylpentane, 3. Benzene, 4. Cyclohexane, 5. 2-Methylhexane, 6. 2,3-Dimethylpentane, 7. 2-Methylhexane, 8. *n*-Heptane, 9. Toluene, 10. 2-Methyheptane, 11. 3-Methylheptane, 12. *n*-Octane, 13. Ethylbenzene, 14. *m/p*-Xylene, 15. Styrene, 16. *o*-Xylene, 17. *i*-Propylbenzene, 18. *n*-Propylbenzene, 19. *m*-Ethyltoluene, 20. *p*-Ethyltoluene, 21.1,3,5-Trimethylbenzene, 22. *o*-Ethyltoluene, 23. 1,2,4-Trimethylbenzene, 24. 1,2,3-Trimethylbenzene, 25. *m*-Diethylbenzene, 26. *p*-Diethylbenzene , 27. Ethane, 28. Ethene, 29. Propane, 30. Propene, 31. *i*-Butane, 32. *n*-Butane, 33. Acetylene, 34. *trans*-2-Butene, 35.

[Figure]

1-Butene, 36. *cis*-2-Butene, 37. Cyclopentane, 38. *i*-Pentane, 39. *n*-Pentane, 40. *trans*-2-Pentene, 41. 1-Pentene, 42. *cis*-2-Pentene, 43. 2,2-Dimethylbutane, 44. 2,3-Dimethylbutane, 45. 2-Methylpentane, 46. 3-Methylpentane, 47. *n*-Hexane, 48. Isoprene, 49. 1-Hexene.

**Figure S4:** A typical chromatogram showing all the target compounds measured during a standard gas calibration experiment using VOC gas standard 1 (Chemtron Science Laboratories Pvt. Ltd., Navi Mumbai, India). NMHCs (2,2,4-trimethylpentane, 2,3,4-trimethylpentane and methylcyclohexane) which were well resolved and separated in calibration chromatograms but poorly resolved in the analysis of some point source samples are highlighted as (*) and were excluded from this study.

[Figure]

[Figure]

**Figure S5:** Normalised profiles of calculated OH reactivity (s⁻¹) in **a)** Paddy stubble burning: Flaming; **b)** Paddy stubble burning: Smouldering; **c)** Garbage burning: Flaming; **d)** Garbage

burning: Smouldering; **e)** Commercial LPG evaporative emissions; **f)** Domestic LPG evaporative emissions; **g)** Petrol evaporative emissions; **h)** Diesel evaporative emissions. The grey colour highlights the aromatics, red colour highlights the alkenes and alkyne and the yellow colour highlights the alkanes. The OH reactivity of the NMHCs are normalised to the NMHC with the maximum OH reactivity in the respective source sample as:

$$f = [Y_i]/[Y_{max}]$$

Where, $[Y_i]$ is the NMHC OH reactivity and $[Y_{max}]$ is the NMHC with the maximum OH reactivity in the respective source sample. Error bars represent the standard error of averaged Normalised ratio.

[Figure]

**Figure S6:** Normalised profiles of calculated OH reactivity (s$^{-1}$) in **a)** CNG vehicular exhaust; **b)** LPG vehicular exhaust; **c)** Petrol two wheeler exhaust; **d)** Petrol four wheeler exhaust; **e)** Diesel three wheeler vehicular exhaust; **f)** Diesel four wheeler vehicular exhaust **g)** Diesel heavy duty

vehicle (HDV) exhaust **h)** Traffic, derived from the TD-GC-FID measurements. Error bars represent the standard error of averaged normalised ratio. The grey colour highlights the aromatics, red colour highlights the alkenes and alkyne and the yellow colour highlights the alkanes.

**Table S1.** Parameters for the GC oven temperature ramp program and FIDs (Flame ionisation detectors).

| Parameter | Value |
| --- | --- |
| **Initial temp** | 30 ºC |
| **Initial hold time** | 12 min |
| **Rate 1** | 5 ºC/min |
| **Final temperature 1** | 170 ºC |
| **Hold time 1** | 1 min |
| **Rate 2** | 15 ºC/min |
| **Final temperature 2** | 200 ºC |
| **Hold time 2** | 1 min |
| **Total Run time** | 44 min |
| **FID Temperature** | 250 ºC |
| **FID Air flow** | 400 mL/min |
| **FID H$_2$ fuel flow** | 40 mL/min |
| **FID N$_2$ makeup flow** | 20 mL/min |

5 **Table S2:** Average concentrations (ppb) and standard deviation (in parentheses) of the compounds >50ppb after dilution.

| Petrol 2 Wheelers (n = 14) | | Petrol 4 Wheelers (n = 9) | | Diesel HDVs (n = 15) | |
|---|---|---|---|---|---|
| **Compounds** | **diluted concentration** | **Compounds** | **diluted concentration** | **Compounds** | **diluted concentration** |
| Toluene | 96.3 (45.7) | Ethane | 81.8 (62.28) | Ethene | 150.4 (35.16) |
| m/p-Xylene | 55.1 (32.2) | i-Pentane | 72.9 (71.55) | Propene | 59.5 (14.26) |
| Acetylene | 95.6 (74.7) | | | | |
| Ethene | 67.5 (48.9) | | | | |
| i-Pentane | 87.2 (56.3) | | | | |
| Toluene | 96.3 (45.8) | | | | |
| m/p-Xylene | 55.1 (32.2) | | | | |
| **Diesel 4 Wheelers (n = 12)** | | **Diesel 3 Wheelers (n = 7)** | | **LPG Vehicles (n = 9)** | |
| Acetylene | 58.9 (37.9) | Acetylene | 51.7 (25.5) | Propene | 107.5 (75.6) |
| Ethene | 164.6 (27.2) | Ethene | 78.1 (18.3) | trans-2-Butene | 71.8 (35.2) |
| Propene | 53.2 (21.8) | | | 1-Butene | 55.2 (40.8) |
| | | | | Propane | 105.5 (57.6) |
| | | | | n-Butane | 132.8 (43.0) |
| | | | | i-Butane | 93.6 (37.7) |
| | | | | Propene | 107.5 (75.6) |
| **CNG Vehicles (n=7)** | | **Paddy fires Flaming (n=3)** | | **Paddy fires Smouldering (n=3)** | |
| Ethane | 166.1 (17.8) | Ethene | 65.4 (23.5) | Ethane | 132.6 (97.67) |
| **Garbage fires Flaming (n=5)** | | **Garbage fires Smouldering (n=5)** | | **Traffic (n=3)** | |
| Acetylene | 76.9 (56.6) | Ethene | 59.2 (65.8) | None | |
| Ethene | 83.8 (54.7) | | | | |
| Propene | 61.2 (35.3) | | | | |
| **Domestic LPG evaporative (n=5)** | | **Commercial LPG evaporative (n=5)** | | **Petrol evaporative (n=10)** | **Diesel evaporative (n=10)** |
| Propane | 147.4 (48.7) | trans-2-Butene | 52.5 (20.6) | None | None |
| n-Butane | 55.1 (22.5) | n-Butane | 127.9 (38.9) | | |
| i-Butane | 53.6 (21.2) | i-Butane | 62.6 (22.4) | | |

**Table S3:** Details of VOC gas standards used in calibration experiments. Mixing ratios reported here are in ppm.

VOC Standard 1: Chemtron Science Laboratories Pvt. Ltd., Navi Mumbai, India.

VOC Standard 2: Apel Riemer Environmental. Inc., Colorado, USA.

| Compounds | Mixing ratios in VOC Standard 1 | Mixing ratios in VOC Standard 2 | Compounds | Mixing ratios in VOC Standard 1 | Mixing ratios in VOC Standard 2 |
|---|---|---|---|---|---|
| **Aromatics (n=16)** | | | | | |
| Benzene | 1.03 | 0.49 | *p*-Ethyltoluene | 1.01 | |
| Toluene | 1.01 | 0.47 | 1,2,3-Trimethylbenzene | 1.01 | |
| Styrene | 0.99 | | 1,2,4-Trimethylbenzene | 0.99 | |
| *m/p*-Xylene | 2.06 | | 1,3,5-Trimethylbenzene | 1.03 | |
| *o*-Xylene | 1.01 | | *i*-Propylbenzene | 1.01 | |
| Ethylbenzene | 1.01 | | *n*-Propylbenzene | 0.99 | |
| *m*-Ethyltoluene | 1.01 | | *m*-Diethylbenzene | 1.02 | |
| *o*-Ethyltoluene | 1.03 | | *p*-Diethylbenzene | 1.01 | |
| **Alkyne (n=1)** | | | | | |
| Acetylene | 0.99 | | | | |
| **Alkenes (n=10)** | | | | | |
| Ethene | 0.99 | | Isoprene | 1.01 | 0.48 |
| Propene | 0.99 | | 1-Pentene | 1.00 | |
| 1-Butene | 0.99 | | *trans*-2-Pentene | 1.04 | |
| *trans*-2-Butene | 0.99 | | *cis*-2-Pentene | 1.01 | |
| *cis*-2-Butene | 0.99 | | 1-Hexene | 1.01 | |
| **Alkanes (n=22)** | | | | | |
| Ethane | 0.99 | | *n*-Hexane | 0.99 | |
| Propane | 0.99 | | 2-Methylpentane | 0.99 | |
| *n*-Butane | 0.99 | | 3-Methylpentane | 0.99 | |
| *i*-Butane | 0.99 | | 2-Methylhexane | 1.01 | |
| *i*-Pentane | 0.98 | | 3-Methylhexane | 0.99 | |
| *n*-Pentane | 1.01 | | 2,3-Dimethylpentane | 1.01 | |
| Cyclopentane | 1.04 | | 2,4-Dimethylpentane | 1.01 | |
| Cyclohexane | 1.01 | | *n*-Heptane | 1.03 | |
| Methylcyclopentane | 1.01 | | *n*-Octane | 1.01 | |
| 2,2-Dimethylbutane | 1.01 | | 2-Methylheptane | 1.03 | |
| 2,3-Dimethylbutane | 0.99 | | 3-Methylheptane | 1.03 | |

TMB: Trimethylbenzene

**Table S4**: Average sensitivity factors (pAs/ppb) and standard deviation (pAs/ppb) for 49 NMHCs obtained from thirteen calibration experiments performed between December 2016 and October 2018 using the calibration standard 1 (Chemtron Science Laboratories Pvt. Ltd., Navi Mumbai, India).

| Compounds | Average sensitivity (pAs/ppb) | Standard deviation (pAs/ppb) | Compounds | Average sensitivity (pAs/ppb) | Standard deviation (pAs/ppb) |
|---|---|---|---|---|---|
| **Aromatics (n=16)** | | | | | |
| Benzene | 67.8 | 5.6 | *p*-Ethyltoluene | 79.3 | 9.6 |
| Toluene | 74.6 | 6.6 | 1,2,3-Trimethylbenzene | 58.7 | 8.0 |
| Styrene | 81.2 | 8.2 | 1,2,4-Trimethylbenzene | 78.8 | 10.6 |
| *m/p*-Xylene | 79.8 | 8.3 | 1,3,5-Trimethylbenzene | 76.6 | 9.3 |
| *o*-Xylene | 91.8 | 9.6 | *i*-Propylbenzene | 90.5 | 9.2 |
| Ethylbenzene | 74.8 | 7.3 | *n*-Propylbenzene | 85.0 | 9.4 |
| *m*-Ethyltoluene | 79.2 | 9.3 | *m*-Diethylbenzene | 68.6 | 9.7 |
| *o*-Ethyltoluene | 80.1 | 9.5 | *p*-Diethylbenzene | 63.5 | 10.0 |
| **Alkyne (n=1)** | | | | | |
| Acetylene | 5.2 | 0.9 | | | |
| **Alkenes (n=10)** | | | | | |
| Ethene | 24.8 | 2.6 | Isoprene | 53.2 | 4.9 |
| Propene | 35.8 | 3.4 | 1-Pentene | 57.8 | 5.1 |
| 1-Butene | 41.4 | 5.1 | *trans*-2-Pentene | 56.6 | 5.3 |
| *trans*-2-Butene | 40.1 | 5.0 | *cis*-2-Pentene | 56.4 | 5.3 |
| *cis*-2-Butene | 40.3 | 4.8 | 1-Hexene | 45.9 | 4.1 |
| **Alkanes (n=22)** | | | | | |
| Ethane | 20.7 | 1.9 | *n*-Hexane | 62.4 | 6.1 |
| Propane | 35.9 | 3.2 | 2-Methylpentane | 73.5 | 7.1 |
| *n*-Butane | 46.4 | 4.5 | 3-Methylpentane | 81.4 | 7.7 |
| *i*-Butane | 45.3 | 4.4 | 2-Methylhexane | 86.5 | 7.3 |
| *i*-Pentane | 56.4 | 5.4 | 3-Methylhexane | 78.8 | 6.8 |
| *n*-Pentane | 58.4 | 5.5 | 2,3-Dimethylpentane | 74.4 | 6.8 |
| Cyclopentane | 51.5 | 4.5 | 2,4-Dimethylpentane | 79.2 | 6.6 |
| Cyclohexane | 71.0 | 5.5 | *n*-Heptane | 77.3 | 6.9 |
| Methylcyclopentane | 68.9 | 5.6 | *n*-Octane | 86.1 | 8.5 |
| 2,2-Dimethylbutane | 70.6 | 6.6 | 2-Methylheptane | 85.7 | 7.4 |
| 2,3-Dimethylbutane | 70.9 | 6.7 | 3-Methylheptane | 86.3 | 8.2 |

**Table S5:** Summary of Tukey pairwise HSD (honestly significant difference) test results performed for the averaged OFP values from the different emission sources. The significant differences in the mean values at confidence interval > 95% are ascertained by p (same mean) < 0.05 and are highlighted in bold.

| | Diesel vehicles | Paddy smouldering | Paddy flaming | LPG vehicles | Garbage flaming | Garbage smouldering | Petrol vehicles | LPG evaporative | Diesel evaporative | Petrol evaporative | CNG vehicles |
|---|---|---|---|---|---|---|---|---|---|---|---|
| Diesel vehicles | - | - | σ | - | **2σ** | **3σ** | **4σ** | **4σ** | **4σ** | **4σ** | **4σ** |
| Paddy smouldering | | - | - | - | - | σ | **2σ** | **2σ** | **4σ** | **4σ** | **4σ** |
| Paddy flaming | | | - | - | - | - | σ | - | **2σ** | **4σ** | **4σ** |
| LPG vehicles | | | | - | - | σ | **2σ** | **2σ** | **3σ** | **4σ** | **4σ** |
| Garbage flaming | | | | | - | - | - | - | σ | **4σ** | **4σ** |
| Garbage smouldering | | | | | | - | - | - | - | **4σ** | **4σ** |
| Petrol vehicles | | | | | | | - | - | - | **2σ** | **4σ** |
| LPG evaporative | | | | | | | | - | - | - | **4σ** |
| Diesel evaporative | | | | | | | | | - | σ | **3σ** |
| Petrol evaporative | | | | | | | | | | - | - |
| CNG vehicles | | | | | | | | | | | - |

**Table S6:** Summary of Tukey pairwise HSD (honestly significant difference) test results performed for the averaged BTEX% from the different emission sources. The significant differences in the mean values at confidence interval > 95% are ascertained by p (same mean) < 0.05 and are highlighted in bold.

| | Petrol vehicles | Garbage smouldering | Garbage flaming | Paddy flaming | Diesel vehicles | Paddy smouldering | Diesel evaporative | Petrol evaporative |
|---|---|---|---|---|---|---|---|---|
| Petrol vehicles | - | - | - | - | **2σ** | **4σ** | **2σ** | **4σ** |
| Garbage smouldering | | - | - | **2σ** | **4σ** | **4σ** | **4σ** | **4σ** |
| Garbage flaming | | | - | - | **2σ** | **3σ** | σ | **4σ** |
| Paddy flaming | | | | - | - | σ | - | **4σ** |
| Diesel vehicles | | | | | - | - | - | σ |
| Paddy smouldering | | | | | | - | - | - |
| Diesel evaporative | | | | | | | - | **2σ** |
| Petrol evaporative | | | | | | | | - |